# Evolution of the germline mutation rate across vertebrates

Lucie A. Bergeron[1✉], Søren Besenbacher[2], Jiao Zheng[3,4], Panyi Li[3], Mads Frost Bertelsen[5], Benoit Quintard[6], Joseph I. Hoffman[7,8], Zhipeng Li[9], Judy St. Leger[10], Changwei Shao[11], Josefin Stiller[1], M. Thomas P. Gilbert[12,13], Mikkel H. Schierup[14] & Guojie Zhang[1,15,16,17✉]

The germline mutation rate determines the pace of genome evolution and is an evolving parameter itself[1]. However, little is known about what determines its evolution, as most studies of mutation rates have focused on single species with different methodologies[2]. Here we quantify germline mutation rates across vertebrates by sequencing and comparing the high-coverage genomes of 151 parent–offspring trios from 68 species of mammals, fishes, birds and reptiles. We show that the per-generation mutation rate varies among species by a factor of 40, with mutation rates being higher for males than for females in mammals and birds, but not in reptiles and fishes. The generation time, age at maturity and species-level fecundity are the key life-history traits affecting this variation among species. Furthermore, species with higher long-term effective population sizes tend to have lower mutation rates per generation, providing support for the drift barrier hypothesis[3]. The exceptionally high yearly mutation rates of domesticated animals, which have been continually selected on fecundity traits including shorter generation times, further support the importance of generation time in the evolution of mutation rates. Overall, our comparative analysis of pedigree-based mutation rates provides ecological insights on the mutation rate evolution in vertebrates.

Germline mutations are the proximate source of genomic innovation and inherited diseases[4]. Consequently, considerable effort has been spent on characterizing the molecular processes underlying these mutations and estimating germline mutation rates (GMRs). Mutations are rare events, yet the frequency at which they are introduced into genomes at each generation varies considerably across taxa, from approximately $10^{-11}$ mutations per site per generation in unicellular eukaryotes up to approximately $10^{-7}$ mutations per site per generation in multicellular eukaryotes[1,5,6]. Inferring the driving forces of GMR evolution has important implications for understanding the mechanisms underlying mutagenesis. Several hypotheses have been proposed to explain variation in GMRs among lineages. Some of these invoke molecular mechanisms such as DNA methylation[7] or microsatellite instability[8], whereas others invoke external factors such as exposure to mutagenic environments[9]. Other studies have argued that life-history traits might explain some of the variation both in the prevalence of mutations and in the ability to repair DNA. In particular, the generation time[10] and the metabolic rate[11] have been suggested to be key life-history traits that could be associated with germline mutations.

From a long-term evolutionary perspective, the 'drift barrier hypothesis' proposes that lower mutation rates may reflect the increased efficiency of natural selection at reducing the occurrence of mutations in species with large effective population sizes[3].

However, a lack of accurate and standardized GMR estimation has so far precluded testing current hypotheses of GMR evolution. Pedigree-based estimates of GMRs per generation have recently been published for a handful of vertebrate species, mainly focusing on humans and primates[12–17]. Furthermore, a recent comparative study of 16 mammalian species identified an effect of lifespan on somatic mutation rates inferred from the sequencing of intestinal crypts[18]. Nevertheless, interspecific comparisons of GMR variation remain restricted in taxonomic scope[19], partly due to the difficulty of comparing GMR estimates derived using different methodologies[2]. For example, alternative bioinformatic pipelines used in different studies can yield GMR estimates that vary by a factor of two, even when applied to the same parent–offspring trios[2]. This highlights the importance of applying consistent analytical pipelines for interspecies comparisons of GMRs. We therefore generated high-depth genome sequences

[1]Villum Centre for Biodiversity Genomics, Section for Ecology and Evolution, Department of Biology, University of Copenhagen, Copenhagen, Denmark. [2]Department of Molecular Medicine, Aarhus University, Aarhus, Denmark. [3]BGI-Shenzhen, Shenzhen, China. [4]BGI Education Center, University of Chinese Academy of Sciences, Shenzhen, China. [5]Copenhagen Zoo, Frederiksberg, Denmark. [6]Parc Zoologique et Botanique de Mulhouse, Mulhouse, France. [7]Department of Animal Behaviour, Bielefeld University, Bielefeld, Germany. [8]British Antarctic Survey, High Cross, Cambridge, UK. [9]College of Animal Science and Technology, Jilin Agricultural University, Changchun, China. [10]Department of Biomedical Sciences, Cornell University, Ithaca, NY, USA. [11]Key Lab of Sustainable Development of Marine Fisheries, Ministry of Agriculture and Rural Affairs, Yellow Sea Fisheries Research Institute, Chinese Academy of Fishery Sciences, Qingdao, China. [12]Center for Evolutionary Hologenomics, The GLOBE Institute, University of Copenhagen, Copenhagen, Denmark. [13]University Museum, NTNU, Trondheim, Norway. [14]Bioinformatics Research Centre, Aarhus University, Aarhus, Denmark. [15]Centre for Evolutionary & Organismal Biology, Women's Hospital, Zhejiang University School of Medicine, Hangzhou, China. [16]Liangzhu Laboratory, Zhejiang University Medical Center, Hangzhou, China. [17]State Key Laboratory of Genetic Resources and Evolution, Kunming Institute of Zoology, Chinese Academy of Sciences, Kunming, China. ✉e-mail: lucie.a.bergeron@gmail.com; guojiezhang@zju.edu.cn

(average coverage of more than 67×) for 323 individuals representing 151 trios of 68 vertebrate species, including 36 mammals, 18 birds, 8 ray-finned fishes and 6 reptiles (Supplementary Table 1). We then quantified species-specific GMRs across this wide range of vertebrate taxa using consistent bioinformatics pipelines to test long-standing evolutionary hypotheses on GMR evolution.

## Per-generation mutation rate variation

We first estimated the per generation GMR ($\mu_{generation}$) for each trio (that is, mother, father and offspring) by comparing parental and offspring genomes (Fig. 1a, Supplementary Tables 2 and 3 and Supplementary Figs. 1–5 for details on the method). Overall, $\mu_{generation}$ varies by a factor of 40 across all species. On average, mutation rates per generation are higher in reptiles (average of all species $1.17 \times 10^{-8}$, 95% CI of the mean = $5.34 \times 10^{-9}$ to $1.80 \times 10^{-8}$) and birds (average of all species $1.01 \times 10^{-8}$, 95% CI of the mean = $6.10 \times 10^{-9}$ to $1.42 \times 10^{-8}$) than in mammals (average of all species $7.97 \times 10^{-9}$, 95% CI of the mean = $7.04 \times 10^{-9}$ to $8.90 \times 10^{-9}$) and fishes (average of all species $5.97 \times 10^{-9}$, 95% CI of the mean = $4.39 \times 10^{-9}$ to $7.55 \times 10^{-9}$). However, the difference among the four major classes of vertebrates is not overall statistically significant (analysis of variance (ANOVA): $F = 1.86$, $P = 0.15$). Furthermore, the amount of variation in $\mu_{generation}$ among species tends to be higher for birds and lower for mammals and fishes (Fig. 1a), although this variation is arguably modest given large differences in life-history traits among these species (for example, there is a 2.8 million-fold difference in the body mass of killer whales and Siamese fighting fish, and there is a 93-fold difference in the generation time between humans and Texas banded geckos).

Species with longer generation intervals are expected to have higher per-generation mutation rates due to a combination of a larger number of cell divisions in spermatogenesis and more time for DNA damage to accumulate[12–14,20]. For the 105 trios for which parental age was known at reproduction, we found a significant positive association between $\mu_{generation}$ and the average parental age at reproduction (linear regression adjusted $r^2 = 0.14$, $P = 3.9 \times 10^{-5}$; Fig. 1b). This pattern is also significant for the 60 mammalian trios with known parental ages (linear regression adjusted $r^2 = 0.37$, $P = 1.6 \times 10^{-7}$) and for the 32 bird trios after excluding a single outlier, the Darwin's rhea (linear regression adjusted $r^2 = 0.31$, $P = 0.0005$). Furthermore, all three of these regressions have similar positive $y$-intercept values on the order of approximately $0.59 \times 10^{-8}$ mutations per site per generation. For the trios with known parental ages, paternal and maternal ages at conception are strongly correlated (linear regression adjusted $r^2 = 0.77$, $P < 2.2 \times 10^{-16}$; Extended Data Fig. 1). However, multiple linear regression showed that the age of the father is the most significant explanatory variable (adjusted $r^2 = 0.15$, $P = 9.3 \times 10^{-5}$; paternal age $P = 0.018$; maternal age $P = 0.785$). Thus, a stronger effect of paternal than maternal age on the mutation rate seems to be universal for birds and mammals due to more germline mutations accumulating throughout the life of the male.

The specific types of de novo mutations (DNMs) observed across the 151 trios are concordant with the results of previous studies of individual species[12–14,21–25], including a ratio of transitions over transversions of 2.3 (95% CI on binomial distribution = 2.2–2.5) and a high proportion (48.5%, 95% CI on binomial distribution = 46.7–50.3%) of transitions from strong base pairing to weak base pairing (C:G > T:A) across all DNMs (Supplementary Table 4). Among C:G > T:A mutations, 42.4% (95% CI on binomial distribution = 39.9–45.0%) occurred at CpG sites. The direction of mutations from one base to another (that is, the spectrum of mutation) differed significantly across vertebrate classes ($\chi^2 = 30.0$, d.f. = 15, $P = 0.012$; Supplementary Table 4 and Supplementary Fig. 6). We also found significant differences among vertebrate classes for A > C mutations ($\chi^2 = 16.2$, d.f. = 3, $P = 0.001$) and for C > A mutations ($\chi^2 = 8.8$, d.f. = 3, $P = 0.032$). In particular, fish species exhibit significantly fewer A > C mutations and significantly more C > A mutations than the other vertebrate classes. However, this mutation pattern does not appear to

be associated with genome-wide CG content, as overall, the CG content of fishes is similar to that of mammals and birds and lower than that of reptiles (Supplementary Fig. 7). Finally, there is no significant difference between the classes of species in the percentage of all mutations located in CpG sites ($\chi^2 = 4.3$, d.f. = 3, $P = 0.23$), implying that high mutation rates at CpG sites are a conserved feature across vertebrates.

## Variable male-driven evolution

In mammals and birds, the much larger number of germ-cell divisions per generation in the male germ line leads to the expectation of a male mutation rate bias, coined the 'male-driven evolution hypothesis'[26,27]. However, very little is known about interspecific variation in the magnitude of the male-to-female ratio of the contribution of germline mutations ($\alpha$). Previous studies have reported high $\alpha$ values in mammals (ranging from 1.0 to 20.1)[28] and birds (ranging from 3.9 to 6.5)[29] based on indirect estimates obtained by comparing rates of sequence divergence on the autosomes and sex chromosomes (see Extended Data Fig. 2 and Supplementary Table 5). However, other evolutionary forces can also act differently on the X chromosome and autosomes. For example, stronger natural selection on the X chromosome could lead to lower than expected divergence from the common ancestor, upwardly biasing estimates of $\alpha$[28]. Furthermore, estimates of $\alpha$ derived in this way are averages over a phylogenetic branch and may thus differ from the contemporary species $\alpha$. Here we directly quantified $\alpha$ by assigning the parental origin of the DNMs. Around 48% of all 3,034 DNMs across all of the trios could be phased to their parental origin (see Supplementary Table 6 for positions of all mutations). Owing to the relatively small number of mutations in each trio (Supplementary Table 2), we analysed male bias after taxonomically grouping the species into classes and orders (Fig. 1c).

Mammals showed a male bias of $\alpha = 2.3$ (95% CI = 2.0–2.6). In general, our $\alpha$ estimates are in line with previous estimates derived for similar species based on genome alignments[30,31]. For example, we found that among mammals, primates have the largest male bias with $\alpha = 3.8$ (95% CI = 2.6–5.7), similar to what was previously reported for several species belonging to this group[12–14,21,22,32,33]. Rodents have the lowest male bias among the mammals in our study, with $\alpha = 2.1$ (95% CI = 1.4–3.1), consistent with a previous study based on mouse pedigrees[34]. This pattern can be explained by the short generation time of rodents, which leads to a smaller difference in cell divisions between the male and female germ lines[35]. However, the variation in $\alpha$ is relatively small given the variation in generation time among species (for example, between 30 years for humans and 8 months for the short-tailed opossum). Thus, an alternative hypothesis to explain the observed $\alpha$ would be a higher contribution of DNA damage, specifically in the male germ line for species with large generation times[31].

Birds also showed an overall high male bias with $\alpha = 3.2$ (95% CI = 2.5–4.1), although there is appreciable variation among different lineages. In particular, passerine birds and waterbirds (Pelecaniformes and Sphenisciformes) exhibited the largest male bias, both with $\alpha = 7.6$ (95% CI = 4.3–13.5 for Passeriformes and 95% CI = 3.5–16.3 for Pelecaniformes and Sphenisciformes). High levels of male–male competition will lead to an increased amount of sperm being produced and faster sperm turnover, which would be expected to cause a higher male bias[36]. Indeed, many passerine birds have large cloacal protuberances[37] and relatively heavy testes[38], which are often used as proxies of sperm competition[39]. For instance, in two of the passerine species included in our study, testes represent between 1.2% (for *Turdus merula*) and over 2% (for *Saxicola maurus*) of the total body mass[38]. Moreover, extra-pair mating is common in many passerine birds[40] as well as in penguins[41], also indicating a high level of sperm competition. Overall, our results lend further support to the male-driven hypothesis in birds and mammals[27].

By contrast, reptiles have a relatively small male bias with $\alpha = 1.5$ (95% CI = 1.2–1.8), whereas fishes appear to have a greater proportion of

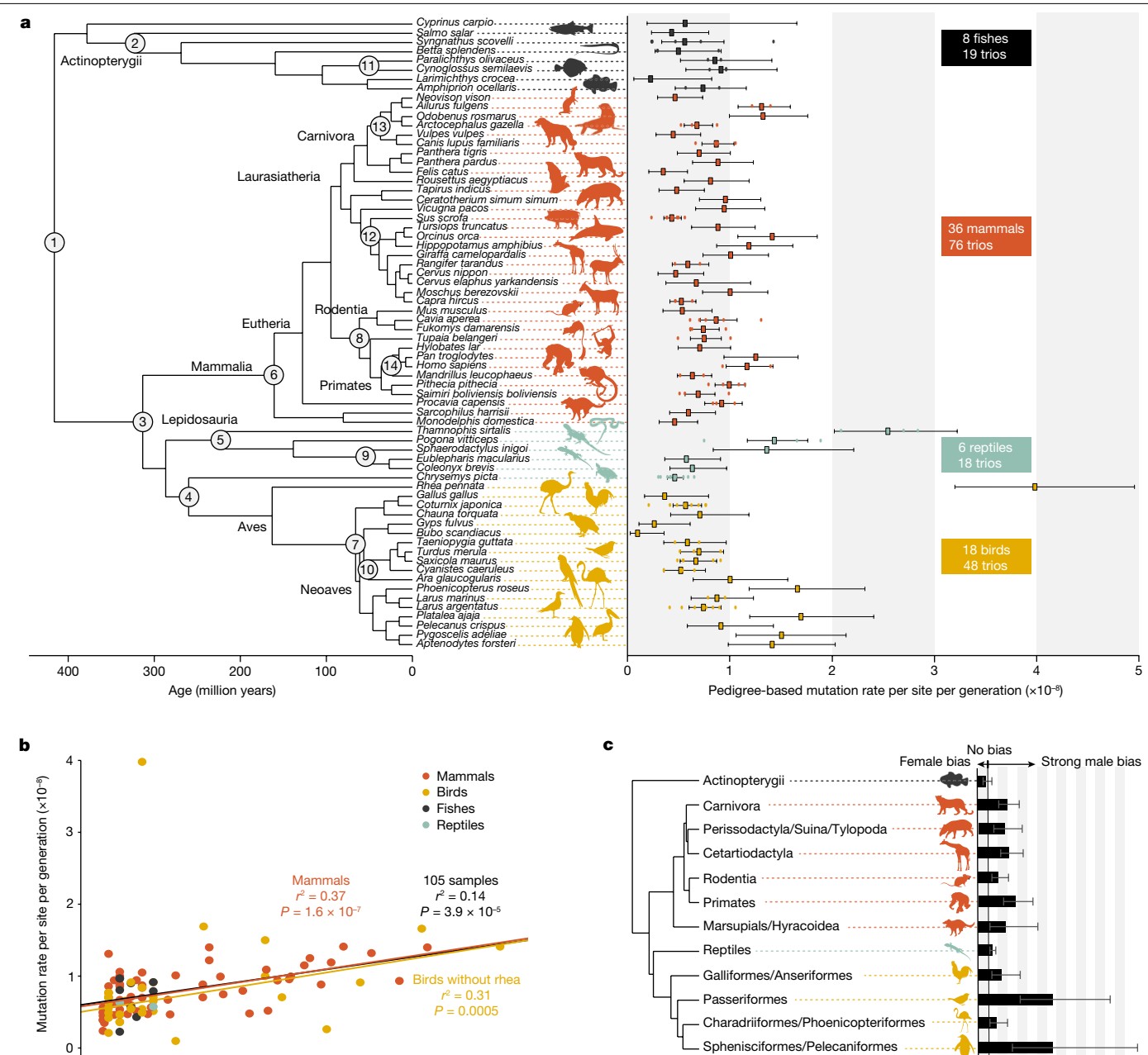

**Fig. 1 | Variation in GMRs and their association with life-history traits across 68 vertebrate species. a**, The phylogenetic tree of 68 species is based on UCE data and is calibrated with fossil data at 14 nodes (see Methods; Extended Data Fig. 3 and Supplementary Fig. 8). The average pedigree-based mutation rates per generation for each species, which are represented by the squares, show 40-fold variation among species. The 95% binomial confidence intervals are shown, and individual trios are represented by round points. See Supplementary Table 8 and Extended Data Fig. 4 for a comparison with published estimated rates of closely related species. **b**, The per-generation mutation rate is significantly associated with the average parental age at the time of offspring production across all individuals with known paternal age (105 trios), using linear regression. For birds, this relationship is statistically

significant after removing a single outlier, the Darwin's rhea. **c**, The male-to-female contribution ratio ($\alpha$) is estimated for groups of vertebrates having at least 30 mutations phased to their parents of origin in each group. The highest male bias (7.6:1) is found in two bird lineages, whereas fishes and reptiles show negligible male bias. The data are represented with 95% confidence intervals based on the binomial variance. The silhouette of *Syngnathus scovelli* was created by J.S. All other silhouettes are from PhyloPic (http://phylopic.org), except for one of the silhouettes of *Sarcophilus harrisii*, which was created by S. Werning, and the silhouette of *Pan troglodytes*, which was created by T. M. Keesey (vectorization) and T. Hisgett (photography); both are available under a CC-BY 3.0 licence (https://creativecommons.org/licenses/by/3.0).

mutations of maternal origin ($\alpha$ = 0.8), although the 95% CI overlaps 1 (95% CI = 0.5–1.4). This variation among vertebrate classes can be explained by differences in the process of gametogenesis. Although most birds and mammals produce sperm cells continuously through time[42], reptiles and fishes tend to be seasonal breeders, producing

sperm cells during a limited period before the mating season[43–45], which will tend to reduce differences in cell division numbers between males and females, leading to more equal $\alpha$. Moreover, female fishes are usually synchronous ovulators[46], producing hundreds to millions of eggs at the same time followed by a proliferation of new oogonia[47]. This

implies that females continually produce germ cells throughout their life, which would further reduce the difference in cell division number between males and females.

Species with lower sex bias also exhibited a larger proportion of shared mutations between siblings, with 12.0% (s.e. of 6.5%) of shared mutations between siblings for fish and 8.1% (s.e. of 5.3%) for reptiles compared with 1.5% (s.e. of 0.7%) for mammals and 2.2% (s.e. of 1.4%) for birds (Supplementary Table 7). An explanation for the repeated occurrence of those mutations is that they appear during the primordial germ cell specification in one of the parents[48]. The occurrence of primordial germ cell specification mutations is independent of parental sex. Consequently, a higher number of primordial germ cell specification mutations in some vertebrate groups could be an alternative explanation for the lower male-biased contribution to DNMs.

## Yearly mutation rates

To use our results for phylogenetic dating and to compare the speed of evolution among species with different generation times, we needed estimates of yearly mutation rates. Different methods have been used in the literature to estimate yearly mutation rates. When sample sizes are small, yearly rates are commonly inferred by dividing the per-generation rate by the average age of the parents (or the generation time if parental age is unknown)[49–51]. However, this method implicitly assumes a constant accumulation of mutations from conception to reproduction, that is, the regression line of mutation rate on parental age should run through the origin. Our results (Fig. 1b), as well as previous studies of mice, humans and cats[20,34], imply that parents always carry a minimum number of mutations in their gametes regardless of their age. This could lead to the yearly rate being overestimated for a given species if the sampled trio (or trios) had young parents compared with the average generation time for that species[52]. Consequently, we built a model that incorporates this mutational contribution at birth. Unfortunately, small per-species sample sizes in our dataset precluded modelling the effects of parental age separately for each species. However, we observed very similar intercepts and slopes across taxonomic groups, allowing us to fit a joint model for all species. A Poisson model explaining the number of mutations in each trio using a mutational contribution at birth and a weighted average of paternal and maternal age fits the data surprisingly well. To incorporate interspecific variation in male bias, we used the per-species fraction of paternal and maternal mutations estimated using read-backed phasing to weigh the average of the parental ages for each trio. Using this model, the number of predicted mutations matches the observed number with an overall $r^2$ of 0.73 (mammalian $r^2 = 0.58$, avian $r^2 = 0.51$; Supplementary Note 1).

The yearly rates inferred with the naive method of dividing the per-generation rate by parental age ($\mu_{yearly}$) and the rates obtained with our model ($\mu_{yearly\_modelled}$) yielded similar results (Pearson's correlation $r^2 = 0.40$, $P = 0.002$), and for 55% of the species, $\mu_{yearly}$ falls within the 95% confidence interval of the $\mu_{yearly\_modelled}$. As expected, the estimates showed the greatest differences for those species in which the parents reproduced far from the generation time, with the model-based estimates being smaller for those species that reproduced earlier than their generation time and larger for those species that reproduced later than their generation time. For example, the pigs in our dataset reproduced at around 6 months of age, which is more than 5 years earlier than the estimated generation time of this species. Thus, $\mu_{yearly} = 8.64 \times 10^{-9}$ mutations per site per year was potentially overestimated compared with the $\mu_{yearly\_modelled} = 1.05 \times 10^{-9}$ mutations per site per year at the generation time. Conversely, the yearly rate of the Texas banded gecko was potentially underestimated at $\mu_{yearly} = 3.17 \times 10^{-9}$ mutations per site per year using the reproductive age of 2 years of age from our dataset, whereas the modelled rate was $\mu_{yearly\_modelled} = 1.96 \times 10^{-8}$ mutations per site per year at a generation time of between 3 and 4 months. Both the naive method and the modelled method have been used in the literature

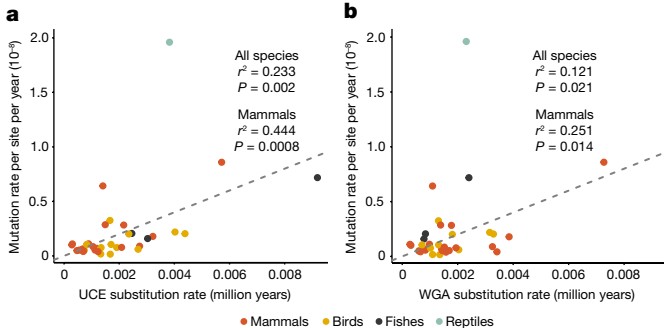

**Fig. 2 | GMRs are associated with long-term substitution rates. a,b,** There is a positive association between the modelled yearly pedigree-based mutation rates and the macroevolutionary substitution rates when using phylogenetic regression (PGLS) on both UCEs and their flanking sequences (**a**) and whole-genome alignments (WGAs) (**b**). The grey dashed lines indicate equality. See Extended Data Fig. 5 for plots of the same data on a log scale and Extended Data Fig. 6 for a comparison of UCE and WGA methods.

to estimate yearly rates and both have caveats owing to the underlying assumptions they require. Bearing this in mind, we decided to use $\mu_{yearly\_modelled}$ for the current analysis as we believe that this measure is more representative of the yearly rate at the generation time for each species (estimated yearly rates are provided in Supplementary Table 9 for comparison).

The estimated average $\mu_{yearly\_modelled}$ varies more than 120-fold among species (Supplementary Note 1 and Supplementary Table 9), with the highest $\mu_{yearly\_modelled}$ estimated for the Texas banded gecko at $1.96 \times 10^{-8}$ mutations per site per year (95% CI = $1.23 \times 10^{-8}$ to $2.83 \times 10^{-8}$), whereas the lowest $\mu_{yearly\_modelled}$ estimates were obtained for two bird species, the griffon vulture and the snowy owl, both with less than $0.18 \times 10^{-9}$ mutations per site per year (snowy owl: $\mu_{yearly\_modelled} = 0.16 \times 10^{-9}$, 95% CI = $0.05 \times 10^{-9}$ to $0.34 \times 10^{-9}$; griffon vulture: $\mu_{yearly\_modelled} = 0.17 \times 10^{-9}$, 95% CI = $0.07 \times 10^{-9}$ to $0.32 \times 10^{-9}$). This large amount of interspecific variation is remarkable given that pedigree-based GMR estimates of individual species assessed by previous separate studies only show an approximately 16-fold variation in yearly GMRs[34,51]. Within primates, we observed a twofold variation across species and found a general trend for rates to be higher in the New World monkeys than in the great apes. This is consistent with previous independent estimates from primates[19] and supports the 'hominoid slowdown' hypothesis[53–56].

Next, we used $\mu_{yearly\_modelled}$ to assess the strength of the association between GMRs and long-term evolutionary substitution rates. To obtain an estimate of the long-term substitution rate, we used the alignment of ultraconserved elements (UCEs), which are more likely to align among taxonomically distant species, plus 1,000 bp of flanking regions on each side of the UCE sequences, which will more closely reflect the neutral substitution rate[57]. We found a significant positive correlation between $\mu_{yearly\_modelled}$ and the UCE substitution rate after excluding domesticated species owing to their overall much higher yearly mutation rates (see the following section; phylogenetic generalized least squares (PGLS): adjusted $r^2 = 0.23$, $P = 0.002$; Fig. 2a). This pattern is especially pronounced for mammals (PGLS: adjusted $r^2 = 0.44$, $P = 0.0008$), even after removing the two outliers (PGLS: adjusted $r^2 = 0.32$, $P = 0.009$). We also found a significant relationship between $\mu_{yearly\_modelled}$ and the long-term substitution rate inferred using whole-genome alignments (PGLS: adjusted $r^2 = 0.12$, $P = 0.02$; Fig. 2b).

## Life-history traits shape GMR variation

To test various hypotheses relating to the causes of GMR variation among species, we tested for associations between the modelled mutation rate per generation ($\mu_{generation\_modelled}$) and life-history traits

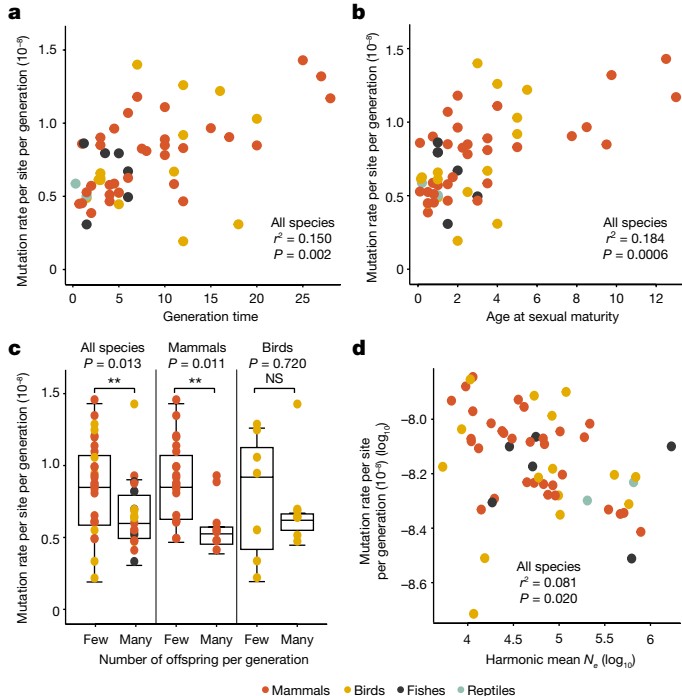

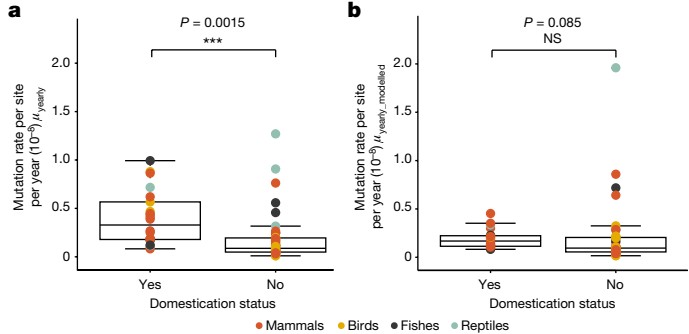

**Fig. 4 | The yearly GMRs are higher in domesticated species than in non-domesticated species. a**, Yearly GMRs are significantly higher in domesticated or farmed species than in wild species (using phylogenetic regression (PGLS) on a total of 68 species). **b**, Using the modelled mutation rate instead (using phylogenetic regression (PGLS) on a total number of 55 species) shows that there is no difference in yearly GMRs between domesticated and non-domesticated animals, suggesting that this difference is mainly driven by the shorter generation time of domesticated species. The box plots represent the median, the interquartile range and the maximum and minimum excluding outliers.

**Fig. 3 | Predictors of interspecific variation in GMRs. a–c**, Significant positive associations are found using phylogenetic regression (PGLS) between the modelled per-generation mutation rates and three life-history traits: species-specific mean generation time (**a**), age at sexual maturity (**b**) and the number of offspring per generation (**c**). In total there are 55 species with modelled per-generation rates, including 32 mammalian and 15 avian species. The box plot in **c** represents the median, the interquartile range and the maximum and minimum excluding outliers. **d**, Species-specific average per-generation mutation rates are negatively associated with the harmonic mean of the effective population size ($N_e$) over the past 1 million years, using phylogenetic regression (PGLS).

including mating system (monogamy versus polygamy), maturation time, body mass, longevity, fecundity and the generation time (Supplementary Table 9). We used the $\mu_{generation\_modelled}$ instead of the $\mu_{generation}$ as the former is less dependent on the age of the parents and is more representative of the rate at generation time for a given species. Although taking into account phylogenetic relatedness, many of these traits are significantly associated with $\mu_{generation\_modelled}$ including the generation time (PGLS: adjusted $r^2 = 0.15$, $P = 0.002$; Fig. 3a), the maturation time (PGLS: adjusted $r^2 = 0.18$, $P = 0.0006$; Fig. 3b) and the number of offspring per generation (PGLS: adjusted $r^2 = 0.10$, $P = 0.013$; Fig. 3c). Species with a higher number of offspring per generation also showed significantly lower $\mu_{generation\_modelled}$ when considering only mammalian species (PGLS: adjusted $r^2 = 0.17$, $P = 0.011$), but this relationship was not significant for birds (PGLS: adjusted $r^2 = -0.066$, $P = 0.720$). Collectively, these traits explain almost 18% of the variation in $\mu_{generation\_modelled}$ (multiple PGLS: adjusted $r^2 = 0.18$, $P = 0.004$). The other life-history traits that we tested, including longevity, mating strategy and body mass, are not significantly associated with $\mu_{generation\_modelled}$ (see Extended Data Fig. 7).

Another key parameter for species evolution is the effective population size ($N_e$), which impacts genetic drift and the efficacy of selection. To investigate the effect of $N_e$ on $\mu_{generation\_modelled}$ and to test the drift barrier hypothesis[3], which predicts the evolution of higher mutation rates in species with small $N_e$, we calculated $N_e$ using the pairwise sequentially Markovian coalescent method based on one randomly selected father per species. To avoid circularity, we estimated $N_e$ based on the substitution rate calculated from the UCE alignment (Supplementary Table 9).

Indeed, if $N_e$ was estimated using the pedigree-based mutation rate, a stronger correlation might arise between $N_e$ and the mutation rate (see Extended Data Fig. 8). We found a significant negative association between $\mu_{generation\_modelled}$ and the harmonic mean $N_e$ per species over the past 30,000–1,000,000 years (PGLS: adjusted $r^2 = 0.08$, $P = 0.020$; Fig. 3d) as would be expected under the drift barrier hypothesis. This relationship is mainly driven by mammals (PGLS: adjusted $r^2 = 0.31$, $P = 0.0006$), a signal that is also observed when using the harmonic average $N_e$ over a smaller timescale (30,000–130,000 years; PGLS: adjusted $r^2 = 0.10$, $P = 0.04$, Extended Data Fig. 8). The most appropriate timeframe used to estimate $N_e$ depends on the evolutionary time necessary for the mutation rate to adapt to changes in $N_e$. However, the pairwise sequentially Markovian coalescent method cannot accurately estimate recent $N_e$. To overcome this limitation, we also estimated $N_e$ as $\pi/4\mu$, with nucleotide diversity ($\pi$) and the substitution rate per site per generation ($\mu$) estimated from the UCE alignments. This results in a similar negative association between $N_e$ and $\mu_{generation\_modelled}$ (linear regression: adjusted $r^2 = 0.83$, $P = 2.2 \times 10^{-16}$; Extended Data Fig. 9), further supporting the drift barrier hypothesis. However, caution should be taken as $N_e$ estimates rely on generation times inferred from contemporary observations, whereas generation times could conceivably have changed over evolutionary timescales. Furthermore, population size depends negatively on the generation time (PGLS $N_e$ in log scale: adjusted $r^2 = 0.20$, $P = 0.0004$). Therefore, a negative association between $N_e$ and $\mu$ could potentially be driven by a large effect of the generation time on per-generation mutation rates.

## High yearly rates in domesticated species

Domestication imposes strong artificial selection, recurrent genetic bottlenecks or both. Our dataset includes 22 domesticated or semi-wild species that have been bred in captivity for many generations. When using the naive method of dividing the per-generation rate by the parental age, these species show significantly higher $\mu_{yearly}$ than the non-domesticated species (PGLS: adjusted $r^2 = 0.13$, $P = 0.0015$; Fig. 4a). The higher mutation rates of domesticated animals are likely due to strong artificial selection for traits such as shorter generation times. Indeed, using $\mu_{yearly\_modelled}$, we found no difference between domesticated and non-domesticated species (PGLS: adjusted $r^2 = 0.037$,

*P* = 0.08; Fig. 4b). Consequently, the higher yearly mutation rate observed in domesticated species is more likely to be explained by the lowering of reproductive age associated with domestication rather than by an inherent change to the mutational process caused by relaxed selection on the mutation rate due to small population sizes and bottlenecks associated with domestication[58,59].

## Conclusions

Here we analysed pedigree-based GMR variation in an unprecedentedly broad phylogenetic context. We showed that there is a consistent male bias in mammals and birds, whereas reptiles and fish exhibited more evenly matched contributions of DNMs between parents. This could be due to contrasting mutagenic processes, such as differences in male and female germline cell division observed in mammals and birds, or differences among species in the proportion of DNMs occurring in primordial germ cell specification versus in the parental germ lines. Our results also support the drift barrier hypothesis, as we found a negative association between the per-generation mutation rate and effective population size. Moreover, our results suggest that an appreciable proportion of the variation in the GMR can be explained by life-history traits, including maturation time and the number of offspring per generation. Our study also highlights the importance of the generation time, as illustrated by the particular case of domesticated animals, in which exceptionally high yearly mutation rate estimates can be explained by artificially induced short generation times. In addition, some of the trio samples in our study were collected from captive animals at zoos or conservation centres. These populations might have different generation times than those in the wild, which could potentially introduce biases into some of our mutation rate estimates. Future studies should focus on wild pedigree samples, which can be accessed from long-term conservation and monitoring programmes[60].

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

## Methods

### Samples

Samples were collected from zoos, zoological museums, research institutes and farms from all over the world. Samples were provided from collaborators for research that was undertaken at the Natural History Museum of Denmark, permit 2020-12-7186-00733 from the Danish Ministry of Environment and Food, and when applicable, CITES Certificate of Scientific Exchange number DK003. Genomic DNA was extracted using DNeasy Blood and Tissue Kits (Qiagen) following the manufacturer's instructions. BGIseq libraries were built in China National GeneBank (CNGB), Shenzhen, China, and whole-genome paired-end sequencing (read length 2 × 100 bp) were performed on the BGISEQ500 platform. We aimed for 60–80× raw sequence coverage per sample. A total of 68 species for which a reference genome was available were retained in the final dataset, representing 151 trios for which whole blood or other tissue material was available for DNA extraction and for which parentage had been genetically determined[61]. Information on the samples is provided in Supplementary Table 1.

### GMR estimation

We applied a similar bioinformatic analysis pipeline to our previous study of rhesus macaques[12]. Raw reads were trimmed with SOAPnuke filter[62]. The mapping was conducted with BWA-MEM version 0.7.15 (ref. [63]). The versions of the reference genomes for each species are provided in Supplementary Table 9. A post-mapping step removed any reads mapping to multiple regions of the genome as well as duplicated reads using Picard MarkDuplicates 2.7.1. We called variants for each individual using HaplotypeCaller in BP-RESOLUTION mode with GATK 4.0.7.0 (ref. [64]). This mode returns a genotype quality and depth for all positions of the genome, not only the polymorphic sites. As recommended by GATK best practices, GenomicsDBImport combined all gVCF files per species into a single file and GenotypeGVCF applied a joint genotyping of all samples within a given species (see Supplementary Table 3 with details of raw sequences coverage, mapping quality, and coverage after mapping and variant calling). Similar filtering methods to those in our previous study were then applied to detect DNMs[12]. Therefore, each trio was filtered as followed:

(1) For site filtering, the variant positions were filtered with the following parameters: QualByDepth (QD) < 2.0, FisherStrand (FS) > 20.0, RMSMappingQuality (MQ) < 40.0, MQRankSum < −2.0, MQRankSum > 4.0, ReadPosRankSum < −3.0, ReadPosRankSum > 3.0 and StrandOddsRatio (SOR) > 3.0 according to previously tested filters[12].

(2) For Mendelian violations, variants that deviated from Mendelian inheritance were selected using GATK SelectVariant and refined with an R script to keep only sites in which both parents were homozygous for the reference allele (HomRef), and the offspring was heterozygous (Het).

(3) For allelic balance filter, in the case of a DNM, approximately 50% of the reads in the offspring should support the alternative alleles. Our allelic balance filter cut-off was 30–70% of the reads supporting the alternative allele, similar to previous studies[12,32,65,66].

(4) For depth filter (DP), only positions with a DP > 0.5 × $m_{depth}$ and DP < 2 × $m_{depth}$ for each individual were kept, with $m_{depth}$ being the average depth of the trio. This strict DP filter minimized the effects of sequencing errors in regions of low sequencing depth and mismapping errors in high-coverage regions.

(5) For genotype quality filter (GQ), to ensure that only high-quality genotypes were retained for the analysis of trios, we removed all sites where one individual of the trio had a GQ < 60 (see Supplementary Fig. 2 for a comparison of various GQ thresholds on a subset of species).

In addition, we called variants with bcftools (version 1.2)[67] in the region of the candidate DNMs and removed the sites that appeared as false-positive calls (that is, at least one parent had the same variant as the offspring or the offspring had no variant). The number of candidates discarded varied among species (Supplementary Table 2). This quality control step produced similar results to a manual check with IGV[68]. Moreover, calling variants with different variant callers has been shown to be an efficient method to reduce false-positive calls[2]. All positions of DNMs are provided in Supplementary Table 6. In addition, we showed that sample type, reference genome quality and mapping quality can affect the results on the number of candidates, the false-positive rate and false-negative rate (FNR), yet, the estimated mutation rates are not affected (Supplementary Figs. 3–5).

To estimate per-generation rates, we divided the number of candidate DNMs, without the apparent false-positive candidates, per the callable genome. A site was considered callable when it passed the same filters as the polymorphic sites, that is, when both parents were HomRef (filter 2) and the three individuals passed the depth filter (filter 4) and the genotype quality threshold (filter 5). On the sites considered callable, we applied a correction for the FNR, that is, the proportion of sites where true DNMs will not be called as such. Two methods have been used in the literature to estimate FNR: one is the simulation of mutations and the other is a correction on the filters that are not accounted for in the callable genome. As in our previous study of GMR[12], we used the latter method, which is more conservative. This corrected for the remaining filters that can only be applied on polymorphic sites, such as the site filters and the allelic balance filter (filter 2). We estimated the proportion of sites that would be filtered away by the site filters on the parameters following a known distribution (FS, MQRankSum and ReadPosRankSum), and the expected sites filtered away by the allelic balance filter as the number of true heterozygote sites (one parent HomRef, the other parent HomAlt and their offspring Het) outside the allelic balance threshold. The mutation rate per site per generation was then estimated per trio as $\mu_{generation} = DNMs/((1 − FNR) × 2 × CG)$. We estimated the 95% binomial confidence interval per species using the binconf() function in R, with the default Wilson score.

To calculate yearly rates ($\mu_{yearly}$), we divided the per-generation rate by the average age of the parents at the time of reproduction weighted by the relative contribution of each parent (inferred with α for 105 trios) or by the generation time (for 46 trios without parental ages). The resulting $\mu_{yearly}$ estimates were averaged per species (for 29 species with multiple trios available). These yearly rates are dependent on the age of reproduction of the parents. Therefore, to calculate a yearly rate at generation time, we first modelled how the mutation rate of a trio was affected by the weighted average of the parental ages (using the paternal fraction estimated for that species as a weight). We then extended the model to fit how each species deviated from the average and used this to correct for differences between the observed reproductive age in our dataset and the expected generation time of a species (see Supplementary Note 1). With this, we estimated a new $\mu_{yearly\_modelled}$ and a $\mu_{generation\_modelled}$ that are more representative of the rate at generation time for each species.

### Phylogenetic analysis

The phylogeny was built based on two sets of UCEs: 5,472 baits for 5,060 UCEs in tetrapods[57] and 2,628 baits for 1,314 UCEs in acanthomorphs[69]. We used the Phyluce software[70] to locate the probes in the reference genomes of our 68 species with 6 additional species contained in our original dataset. We extracted a flanking region of ±1,000 bp for each probe and aligned them with Mafft aligner version 7.470 (ref. [71]). We then created a 75% completion matrix, that is, each alignment contains at least 75% of the taxa (55 species), resulting in 63 alignments from the acanthomorph set and 2,742 probes from the tetrapod set (all alignments are available on Figshare). A phylogenetic tree was built using IQ-TREE version 2.0.3 (ref. [72]), with the appropriate substitution model inferred for each of the 2,805 alignments, a maximum likelihood tree search and 1,000 bootstrap replicates. To validate our tree, we also

estimated a second tree based on a MultiZ alignment to the human genome and obtained similar results (Extended Data Fig. 9). The phylogenetic tree was calibrated to absolute time using the chronos function of the 'ape' package in R, with a smoothing parameter lambda of 0 and a 'relaxed' model[73,74]. Fourteen nodes were calibrated following previously published calibrations. The robustness of the tree was assessed by removing each node independently (see Extended Data Fig. 3).

(1) Actinopterygii/Sarcopterygii: divergence time 416 million years ago (Ma), upper bound 425.4 Ma[75]
(2) The first node in the Actinopterygii group: divergence time 378.2 Ma[76]
(3) Sauropsida (birds and reptiles)/Synapsida (mammals): divergence time 313.4 Ma[77]
(4) Archosauria (birds)/Testudines: divergence time 260 Ma[78]
(5) The basal nodes of the Lepidosauria: divergence time 222.8 Ma[79]
(6) First mammalian node, Eutheria/Metatheria: divergence time 160.7 Ma[75]
(7) Galloanserae/Neoaves: divergence time 66 Ma[77]
(8) Glire/Primates: divergence time 61.7 Ma[77]
(9) Basal gekkotan node: divergence time 54 Ma[80]
(10) Passeriformes/Psittaciformes: divergence time 51.81 Ma[81]
(11) Cynoglossidae/Paralichthyidae: divergence time 50 Ma[76]
(12) *Sus scrofa*/other Cetartiodactyla: divergence time 48.5 Ma[77]
(13) Canidae/Arctoidea: divergence time 37.1 Ma[75]
(14) Hominoidea/Cercopithecoidea: divergence time 23.5 Ma[77]

## Mutational spectrum and sex bias

To analyse the spectrum of mutation, we grouped the trios into higher taxonomic levels, that is, mammals, birds, fishes and reptiles. Thus, the percentages reported are based on the total candidate mutations from each group of species. We explored the genomic context of the mutations from a C or a G base to determine whether they were located in CpG sites (respectively followed by a G or preceded by a C) (see Supplementary Table 4). We phased the DNMs to their parental origin using the read-backed phasing method described previously (GitHub: https://github.com/besenbacher/POOHA)[82]. This method uses the read-pairs containing a DNM and another heterozygous variant to determine the parental origin of the mutation when the heterozygous variant is present in both the offspring and one of the parents. The phasing allowed us to identify parental biases in the contribution of the DNMs by grouping multiple species to increase the number of phased mutations and obtain a minimum of 30 phased mutations per taxon. From this analysis, we omitted the Egyptian roussette (*Rousettus aegyptiacus*), Chinese tree shrew (*Tupaia belangeri*), griffon vulture (*Gyps fulvus*), blue-throated macaw (*Ara glaucogularis*), snowy owl (*Bubo scandiacus*) and Darwin's rhea (*Rhea pennata*), as these could not be grouped with another monophyletic clade. To quantify the effect of parental age, a linear regression between the per-generation mutation rate and the average parental age at the time of reproduction was implemented using the lm function in R. Multiple linear regression was also used to identify whether paternal or maternal age was the strongest predictor of the empirical mutation rate.

## Life-history trait analysis

We tested the effect of various life-history traits (fitted as continuous and discrete variables) on the yearly rate for each species using PGLS analysis in the R package 'caper'[83] (see Supplementary Table 9 for details about each life-history trait).

## Effective population size

We used pairwise sequentially Markovian coalescent (PSMC) models to estimate the effective population size of each species[84]. Fastq sequences were obtained using bam format aligned sequences of one randomly selected father per species and were converted into fastq format using samtools mpileup command and vcf2fq. As recommended,

the minimum depth was set to one-third of the average for the sample and twice the average for the maximum. For mammals, fish and reptiles, the parameters of the PSMC were set to −N25 for the maximum number of iterations of the algorithm, −t15 as the upper limit for the time to the most recent common ancestor, −r5 for the initial θ/ρ value, and finally the atomic intervals −p of '4 + 25 × 2 + 4 + 6'. These parameters were used previously for PSMC analysis of various species, including primates[84,85], cetaceans[86], Felidae[87], fishes[88,89] and turtles[90]. For birds, we used different parameters according to the literature with −N30 −t5 −r5 (ref. [91]). Finally, to simulate the history inferred by PSMC, we parameterized the generation time and the mutation rate inferred from the UCE alignment. We then explored the effect of the harmonic mean $N_e$ over windows of 30,000 years to 1,000,000 years. We also compared $N_e$ estimated obtained with this method with those estimated based on $N_e = \pi/4\mu$. Nucleotide diversity ($\pi$) was calculated using ANGSD[92]. This approach was implemented in three consecutive steps. From the alignment files, a global estimate of the site frequency spectrum was inferred using a maximum likelihood method, then the empirical $\pi$ value was estimated per site, and finally, a sliding window approach was used to estimate $\pi$ for each species. We used a window size of 50 kb and a step size of 10 kb together with an average pairwise estimation of the $\pi$ to obtain global estimates of $\pi$. This analysis was restricted to unrelated individuals from each species, which corresponded to the 2 unrelated parents for 55 species, between 3 and 7 individuals for 10 species, and 3 species were excluded from this analysis as the parents were first-degree relatives.

## Reporting summary

Further information on research design is available in the Nature Portfolio Reporting Summary linked to this article.

## Data availability

Whole-genome sequences of all species except humans are accessible in the National Center for Biotechnology Information under the BioProject ID PRJNA767781. The human sequences are available on request to L.A.B. and should be used only for GMR studies, based on the participant's request. The alignments for the UCE tree are available on Figshare (https://doi.org/10.6084/m9.figshare.19221693.v1). All animal silhouettes are from PhyloPic (http://phylopic.org/), except for the silhouette of *S. scovelli*, which was created by J.S. The silhouette of *P. troglodytes* was created by T. M. Keesey (vectorization) and T. Hisgett (photography), and the one of *S. harrissi* silhouettes was created by S. Werning; both are available under a CC-BY 3.0 license (https://creativecommons.org/licenses/by/3.0/); the other silhouettes are available under a Public Domain Mark 1.0 licence.

## Code availability

The bioinformatics pipeline to analyse the genomes and all other data analyses are available on GitHub (https://github.com/lucieabergeron/vertebrate_rate).

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

**Acknowledgements** The authors thank the following contributors of samples for this study: A. Girard, C. Small, E. Couture, E. Gangloff, A. Bronikowski, F. Yu, H. Fernández, A. Carbajal Brossa, the Barcelona Zoo Biological Bank, J. Partecke, J. Judson, F. Janzen, J. Fjeldså, K. Thorup, K. Glover, L. Koren, M. Nagel, M. Fredholm, M. Liedvogel, T. Aquarium, P. Vullioud, S.-J. Luo, T. Gamble, Y. Yovel, J. Bakker, C. Bombis, T. Charlton, A. Corl, A. Foote, N. Geli, M. Guille, K. L. Hansen, W. Huizinga, M. Hunter, T. Knauf-Witzens, T. Lund Koch, S. Potier, A. Prahl, K. Robertson, C. Scala, M. Schellerup, I. Schnell, K. Vesterdorf, K. Wendelin, K. Worm and W.-z. Wang; G. Pacheco for valuable advice in the laboratory; K. Boomsma for stimulating conceptual discussions and for providing comments on the final version of this manuscript; and GenomeDK at Aarhus University for providing computational resources and support for this study. All sequencing data were generated with MGI-sequencers at the China National Genebank of BGI-Shenzhen. This project was funded by the Strategic Priority Research Programme (XDB13020000) and the International Partnership Programme (no. 152453KYSB20170002) of the Chinese Academy of Sciences, a Villum Investigator grant (no. 25900) from The Villum Foundation, and a Carlsberg Foundation Grant to G.Z. (CF16-0663). L.A.B. was supported by the Carlsberg Foundation and the Villum Foundation. M.H.S. was funded by the Novo Nordisk Foundation (NNF18OC0031004). J.I.H. was funded by the German Research Foundation (DFG) as part of the SFB TRR 212 (NC³) (project nos. 316099922 and 396774617), the priority programme "Antarctic Research with Comparative Investigations in Arctic Ice Areas" SPP 1158 (project no. 424119118) and the sequencing costs in projects scheme (project no. 497640428).

**Author contributions** G.Z., M.H.S., S.B. and L.A.B. conceived this work. M.F.B., B.Q., J.I.H., Z.L., J.S.L. and C.S. provided samples for several species, as well as input into the writing and results interpretation. L.A.B., J.Z., P.L. and M.T.P.G. participated in the extraction, library preparation and sequencing. All analyses were conducted by L.A.B. with input from J.S. for the phylogenetic analysis and S.B. for the mutation rate estimation. L.A.B., G.Z., S.B., M.H.S. and J.I.H. wrote the initial draft of the manuscript with input from all co-authors. G.Z. supervised this project.

**Competing interests** The authors declare no competing interests.

**Additional information**
**Correspondence and requests for materials** should be addressed to Lucie A. Bergeron or Guojie Zhang.

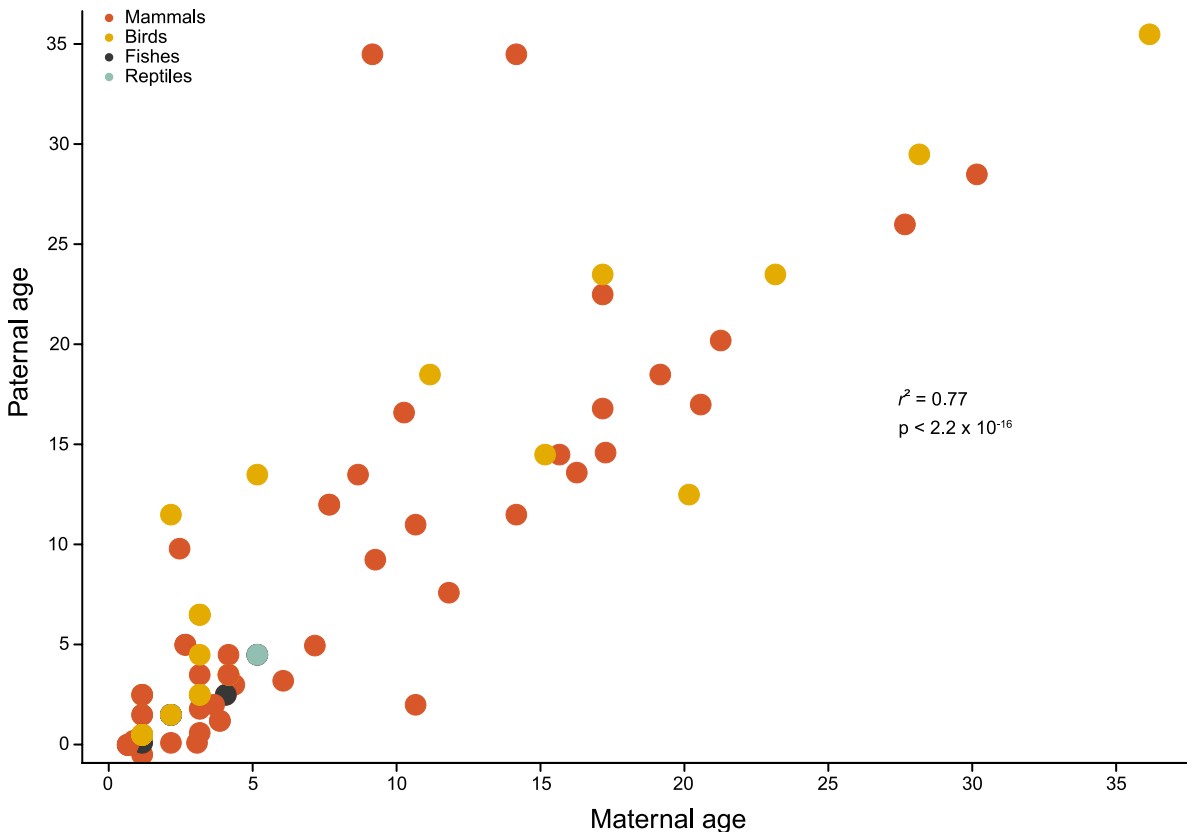

**Extended Data Fig. 1 | Association of parental ages.** Maternal and paternal ages are significantly positively correlated for the 105 trios with known parental age at reproduction (linear regression; adjusted $r^2$ = 0.77, F = 342.3 on 1 and 103 DF, p < $2.2 \times 10^{-16}$).

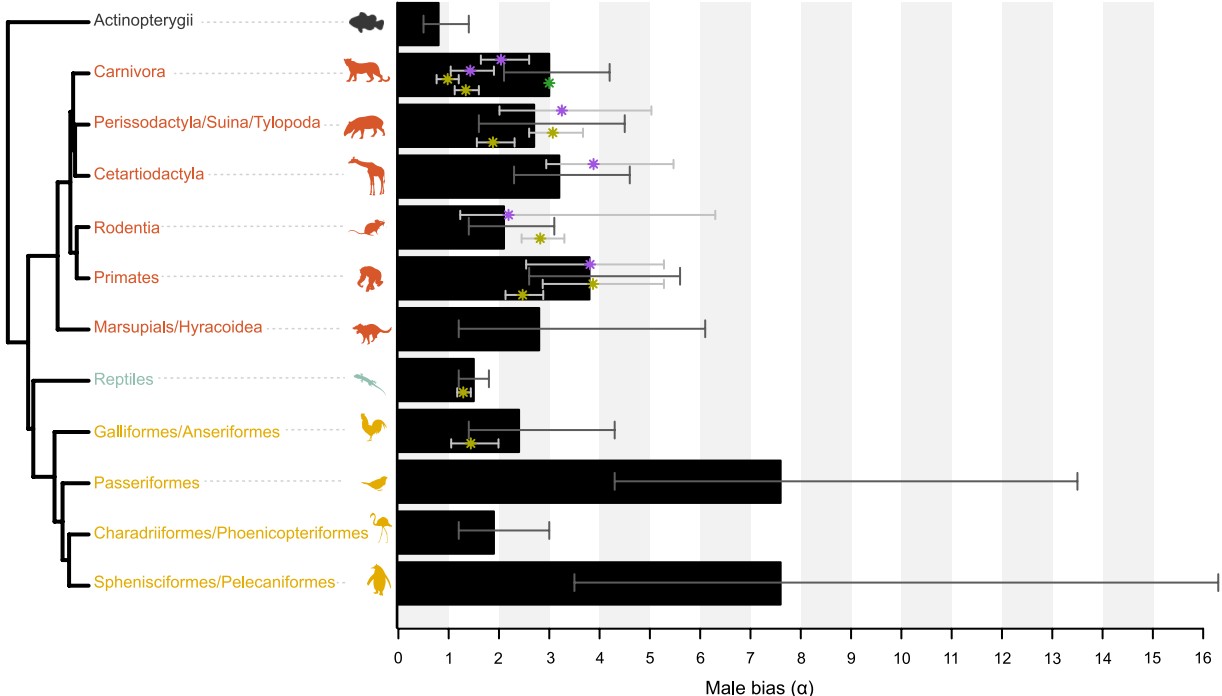

**Extended Data Fig. 2 | Comparison of published male bias estimates (α) using genome alignments and our male bias estimates (modified Fig. 1c of the main text).** The yellow points are α estimates from Wilson Sayres et al.[28], and the purple points are α estimates from Wu et al.[31]. Most of the common species reveal similar estimates with overlapping 95% confidence intervals. However, the estimates of α based on genome alignments are generally lower for dogs and cats than our estimates, yet the pedigree-based estimate of α for cats (Wang et al.[20]; green point) is similar to our estimate. See also Supplementary Table 5. The barplots represent male biases estimated by clustering different species per group (to have a minimum of 30 phased mutations per group) and the 95% confidence intervals were based on the binomial distribution. The silhouette of *Sygnathus* was created by J.S. All other silhouettes are from PhyloPic (http://phylopic.org), except one of the silhouettes of *Sarcophilus harrissi*, which was created by S. Werning, and the silhouette of *Pan troglodytes*, which was created by T. M. Keesey (vectorization) and T. Hisgett (photography); both are available under a CC-BY 3.0 licence (https://creativecommons.org/licenses/by/3.0).

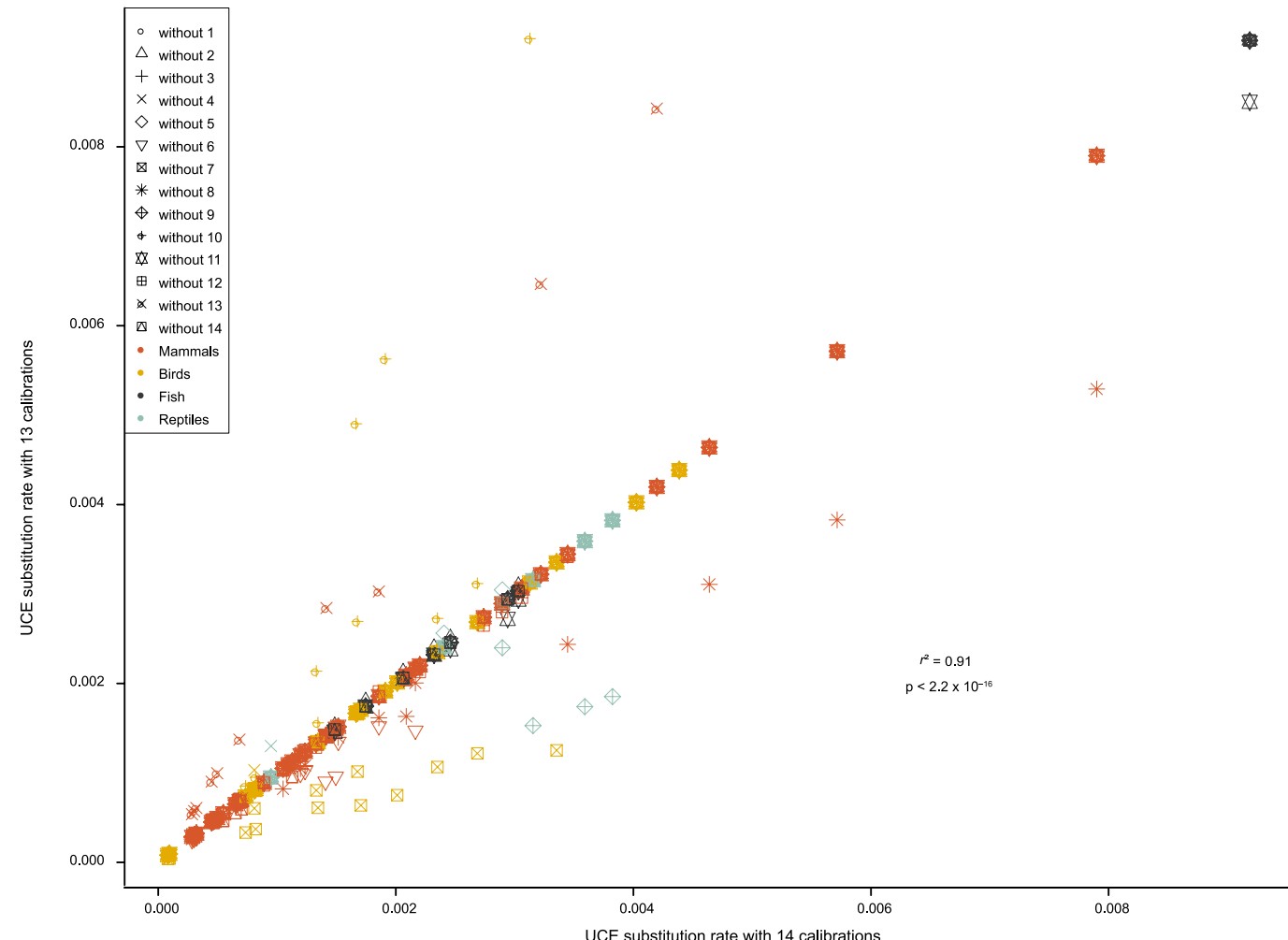

**Extended Data Fig. 3 | Robustness of the calibration.** We compared the estimated substitution rates using the 14 initial calibration points with the inferred substitution rates using only 13 calibration points (with 14 iterations to remove each calibration node one by one). We found a strong relationship between the rates estimated with 14 and 13 calibrations (linear regression adjusted $r^2$ = 0.91, F = 9416 on 1 and 950 DF, p-value: < $2.2 \times 10^{-16}$). However, some of the calibration points had a stronger impact on the estimated substitution rates. For instance, removing the two bird nodes (7 and 10), the gekko node (9), the Canidae/Arctoidea node (13) and the Glires/Primate node (8) altered some of the substitution rate estimates.

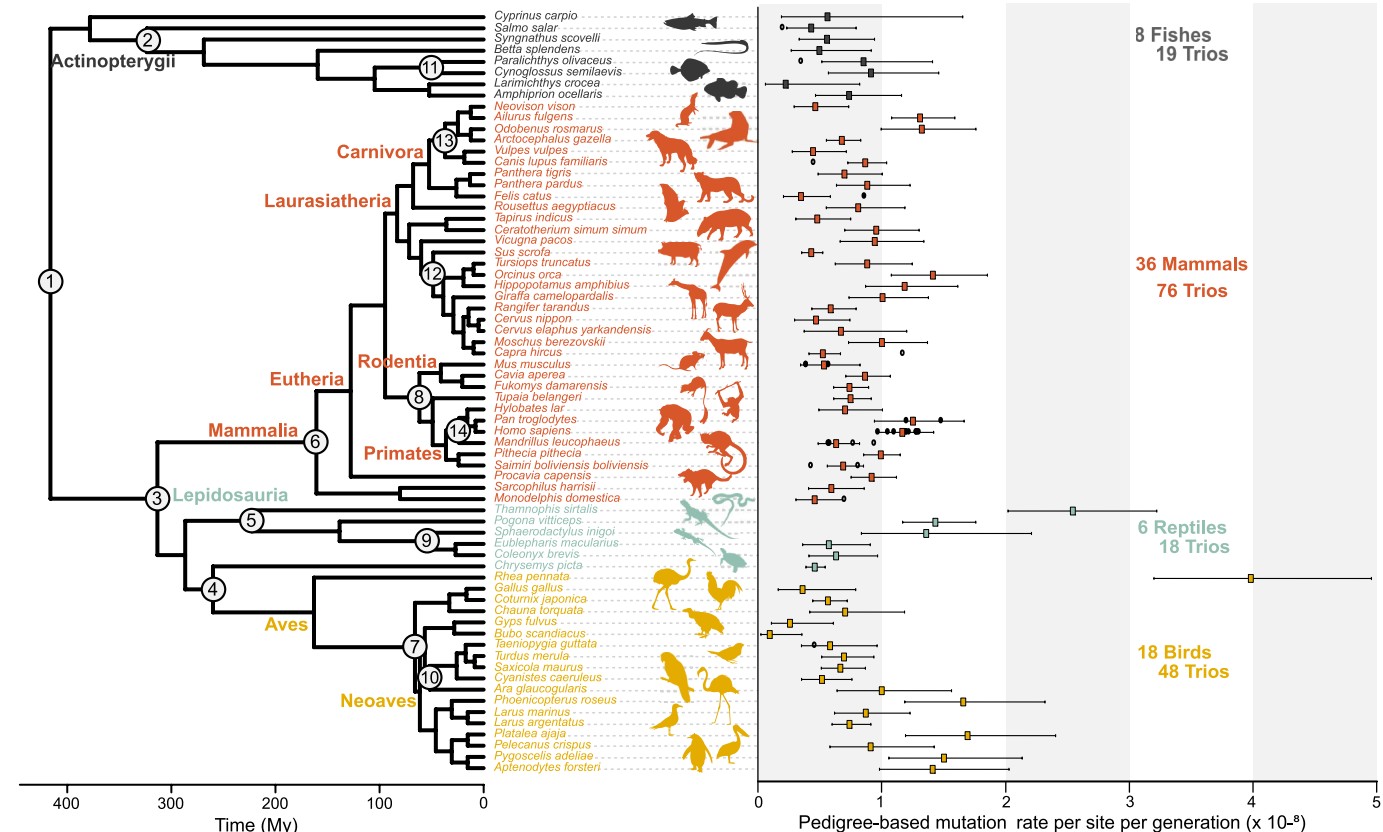

**Extended Data Fig. 4 | Per-generation mutation rates (similar to Fig. 1a) including published data on closely related species.** For each species, the colored squares represent the average per-generation observed rate, along with the 95% confidence intervals based on the binomial distribution, and the black points represent published estimates from similar or closely related species to those included in our dataset. For most of the species, these estimates lie within the 95% confidence intervals of our estimates. Published estimates are from: *Felis catus* (Wang et. al.[20]), *Mus musculus* (Milholland et al.[93], Lindsay et al.[34]), *Pan troglodytes* (Venn et al.[21], Tatsumoto et al.[23], Besenbacher et al.[16]), *Homo sapiens* (Conrad et al.[94], Kong et al.[65], Francioli et al.[32], Rahbari et al.[95], Wong et al.[96], Jónsson et al.[22], Maretty et al.[82], Turner et al.[97], Sasani et al.[98], Kessler et al.[99]). The closely related species are from: close to the *Salmo salar*, *Clupea harengus* (Feng et al.[51]), close to *Paralichthys olivaceus*, the Cichlid (Malinsky et al.[100]), close to *Canis lupus familiaris*, *Canis lupus* (Koch et al.[101]), close to *Capra hircus*, *Bos taurus* (Harland et al.[102]), close to *Mandrillus leucophaeus*, *Papio anubis* (Wu et al.[13]), *Macaca mulatta* (Wang et al.[14], Bergeron et al.[12]), and *Chlorocebus sabaeus* (Pfeifer[103]), close to *Saimiri boliviensis boliviensis*, *Aotus nancymaae* (Thomas et al.[17]), close to *Monodelphis domestica*, *Ornithorhynchus anatinus* (Martin et al.[49]), close to *Taeniopygia guttata*, *Ficedula albicollis* (Smeds et al.[50]). See also Supplementary Table 8. The silhouette of *Sygnathus* was created by J.S. All other silhouettes are from PhyloPic (http://phylopic.org), except one of the silhouettes of *S. harrissi*, which was created by S. Werning, and the silhouette of *P. troglodytes*, which was created by T. M. Keesey (vectorization) and T. Hisgett (photography); both are available under a CC-BY 3.0 licence (https://creativecommons.org/licenses/by/3.0).

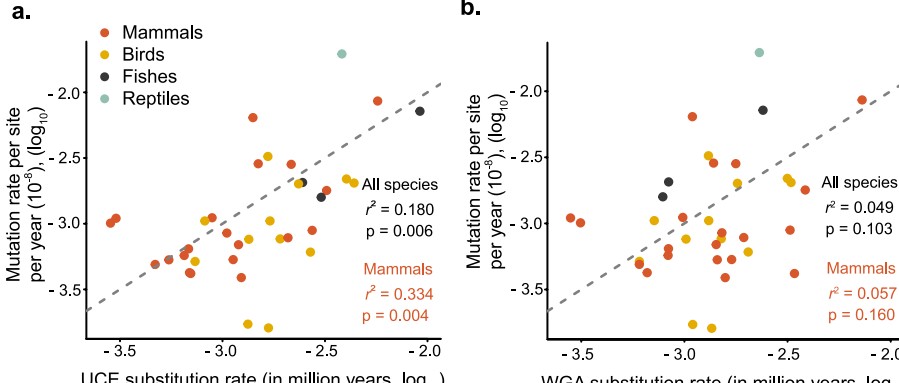

**Extended Data Fig. 5 | Germline mutation rates are associated with long-term substitution rates.** This figure is similar to the main Fig. 2 but uses phylogenetic regression (PGLS) on a log scale. The grey dashed lines indicate equality. **a**. Using a log scale, there is a significant positive correlation between the per-year rates and the rates derived from Ultraconserved elements (UCEs) and their flanking sequences. **b**. However, this correlation is not significant when comparing the per-year rates with the rates derived from the whole genome alignments (WGAs).

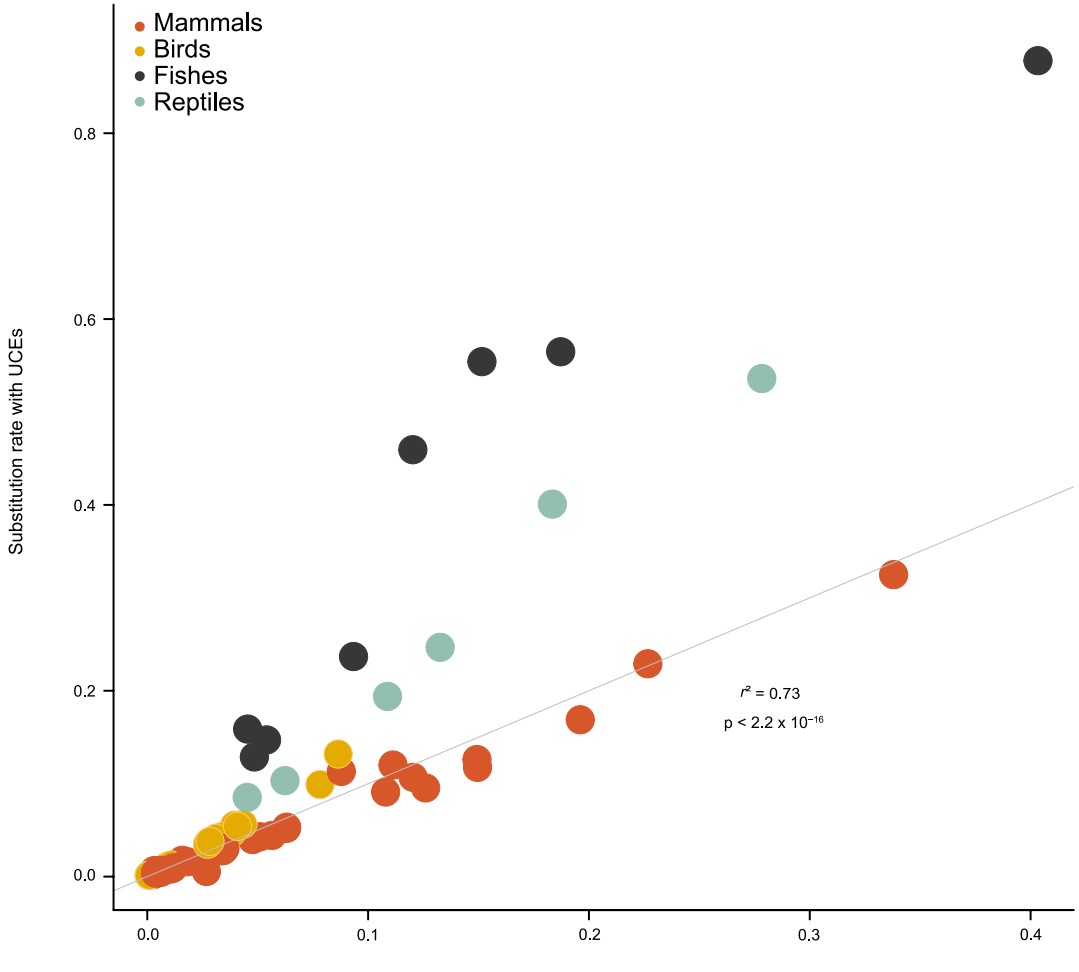

**Extended Data Fig. 6 | Comparison of substitution rates estimated with Ultra Conserved Elements (UCEs) and MultiZ alignments.** The substitution rates estimated with the two methods are highly correlated (linear regression: adjusted $r^2 = 0.73$, F = 179.9 on 1 and 66 DF, p < $2.2 \times 10^{-16}$).

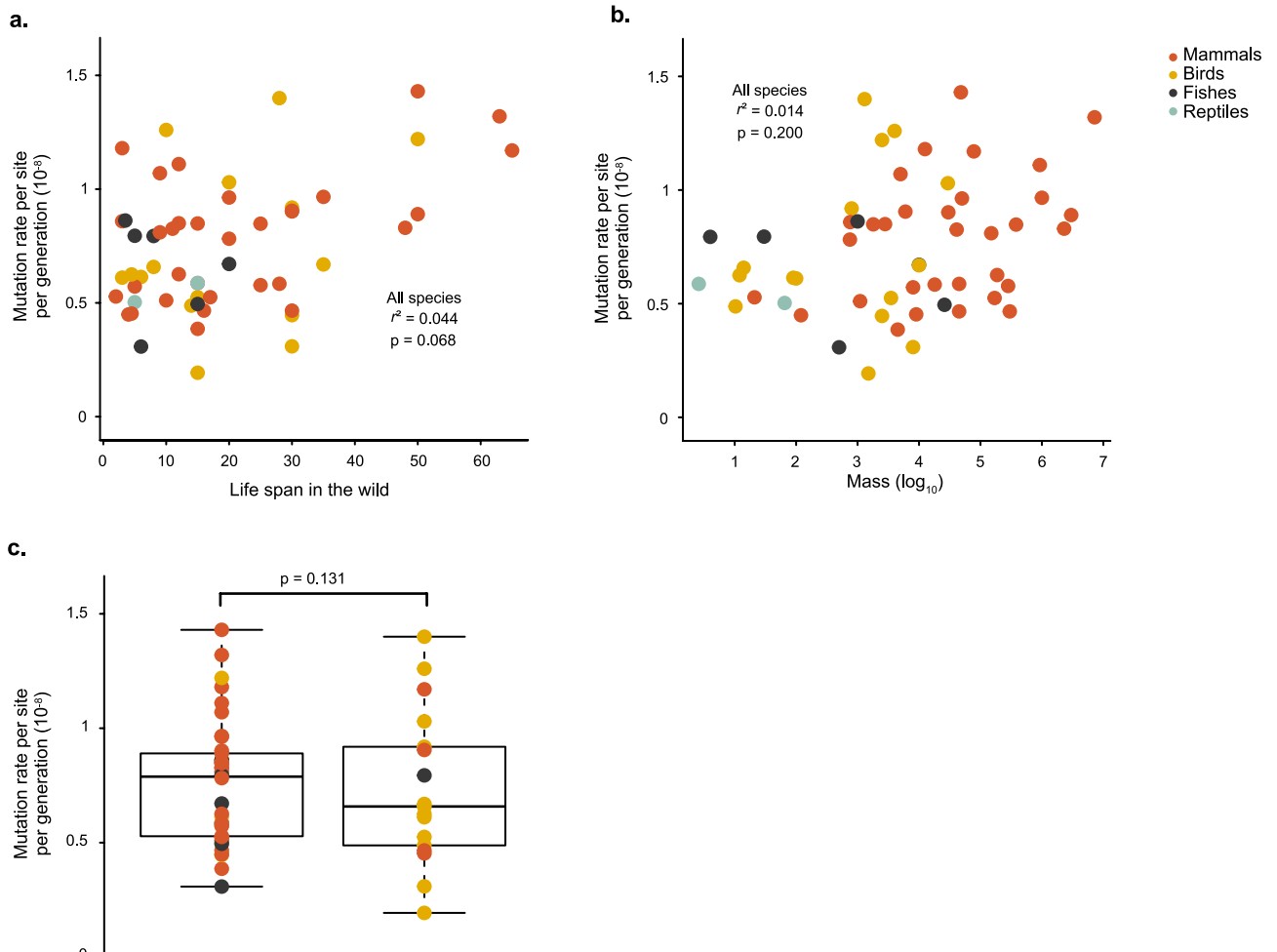

**Extended Data Fig. 7 | Three life-history traits are not significantly associated with the per-generation mutation rate. a.** lifespan in the wild, **b.** body mass and **c.** the mating system (polygamy versus monogamy). The total number of species with modeled per-generation rate was 55 for the phylogenetic regression (PGLS). The boxplots represent the median, the interquartile range, and the maximum and minimum excluding outliers.

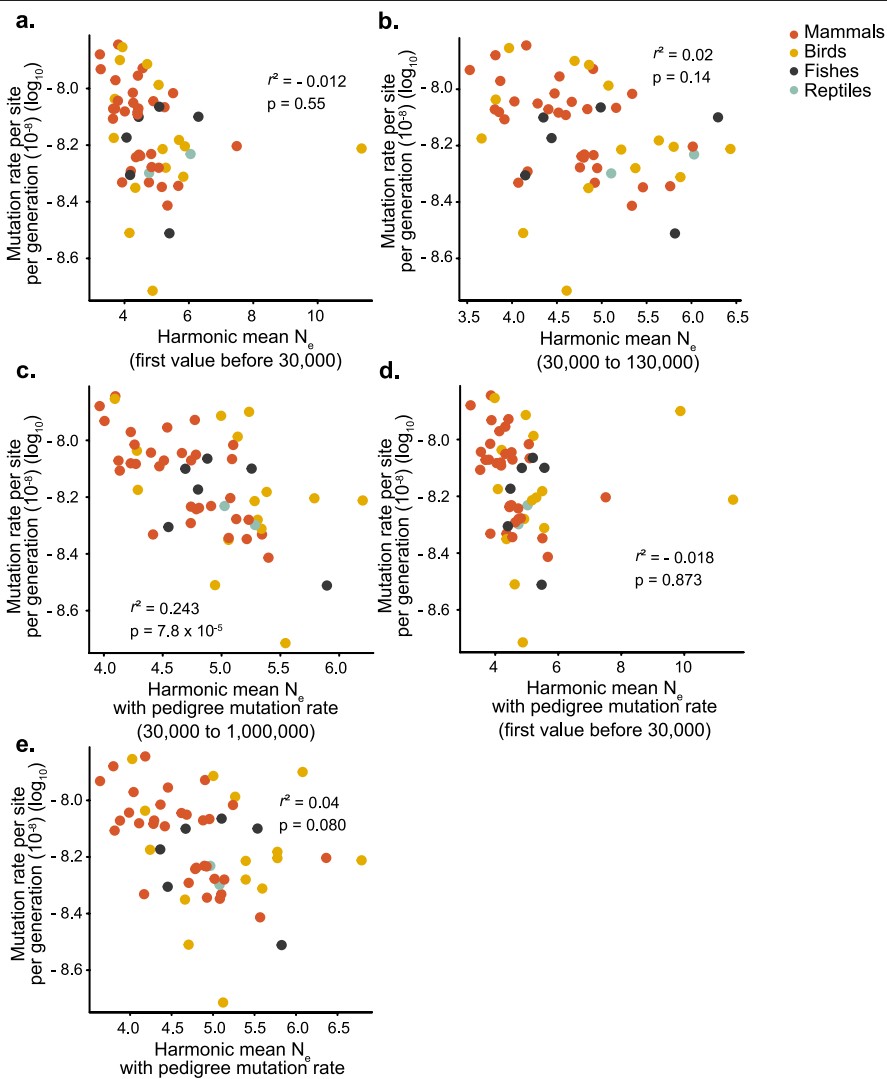

**Extended Data Fig. 8 | The drift barrier hypothesis on different times and different mutation rate parameters used to estimate $N_e$ with phylogenetic regression (PGLS). a**. The correlation between $N_e$ and the mutation rate per generation is not significant when using the most recent value before 30,000 years estimated by PSMC. **b**. The relationship is also not significant when using the harmonic mean over a more recent period of time (30,000 years to 130,000 years ago). However, this relationship is significant for mammals (adjusted $r^2 = 0.104$, $p = 0.04$). We used the harmonic mean over the past million years in the main text, as PSMC is not reliable over recent periods. **c**. When looking at the relationship between the mutation rate and $N_e$, estimated using the pedigree-based mutation rate, we find a stronger signal over the past 1,000,000 years, probably due to the circularity of this analysis. **d**. However, the relationship is still not significant when using the most recent time point or **e**. the average over the past 100,000 years.

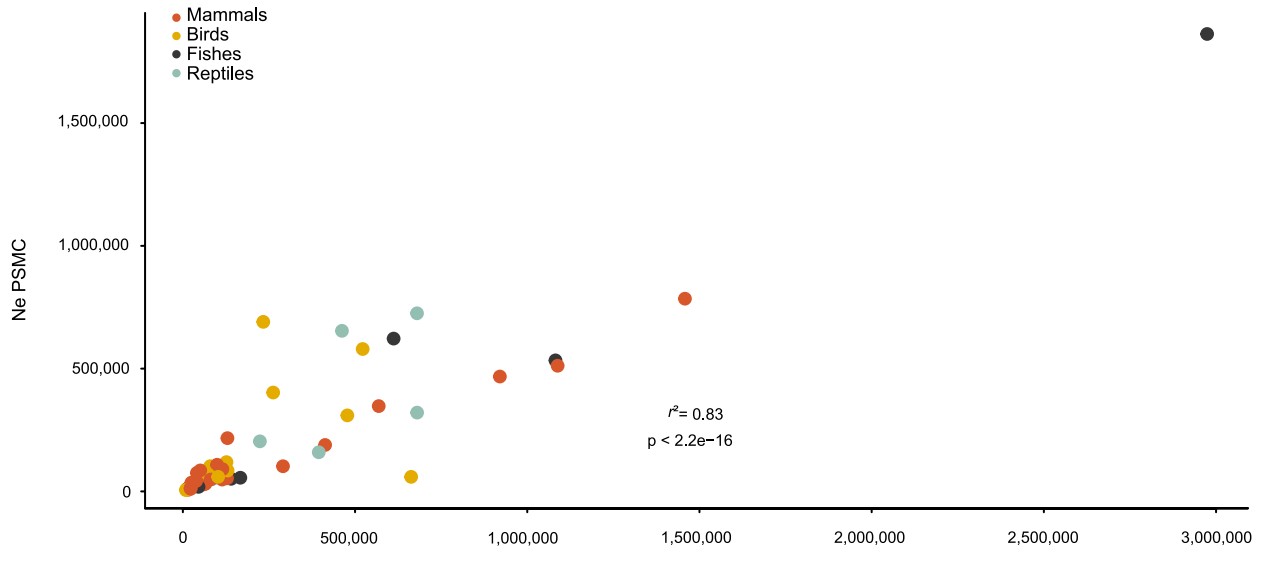

**Extended Data Fig. 9 | Effective population sizes calculated with two different methods (see main text) are significantly correlated.** The harmonic mean of the population size estimated with PSMC from 30,000 to 1,000,000 years ago is significantly correlated with the effective population size estimated from $N_e = \pi/4\mu$ (linear regression: adjusted $r^2 = 0.83$, F = 316.3 on 1 and 63 DF, p < $2.2 \times 10^{-16}$).

# Reporting Summary

## Statistics

For all statistical analyses, confirm that the following items are present in the figure legend, table legend, main text, or Methods section.

| n/a | Confirmed | |
|---|---|---|
| ☐ | ☒ | The exact sample size ($n$) for each experimental group/condition, given as a discrete number and unit of measurement |
| ☐ | ☒ | A statement on whether measurements were taken from distinct samples or whether the same sample was measured repeatedly |
| ☐ | ☒ | The statistical test(s) used AND whether they are one- or two-sided<br>*Only common tests should be described solely by name; describe more complex techniques in the Methods section.* |
| ☐ | ☒ | A description of all covariates tested |
| ☐ | ☒ | A description of any assumptions or corrections, such as tests of normality and adjustment for multiple comparisons |
| ☐ | ☒ | A full description of the statistical parameters including central tendency (e.g. means) or other basic estimates (e.g. regression coefficient) AND variation (e.g. standard deviation) or associated estimates of uncertainty (e.g. confidence intervals) |
| ☐ | ☒ | For null hypothesis testing, the test statistic (e.g. $F$, $t$, $r$) with confidence intervals, effect sizes, degrees of freedom and $P$ value noted<br>*Give P values as exact values whenever suitable.* |
| ☐ | ☒ | For Bayesian analysis, information on the choice of priors and Markov chain Monte Carlo settings |
| ☐ | ☒ | For hierarchical and complex designs, identification of the appropriate level for tests and full reporting of outcomes |
| ☐ | ☒ | Estimates of effect sizes (e.g. Cohen's $d$, Pearson's $r$), indicating how they were calculated |

*Our web collection on statistics for biologists contains articles on many of the points above.*

## Software and code

Policy information about availability of computer code

| | |
|---|---|
| Data collection | No software was used for data collection |
| Data analysis | The software used were: samtools 1.2 and 0.1.18, bcftools 1.2 and 1.9, bwa 0.7.15, Picard MarkDuplicates 2.7.1, R 3.5.1 and 4.0.2, IQTREE 2.0.3, Phyluce, ANGSD 0.920, soapnuke 1.5.6, python 3.7.3, java 1.8.0_222, gatk 4.0.7.0.<br>The entire pipeline used is available at https://github.com/lucieabergeron/vertebrate_rate. |

For manuscripts utilizing custom algorithms or software that are central to the research but not yet described in published literature, software must be made available to editors and reviewers. We strongly encourage code deposition in a community repository (e.g. GitHub). See the Nature Portfolio guidelines for submitting code & software for further information.

## Data

Policy information about availability of data

All manuscripts must include a data availability statement. This statement should provide the following information, where applicable:
- Accession codes, unique identifiers, or web links for publicly available datasets
- A description of any restrictions on data availability
- For clinical datasets or third party data, please ensure that the statement adheres to our policy

Whole-genome sequences of all species except humans are accessible in NCBI (National Center for Biotechnology Information) under the BioProject: PRJNA767781

## Human research participants

Policy information about studies involving human research participants and Sex and Gender in Research.

| | |
|---|---|
| Reporting on sex and gender | Not applicable |
| Population characteristics | Not applicable |
| Recruitment | Not applicable |
| Ethics oversight | Not applicable |

Note that full information on the approval of the study protocol must also be provided in the manuscript.

# Field-specific reporting

Please select the one below that is the best fit for your research. If you are not sure, read the appropriate sections before making your selection.

☐ Life sciences     ☐ Behavioural & social sciences     ☒ Ecological, evolutionary & environmental sciences

For a reference copy of the document with all sections, see nature.com/documents/nr-reporting-summary-flat.pdf

# Ecological, evolutionary & environmental sciences study design

All studies must disclose on these points even when the disclosure is negative.

| | |
|---|---|
| Study description | This study estimates the germline mutation rates for 68 species of vertebrates using whole genome comparison of pedigrees. |
| Research sample | We analyzed 151 trios for 68 species of vetebrates, including birds, reptiles, fishes and mammals. |
| Sampling strategy | We collected trios (mother, father, offspring) for species with reference genome available. Our sample size for each species was limited by the pedigree available. |
| Data collection | Lucie A. Bergeron collected all the data, with the help of collaborators from zoo, research centers and farm. |
| Timing and spatial scale | Data were collected between October 2017 and December 2018. |
| Data exclusions | After data analysis, some of the samples were not as related as stated by the samples providers, therefore, they were excluded. |
| Reproducibility | All sequences and scripts are provided for reproducibility of our results. |
| Randomization | This is not relevant to our study, randomization was not possible as the sample size was small per species. |
| Blinding | Blinding was not relevant to our study as there were no expected data most of the rates were never estimated before from pedigrees. |

Did the study involve field work?     ☐ Yes     ☒ No

# Reporting for specific materials, systems and methods

We require information from authors about some types of materials, experimental systems and methods used in many studies. Here, indicate whether each material, system or method listed is relevant to your study. If you are not sure if a list item applies to your research, read the appropriate section before selecting a response.

## Materials & experimental systems

| n/a | Involved in the study |
|-----|----------------------|
| ☒ | Antibodies |
| ☒ | Eukaryotic cell lines |
| ☒ | Palaeontology and archaeology |
| ☒ | Animals and other organisms |
| ☒ | Clinical data |
| ☒ | Dual use research of concern |

## Methods

| n/a | Involved in the study |
|-----|----------------------|
| ☒ | ChIP-seq |
| ☒ | Flow cytometry |
| ☒ | MRI-based neuroimaging |

