## [Peer Review File · Nature]

Manuscript Title: Evolution of the germline mutation rate across vertebrates

Reviewer Comments & Author Rebuttals

Reviewer Reports on the Initial Version:

Referees' comments:

Referee #1 (Remarks to the Author):

This paper represents a massive advance in our understanding of mutation-rate evolution, at least in vertebrates. Using trio data, the authors have been able to estimate the mutation rate in a wide range of vertebrate species, with replicate family analyses being performed within a substantial fraction of them. I've been waiting to see someone do this for quite a long time. The best estimates of mutation rates come from long-term (100s to 1000s of generations) mutation-accumulation studies in large numbers of replicate lines (often 50 or more). Although this gold-standard approach is unfeasible for vertebrates, the next best option, single-generation parent-offspring trios, becomes feasible owing to the large genomes and the relatively high per-generation mutation rates.

I am strongly enthusiastic about this paper, although I also have a number of issues with it from both an analysis and interpretation perspective. I raise these issues in the interest of enhancing the quality of the overall presentation, and I am confident that the authors will be able to do so. The following comments are in approximate order of appearance in the paper.

Line 56 – This is perhaps a slight misstatement – the lowest rates are generally in unicellular eukaryotes, not prokaryotes.

Line 89 – typo; should be highest.

At the outset -- Some indication of the reliability of the individual data needs to be given (this is never displayed in the main paper). From the first supplementary figure and the 2nd supplementary table, it is clear that the number of observed mutations per almost all trios is in the range of just 1 to 50, with about half apparently below 20. There are standard ways to get confidence limits based on an assumed Poisson distribution of numbers of hits, so this could be done for each estimate. Knowing this would be helpful in interpreting a number of the patterns the authors point out – some reviewers might object that these are in some cases not particularly strong, but if they hold in the face of error-prone data, this objection will be somewhat dissipated.

Also of considerable interest here is the variation among trios within the same species. In principle, the number of observed mutations should be close to Poisson (assuming similar numbers of sites analyzed). However, this need not be true in a comparison among trios in the same species. As for any trait, there must be standing genetic variation for the mutation rate (and a few studies in invertebrates have shown up to a 3 or 4 fold range of variation; and I'm surprised that in a study of

this size an occasional strong mutator wasn't found), so it would be nice to know the degree to which the observed variation exceeds the null expectation. If the level of variation is too low, I would be worried that the analyses have been overly aggressive in trying to confine their final results to some preconceived range of reality. The bottom line here is that the authors have a real opportunity here to clear up a common problem in the literature – it is common for authors of mutation-rate papers to make a big deal about two- or three-fold differences in mutation rates among species, and to try and concoct ecology (or otherwise) explanations; the natural range of variation within species is at this level, this has significant implications for the validity of these types of arguments.

Age effects:

To what extent are the per-generation results a function of the age of the parents; ideally, one would want these ages to be roughly the mean age at reproduction in the wild. This becomes particularly relevant as the authors try to normalize everything to per-year rates. Ideally, what we want to know is the average mutation rate for parents at the average age of reproduction in the wild. A number of the ages of individuals (sup table 3) seem overly high or overly low for this target age, so this warrants a bit more attention. Even if we do not always know the average situation in nature, this is where the mutation rate evolved.

Scaling by absolute time does not necessarily resolve this issue, as the scaling factor could differ among species. Indeed, there is a more general need for caution here because. To my knowledge, there is no compelling evidence that the mutation rate scales linearly with absolute time, which is the scaling the authors resorted to. Supposing the extreme, where for example, mutations in the germline might set down in the initial oocytes/spermatocytes with quiescence thereafter, or vice versa, initial quiescence followed by a progressive increase after maturity. In either case, the sort of treatment employed in this study could be quite misleading, for without accounting for the nonlinearity, the per-year estimates will be potentially strongly biased by the age of the parents. In the former case, lower values would tend to arise in species with longer generation lengths, purely as a consequence of dividing by a larger number; the data in Figure 1 vaguely look like this. The authors seem to acknowledge this issue, but do not account for it (lines 190-208).

Of course, the information to do so is largely lacking, but a more forceful caveat is probably warranted, and as noted below I'm not entirely convinced that per-year rates are what is relevant from an evolutionary perspective. Natural selection operates on the per-generation rate, which is the natural time scale for population-genetic processes.

Line 110 – what is meant by a “weak base”?

Plots 3f and 3g – Presumably the scale in the plot is based on natural logs, but things will be more easily interpretable if log₁₀ is used.

Issues regarding the drift-barrier hypothesis:

Here the authors have an enormous opportunity, as one of the leading hypotheses for the

evolutionary diversification of the mutation rate across the Tree of Life is the idea that the efficiency of selection for mutation-rate reduction is compromised in species with low N_e .

The authors use the PSMC method to estimate harmonic N_e in their study species, but this needs some further discussion, as the PSMC often results in very similar demographic patterns in diverse data sets, suggesting that the method can lead to biased results owing to the many internal assumptions being made. There are alternatives that could be used, and perhaps should be explored if for no other reason than to demonstrate the robustness of the results.

The simplest approach would be to simply start with silent-site heterozygosity, set this equal to $4 * N_e * u$, and then factor out the mutation rate (u). This does assume that average heterozygosity within a single individual is representative of the species-wide diversity, but the PSMC makes the same assumption about the utility of single individuals.

With either approach, the mutation rate itself must be used to get an estimate of N_e (a point that is not made by the authors). This introduces a couple of key issues. 1) The measure on the x axis (N_e) is subject to estimation error, which means that the true slope of the regression must be stronger than what the authors show (this effect might be factored out by subtracting the error variance of estimate of N_e from the total variance of the N_e used in the denominator of the regression coefficient. 2) Because estimates of N_e are a function of the estimate of u used, there can be an intrinsic negative correlation between the two estimates (i.e., if u is overestimated, the derived N_e estimate will be underestimated). This second problem is not too severe if the errors in estimation of u are sufficiently small (it wouldn't be a problem at all if u were measured without error....) In any event, these two sources of potential error operate in opposite directions. Moreover, the first source of error applies to all of the regressions that the authors apply, including those based on life-history traits (which are certainly not error-proof).

Still another kind of issue here concerns the interpretation of the various regressions on life-history parameters. As the authors note, such parameters are often inter-correlated, making strict interpretations difficult. However, life-history traits also may often be correlated with N_e , leaving open the question of causality vs. indirect correlation in the evolution of the mutation rate. For example, the observed increase in the per-generation rate with generation length could be a consequence of the drift barrier, with longer-lived species tending to be larger in size and with lower N_e .

Correlations between mutation rates and heterozygosities:

From what is describe in the text, these appear to be based on genome-wide variation, whereas, as noted above, a comparison against silent-site diversity would be more meaningful. Indeed, it is difficult to ascertain what the authors are trying to do here. Total variation need not strictly be correlated with mutation rates (u). On the other hand, under the assumption of neutrality, silent-site variation (at mutation-drift equilibrium) as an expected value of $4 * N_e * u$.

However, this part of the manuscript is quite confusing, as at line 228, it is argued that π is

proportional to N_e and u , leaving me to wonder whether the authors really are reporting silent-site variation. To make things even more confusing, it is stated that N_e “depends on the mutation and generation time.” I am unaware of the basis of this statement, but it is not the reference given, and to my knowledge is not something that is commonly believed. Arguably, the direction of causality is reversed here.

Use of per-year estimates:

Above, I outlined some of the problems in making these estimate (and I wish to emphasize that the power of the paper is in the observed per-generation rates), but there are additional interpretative issues when arguments towards the end of the paper rely on the former.

Is the ~100-fold range of variation in the yearly mutation rate really surprising? As noted above, selection operates on the per-generation rate, which is the natural timescale of evolution, and so depending on the life-history strategy, the per-year rates will just be adjusted accordingly. More generally, all evolutionary processes are most reasonably viewed as per-generation processes, so I am not convinced these data represent variation in “the capacity of lineages to evolve novel phenotypes, consistent with GMR-induced phenotypic evolvability across lineages and therefore plays a major role in long-term macroevolutionary processes.” On a per-generation basis, there is only a few-fold range of variation of the mutation rate among diverse vertebrate species, and it should be kept in mind that mechanisms of evolution depend on the joint distribution of mutation rates and N_e , so this quote strikes me as a bit of an exaggeration that I don’t think is an essential selling point for this paper.

The following conclusion also does not ring correct to me: “The strong correlation between fecundity and GMR is consistent with an evolutionary trade-off between the number and quality of offspring, such that GMR is a general indicator of average offspring quality optimized by natural selection in each species’ gene pool.” If there is any tradeoff between offspring number and quality, it is only in a physiological sense (more offspring implying less provisioning), not in a genetic sense that has anything to do with mutation or genetic variation. So either more explanation here is desirable, or I would drop this as well.

Similarly, “Because domesticated animals have been consistently artificially selected for high fecundity rates, their exceptionally high yearly mutation rates can be considered as a direct test of our conclusions based on wild animals.” It is unclear that the animals the authors are referring to are really under selection for higher growth rate (guinea pigs, cats, and dogs?), and in any event, domesticated species have very low historical effective population sizes, so the pattern observed could be a simple consequence of the drift barrier. The authors do mention this, but combined with the more general relationship with N_e in nondomesticated species, this would seem to be a more unifying explanation.

Referee #2 (Remarks to the Author):

I like this paper and I think it will make a valuable contribution to the literature – studies of mutation rate variation have tended to either focus on single species with relatively large amounts of data (through parent-offspring analysis) or across many species with relatively small amounts of data (using homologous DNA sequences), but this study combines the strengths of both approaches. However, the study currently has some weaknesses that could certainly be overcome with a little additional analysis, and then the conclusions would be much more robust.

The study relies on the phylogenetic analysis, which is dependent on strong assumptions about calibrations and other aspects of the evolutionary process, yet the phylogenetic analysis is described in only a few sentences. Many of the assumptions underlying this analysis could be questioned (for example the calibration Primate/Glires split at 61). The point here is not to say whether the assumptions are wrong or right, but that we rarely have confidence that we know for certain that particular models, calibration dates etc are appropriate, and yet the results might be quite strongly dependent on particular dates. This could be circumvented by showing the results are robust to a wide range of assumptions (ie not just running on a sample of trees from one analysis with one model but on many independent runs with different assumptions), or by using an approach that does not require dating the tree or resolving the lower nodes of the tree – for example a sister pairs approach (This would also help to provide a counterpoint to PGLS, which relies on strong assumptions about the model of evolution underlying the traits - which are not discussed or explored in this paper). Simply saying that other people have used these calibrations is not sufficient defence of their veracity – actually many different calibrations have been used in the literature, and there are consequently a very wide range of date estimates for some of these nodes. The authors could rerun their analysis on different trees or on a wider range of phylogenetic hypotheses to prove that the results are robust.

I was confused by the macroevolutionary analysis that claimed to link the rates from the parent-offspring estimates to lineage level. This is done by estimating the average genome-wide heterozygosity for each species, but I am not clear on the logic of why this tests the continuity of microevolution and macroevolution. This needs much clearer explanation. Why not also estimate the mutation rate from the phylogeny by inferring the synonymous mutation rate? That way, the study could compare the rates from the parent-offspring estimates to the lineage level averages – this has been a topic much discussed and debated so this study could make a useful contribution.

The higher mutation rates in domesticated animals is interesting and reasonable, but actually this result is not particularly strong. The distribution of values between the domesticated and non-domesticated are not distinctly different, and there are many non-domesticates with higher values than the domesticates (fig 4), so its simply not true to say “the exceptionally high mutation rates

observed in domesticated animals". The average is higher but this average is meaningless without phylogenetic correction, and actually there are many non-domesticates with higher rates than most of the domesticates. A better way to analyse this would be to compare each of the domesticated lineages to their non-domesticated relatives, not to lump them all together. I was surprised that there no reference at all to any published studies on this topic, despite decades worth of research on this topic, and also no reference to alternative explanations such as relaxed selection or selection for high degree of generation of genetic novelty (as proposed, I think, in the classic paper by Burt and Bell on chiasma frequency in domesticated mammals). This needs to be more thoroughly investigated and argued more robustly if it is to form a key finding of this study – at the moment it seems to be essentially descriptive, anecdotal and not considered in the context of a large body of existing evidence on this topic.

In fact, the study does seem quite light on references to previous studies of variation in mutation rate between species, which given the amount of research done on life history patterns in mutation rates seems rather odd.

Final point: please make the alignments freely and easily available when published. This is critical not just for people to be able to repeat this analysis for themselves, but also provides a useful resource for future studies.

Summary of the key results: OK

Originality and significance: if not novel, please include reference: the results are in line with previous studies (e.g. Welch et al doi: 10.1186/1471-2148-8-53.) but here with a bigger and better dataset, so the originality is in the data used rather than the findings.

Data & methodology: validity of approach, quality of data, quality of presentation: OK but not enough detail given

Appropriate use of statistics and treatment of uncertainties: Little formal treatment of uncertainty.

Conclusions: robustness, validity, reliability: Not possible to evaluate given the information in the paper.

Suggested improvements: experiments, data for possible revision; suggest applying to range of phylogenetic hypotheses or using an approach robust to phylogenetic uncertainty e.g. sister pairs.

References: appropriate credit to previous work? No.

Clarity and context: lucidity of abstract/summary, appropriateness of abstract, introduction and conclusions: yes, with the exception of the microevolution-to-macroevoolution argument which is not clearly explained.

Referee #3 (Remarks to the Author):

The germline mutation rate (GMR) is a key parameter to measure the pace of evolution. However, it is difficult to estimate GMR accurately or even to establish whether (or rather, how) it varies over evolutionary times and is influenced in response to molecular mechanism (i.e. mutagenesis and/or DNA repair), environmental factors or life-history traits (i.e. metabolic rate, generation time, reproduction mechanism, fecundity, population size...). While there are many different ways to estimate GMR, in this study, the authors have used direct whole genome sequencing (WGS) of 151 vertebrate parent-offspring trios from 68 species (mammals, fishes, birds and reptiles) to directly quantify species-specific transgenerational GMR and mutational patterns.

They report ~40-fold variation of GMR across species, which corresponds to a large (~ 125-fold) variation in yearly mutation rate ('u-yearly'), and a variable level of parent-of-origin effects among species/taxa. Their data point to fecundity (and selection via domestication) has a major life-history trait that may explain the variation in mutation rate among species. Their data are consistent with the concept that GMR can account for the observed genetic diversity (within individual genomes, their species or across species), confirming the idea that genetic changes (micro-evolution) and genomic (selective) macro-evolution occur as part of a continuum of effects.

I read this well-written manuscript with great interest and enthusiasm. In my opinion, this work represents a tour-de-force in sample collection, data acquisition and analysis. The authors have generated a very large, high-quality dataset of genomic sequences of family trios following a systematic and consistent approach. This represents a unique and invaluable resource for the research community. It gives them the ability to compare directly de novo mutation (DNM) load across species and taxa, and explore patterns of mutations to test hypotheses (male/female ratio, mutation signatures, micro- vs macro evolution patterns...).

However, the drawback of presenting such an extensive dataset is the difficulty in choosing what to analysis or how to interpret the data given the many biological/environmental/reproductive strategies. While, overall, the authors have done a nice job in singling-out some features and general trends, the discussion (in comparison to the wealth of data) felt relatively superficial. I am not sure if there is a way to remedy this and overall the discussion is well balanced in highlighting the limitations of the work presented.

My comments below are fairly minor and mainly focus on the presentation of the discussion:

While I understand the importance of giving an overview of the data, I feel that the focus on the very wide variation of GMR across species (page 3 paragraph, entitled "More than two orders of magnitude variation in the GMR of vertebrates") may be quite misleading. There are clearly limitations about the approach that will unavoidably result in uncertainty in the GMR estimates for the trio and/or the extent to which it reflects the GMR of its cognate species:

1. As mentioned by the authors, DNMs are rare events. Some of the GMRs (u-generation) are based on a very limited number of DNMs per trio (for example for the snowy owl only 2 DNMs were called,

and 5 DNMs for the Griffon vulture – of note, these are the 2 species that are singled-out in the text for the lowest u-yearly rate). The ability to call DNMs depends on the quality of the reference sequence and the proportion of the ‘callable’ genome’ which may be problematic for the less-well characterized species (for example down to 17% for the Common carp) and/or for the species whose genomes contain large segmental duplications/genome duplication events.

While u-generation is described as varying by 40-fold across species, the general trend described on page 3 of GMRs being highest in birds and lowest in fish with mammals and reptiles in between (birds<mammals<reptiles<fish) only differs by 7.7-fold, with largely overlapping CIs. To my mind, this highlights the remarkable conservation of GMRs across species/taxa with very different life-traits and mechanisms of reproduction rather than their differences?

- Moreover, to calculate the yearly GMR (u-yearly), the authors used the combined parental ages of the specific trio (when available). Using ‘u-yearly’ allows them to make the claim that GMR varies by 125-fold across species. While it may appear logical to use the average parental age of the trio to derive u-yearly, this introduces more uncertainty on the GMR estimates as parental age of the trio may not reflect the generation time or be indicative of the average reproduction age/sexual maturity of the species in the wild. (i.e. How relevant are measurements from trios maintained ‘artificially’ in a zoo in respect to wild populations – for example, the emperor penguins pair are 36 yrs old, but would only live a max of 15-20 yrs in the wild). Moreover, some of the longer-lived species will have relatively long pre-pubertal period (which I assume have not been subtracted – for example, the emperor penguins reach sexual maturity ~5-6 yrs of age), likely further biasing u-yearly estimates. Importantly if u-yearly allows a direct comparison for species with very different lifespan, the key impact of GMR is not the yearly rate but the rate across generations. Hence generation time is a key parameter in the interpretation of GMR variability.

2. The authors also assessed the ratio of DNMs that could be phased to their parental chromosomes, to derive the male-to-female ratio (α). Again in this case, they analyse α by taxa and convincingly discuss some general trends in the paragraph entitled “Variable male-driven evolution across vertebrates” on page 6 (and Sup Table S3) showing that overall their data are consistent with the idea that differences in gonadal stem cells biology between males and females drive GMR sex bias across taxa.

As mentioned by the authors at the end of page 7, “parents always carry a certain minimum number of mutations in their gametes regardless of their age”. An important source by which these ‘age- and sex-independent’ mutations can arise in parental gametes is gonadal mosaicism. With this in mind, I note that on page 3 the authors mentioned that: “we generated high-depth genome sequences (average coverage > 67X) for 323 individuals, representing 151 trios of 68 vertebrate species”. Although I can see that a parent pair can be part of several trios, I wonder how many ‘families’ actually had more than one offspring – i.e. some families will be quartets/quintets. Families with multiple offspring offer the possibility to quantify the impact of mosaicism on GMR, which maybe an important contributor to GMR. In a BioRxiv study from 2017, Harland et al (<https://doi.org/10.1101/079863>) showed that in cattle, a significant proportion (up to 30-50%) of apparently new mutations actually occurred in early cleavage embryos resulting in frequent mosaicism and a high rate of sharing of multiple mutations between siblings. Whether this phenomenon is common to other species (or other domesticated animals in particular) would be of great interest and could potentially alter the GMR.

3. It is likely that the dataset is not large enough, but it may be of interest to quantify GMR within the coding genome of the different species - some species exhibit very 'compact' non-coding genomes and may not be able to accommodate higher GMR for this reason.

4. 'Domestication' and selection for breeds can occur for different reasons – for example could the higher mutation rate of domesticated species be driven through changes in effective population size of a 'pure breed'?

5. If the authors are not aware of the Biorxiv manuscript from Cagan et al, (doi: <https://doi.org/10.1101/2021.08.19.456982>) entitled "Somatic mutation rates scale with lifespan across mammals" they may want to consult it. In this work, Cagan et al show that despite widely different life histories, including ~30-fold variation in lifespan and ~40,000-fold variation in body mass, the somatic mutation burden at the end of lifespan varied only by a factor of ~3, suggesting that, surprisingly perhaps, somatic mutation rates are evolutionarily constrained across mammals. Again this illustrates my first point, the GMRs are remarkably conserved given all the factors that influence mutagenesis and result in DNA errors.

Referee #4 (Remarks to the Author):

The authors have generated high coverage whole genome sequences for 151 trios from 68 species of vertebrates including 36 mammals, 18 birds, 6 reptiles, and 8 fishes. All the data was uniformly analyzed, and dataset and pipelines are expected to be released upon publication. This is the most comprehensive study of pedigree mutation rates in vertebrates and is an important contribution for understanding the determinants of mutation rate. Moreover, it will be an important resource for studies of molecular evolution and molecular dating. The analysis performed is interesting, but I have some technical concerns about the results that I detail below.

1. Large variation in rates across individuals within a species, within a clade and across clades: Figure 1 and Table S2 show that there is large variation in per generation and yearly mutation rates across individuals within a species and also within clades. This variation seems larger than expected based on life history traits alone. For instance, across 8 trios of quails (same parental ages) the per generation mutation rate varies by 4x, across 6 trios of guinea pigs by 3x and even across two trios of humans by 40% ($0.99-1.40 \times 10^{-8}$), despite very similar parental ages. Moreover, there is substantial variation within clades as much as across clades. It is not clear how much of this variation is random variation (due to small sample sizes) and systematic variation (due to pipelines – see #2). I think the authors need to try to quantify these effects before any inferences can be gained from this analysis. They should provide uncertainty for all estimated parameters– mutation rate per generation, yearly rates, mutation spectrum etc. and also address #2 for characterizing the systematic biases that can be introduced due to the pipeline.

2. DNM identification in trios

Given that DNMs are very rare and there only ~70-100 mutations (out of 6 billion bp) in humans per generation, it remains challenging to reliably infer DNMs given sequencing errors and somatic

mutations. To perform a reliable comparison across species, the authors apply a uniform pipeline across all the trios in their study. However, given that the source of DNA and reference genome quality differs vastly this may not be justified. Specifically, I am concerned about the following:

a. Higher somatic mutations in DNA extracted from tissues: The authors have used a range of samples to obtain DNA for sequencing including whole blood, tissues, DNA (possibly extracted many years ago). Each of these would have a different % of somatic mutations. It would be useful if the authors can include specific details about the tissues used in the study, investigate how the source of DNA impacts the estimated number of DNMs in the samples and adjust the pipeline accordingly so the inferred DNMs are in fact germline DNMs and not somatic mutations in the offspring.

b. Estimation of false positive rate: To infer the false positive rate, the authors applied another variant caller (bcftools) in the region of the candidate DNMs and removed sites which did not show similar genotypes as GATK. While using multiple variant callers is a useful way to reduce sequencing errors, applying it only to candidate DNMs is likely to bias the mutation rate downward. Importantly, one is more likely to miss a het in the parents and hence its critical to apply the same approach to all regions in the genome and not just the numerator. An important limitation of this approach for inferring FPR is that it is not an independent validation – since it is unlikely to identify errors/ biases in the earlier steps of the pipeline. It would be useful if the authors can show the validity of this approach by validating a subset of mutations by another method – either sanger sequencing or using publicly available multi-generation pedigrees.

c. Estimation of false negative rates: To estimate the false negative rates, the authors apply filters to polymorphic sites and estimate the proportion of sites where true polymorphic site will be removed. This approach by design assumes that polymorphic sites and DNMs are the same, which is not necessarily true as variant calling approaches like GATK use machine learning algorithms that give higher confidence to heterozygous sites previously seen in the sample and hence lower confidence to rare variants and DNMs. Thus, a more reliable approach to inferring the FNR would be by simulations using methods such as bamsurgeon (<https://github.com/adamewing/bamsurgeon>) where a DNM can be artificially added to the reads and then the entire pipeline is applied to recover the DNMs. This approach should be applied independently for each species – while a lot of work – given the differences in genome quality and resources, estimates in one species may not apply to others.

d. Impact of filter cutoffs: The authors use specific cutoffs and thresholds for filters. Some of these filters have previously been shown to have a major impact on the estimated DNM rates, not just count (Besenbacher et al. 2019). Thus it would be useful if the authors can explore the impact of different threshold (for instance for depth, GQ etc.) and present results for other thresholds in the supplement.

e. Association of genome quality and DNM rate: Given the genome quality across vertebrates varies vastly, it would be useful to see some results to test if the genome quality (N50 scaffold size, or other metrics) impacts the estimated rate.

f. Post-mapping QC: It would be useful if the authors can include a table with quality metrics for their

dataset, including post-mapping coverage, ti/tv ratio, % CpG transitions, etc. The authors mention this in the main text but I couldn't find a table with the details in the supplement.

3. UCE phylogeny: Comparing mutation rates with phylogenetic estimates is a good idea. However, I am surprised why the authors use UCE to compare the rates. The authors should be using putatively neutral sites for the comparison vs. UCE which is under strong constraint across species and an unreliable proxy for neutral substitution rates. The authors can use published phylogenies from UCSC browser for this analysis or perform the phylogenetic analysis for regions away from genes or ancestral repeats to get a more reliable inference. Moreover, it would be useful if the authors can show that the estimated rates are not sensitive to the assumed divergence times which are tentative and likely disagree with the mutation rates inferred in this study. Finally, some quantitative assessment of phylogenetic and pedigree rates would be useful (more on this #6).

4. Effect of domestication: These conclusions seem speculative. The authors compare diverse species that vary in a range of criteria and suggest that the differences observed are related to domestication. Instead, the authors should perform comparisons of closely related species- for example, wolf and dog, and then see if after accounting for the within species differences, there are significant differences across species that could be associated to domestication. It is possible there are very few such comparisons available, in which case result should be removed or described as a tentative association.

5. Comparison with previous studies

A number of studies have previously reported pedigree-based mutation rates. It would be useful if the authors compare those findings with rates/ patterns inferred here.

a. Estimates of previously published mutation rates: For over a dozen species, pedigree mutation rates are available from previous analysis often with larger number of trios per species. It would be useful if the authors can show how their results compare with previous reports, especially after accounting for parental age.

b. Estimates of male bias: A recent study (Wu et al. 2020) showed that across mammals the male bias in mutation rate is ~3-4:1 across a range of mammals. This seems consistent with the authors results, though they find a non-negative slope across vertebrates. It would be useful if the authors can explicitly discuss the differences in the observations in the two studies and discuss how this might relate to the relative contributions of DNA damage and repair in contributing to new DNMs.

7. Comparison of micro and macro evolution: It would be useful if for pairs of species, the authors can compare how the yearly mutation rate compares with heterozygosity and pi and phylogenetic rates. The authors compare broad scale patterns but would be useful to provide more detailed comparison.

For example, the authors suggest their results are consistent with hominoid rate slowdown hypothesis. However, given the large variation in the rates within species groups, the signal is not obvious. The rates in pedigrees of OWM in this study are puzzlingly lower than previous estimates, including from another study led by the same authors. The ratio of mutation rates across primates

are -

Chimp/H = 1.59

Gibbon/ H = 1.97

OWM/H = 0.83-1.43

NWM/H = 1.55-3.53

More importantly, the quantitative estimates from phylogenetic analysis suggest much lower differentiation in substitution rates across these species pairs, see Moorjani et al. PNAS 2016, Kim et al. 2009. While selection is a possibility, the authors should include quantitative comparisons of the phylogenetic and pedigree rates across species and reassess if the patterns agree / disagree with previous model predictions.

Minor comments:

1. Heterozygosity: The authors infer heterozygosity for all individuals in the trio, which is likely to be underestimated as related individuals are included. The authors should exclude the offspring and rerun this analysis only for the parents. Also removing closely related parents is advisable (which can be assessed by estimating the %IBD shared across the two parental genomes).

2. Drift-barrier hypothesis: In order to investigate this, the authors compare current mutation rate with historical N_e . This choice is odd as I don't see how historical effective population size should have any impact on the current mutation rate. Moreover, N_e is correlated to a number of LH and so the authors should examine if the effects are driven by N_e or other factors. Perhaps using the PGLS or PPA approach.

3. Estimation of yearly mutation rate: The authors use a "weight of paternal contribution" – Is this the alpha or male bias estimated using phased DNMs? Please clarify?

4. It would be useful if the authors can discuss a limitation of their study that it is based on samples from zoos and animal centers, which may have different life history traits than the wild. For many analysis, they interchangeably use the life history parameters of their sample or in the wild, current or historical estimates. These approximations maybe necessary but could lead to a bias the associations.

5. The following sentence should be revised to reflect that the rates in human trios differ by 40% or removed.

"Note that our estimate for humans is consistent with the previous estimates based on larger pedigrees at $\mu_{\text{yearly}} \sim 0.42 \times 10^{-9}$ mutations per site per year 25,72."

6. Supplementary table with all DNM information should be provided.

For each species, please report

<species> <trio> <genomic location> <paternal> <maternal>

These results will be helpful resource for others interested in investigating mutation rates and

patterns in future studies.

8. Rates for X and Y chromosome: It is unclear why most DNM studies leave out sex chromosomes. It would be useful if the authors can provide DNM rates for X and Y chromosomes as these can be very useful for understanding sex-bias evolutionary patterns.

Author Rebuttals to Initial Comments:

Please find point by point response to reviewer comments below

Reviewer #1

This paper represents a massive advance in our understanding of mutation-rate evolution, at least in vertebrates. Using trio data, the authors have been able to estimate the mutation rate in a wide range of vertebrate species, with replicate family analyses being performed within a substantial fraction of them. I've been waiting to see someone do this for quite a long time. The best estimates of mutation rates come from long-term (100s to 1000s of generations) mutation-accumulation studies in large numbers of replicate lines (often 50 or more). Although this gold-standard approach is unfeasible for vertebrates, the next best option, single-generation parent-offspring trios, becomes feasible owing to the large genomes and the relatively high per-generation mutation rates.

I am strongly enthusiastic about this paper, although I also have a number of issues with it from both an analysis and interpretation perspective. I raise these issues in the interest of enhancing the quality of the overall presentation, and I am confident that the authors will be able to do so. The following comments are in approximate order of appearance in the paper.

Response: We thank the reviewer for this enthusiastic comment on our work.

Line 56 – This is perhaps a slight misstatement – the lowest rates are generally in unicellular eukaryotes, not prokaryotes.

Response: We thank the reviewer for this comment and we have now corrected this statement (page 2 line 62).

Line 89 – typo; should be highest.

Response: We have corrected the typo.

At the outset -- Some indication of the reliability of the individual data needs to be given (this is never displayed in the main paper). From the first supplementary figure and the 2nd supplementary table, it is clear that the number of observed mutations per almost all trios is in the range of just 1 to 50, with about half apparently below 20. There are standard ways to get confidence limits based on an assumed Poisson distribution of numbers of hits, so this could be done for each estimate. Knowing this would be helpful in interpreting a number of the patterns the authors point out – some reviewers might object that these are in some cases not particularly strong, but if they hold in the face of error-prone data, this objection will be somewhat dissipated.

Response: The reviewer is correct that confidence intervals on estimated rates were missing from the previous version of the manuscript. To also answer subsequent comments from this and the other reviewers, we have now updated Figure 1a to show the per generation rate with confidence intervals instead of the yearly rate. The confidence intervals were calculated based on binomial variance using the binconf function in R based on the number of hits on the

callable genome. When multiple trios were available, the 95% CI was estimated based on the sum of hits for multiple trios on the sum of callable genomes.

Also of considerable interest here is the variation among trios within the same species. In principle, the number of observed mutations should be close to Poisson (assuming similar numbers of sites analyzed). However, this need not be true in a comparison among trios in the same species. As for any trait, there must be standing genetic variation for the mutation rate (and a few studies in invertebrates have shown up to a 3 or 4 fold range of variation; and I'm surprised that in a study of this size an occasional strong mutator wasn't found), so it would be nice to know the degree to which the observed variation exceeds the null expectation. If the level of variation is too low, I would be worried that the analyses have been overly aggressive in trying to confine their final results to some preconceived range of reality. The bottom line here is that the authors have a real opportunity here to clear up a common problem in the literature – it is common for authors of mutation-rate papers to make a big deal about two- or three-fold differences in mutation rates among species, and to try and concoct ecology (or otherwise) explanations; the natural range of variation within species is at this level, this has significant implications for the validity of these types of arguments.

Response: We agree that the variation among trios of the same species is key information and we have now added it to Figure 1a for the species with multiple trios available. However, we should acknowledge that the null expectation for the variation distribution is unknown for most species. From our data, we can see that at least some of the point estimates for several species with multiple trios are outside the binomial confidence interval (17 species with point estimate outside the 95% CI out of 29 species with multiple trios). We believe it is not trivial to interpret this pattern of variation when the null distribution is unknown for so many species. We also note that a very recent study (doi: 10.1038/s41586-022-04712-2) using more than 10,000 human trios reported hypermutability in only 12 individuals. This hypermutability could in five cases be ascribed to chemotherapy, suggesting that modifiers of significant effects are relatively rare in contemporary humans.

Age effects:

To what extent are the per-generation results a function of the age of the parents; ideally, one would want these ages to be roughly the mean age at reproduction in the wild. This becomes particularly relevant as the authors try to normalize everything to per-year rates. Ideally, what we want to know is the average mutation rate for parents at the average age of reproduction in the wild. A number of the ages of individuals (sup table 3) seem overly high or overly low for this target age, so this warrants a bit more attention. Even if we do not always know the average situation in nature, this is where the mutation rate evolved.

Response: We completely agree with this point. Estimating a yearly rate from a single trio with parents reproducing at an age very different from their generation time in nature will cause estimates to be heavily dependent on the assumption about how the rate scales with parental ages. Previously, we assumed a linear dependency which is at odds with the observation here, and in previous studies, of a large influx of early mutations (during early development and pre-puberty). To account for this, we have now chosen to model the

expected relationship of generation mutation rates as a function of generation time using a model that estimates both an initial influx of mutations followed by accumulation in males and females at different rates (details of the model are provided in a new Supplementary Note 1). We show that this model provides an acceptable fit to our combined data and it emphasizes a remarkable constancy of early mutations across the many vertebrate species investigated. We believe that our model-based approach provides more reliable estimates of both yearly and per generation rates for species at generation time.

Scaling by absolute time does not necessarily resolve this issue, as the scaling factor could differ among species. Indeed, there is a more general need for caution here because. To my knowledge, there is no compelling evidence that the mutation rate scales linearly with absolute time, which is the scaling the authors resorted to. Supposing the extreme, where for example, mutations in the germline might set down in the initial oocytes/spermatocytes with quiescence thereafter, or vice versa, initial quiescence followed by a progressive increase after maturity. In either case, the sort of treatment employed in this study could be quite misleading, for without accounting for the nonlinearity, the per-year estimates will be potentially strongly biased by the age of the parents. In the former case, lower values would tend to arise in species with longer generation lengths, purely as a consequence of dividing by a larger number; the data in Figure 1 vaguely look like this. The authors seem to acknowledge this issue, but do not account for it (lines 190-208).

Response: We thank the reviewer for raising this issue and we agree with the reviewer that how the mutation rate scales with time remains an unresolved question. Due to this comment and further comments by other reviewers, we decided to estimate the mutation rate based on a model that takes into account the potential non-linearity of mutation accumulation over time, as explained above and in the new supplement. Indeed, in Figure 1b, it can be seen that the accumulation of mutations before birth seems to be similar across all of the species. Due to a large intercept at maturation time, dividing the per generation rate by the age of the parents for the species that reproduced right after maturity seems to overestimate the yearly mutation rate. Moreover, we now used this model to re-estimate a per generation rate, at the generation time for each species. These model-based estimates of the per generation rate are now also used for the analysis of the variation in rates across species and the correlations of the estimated per-generation rates with life-history characteristics including investigation of the drift-barrier hypothesis.

Of course, the information to do so is largely lacking, but a more forceful caveat is probably warranted, and as noted below I'm not entirely convinced that per-year rates are what is relevant from an evolutionary perspective. Natural selection operates on the per-generation rate, which is the natural time scale for population-genetic processes.

Response: We agree with the reviewer that natural selection should primarily act on the per generation rate. Another recent study appeared after our initial submission (Cagan et al. doi: 10.1038/s41586-022-04618-z), though, suggested that lifespan (which correlates with generation time) may also be an important determinant, since the total somatic mutation rate in the colon was found to be fairly constant across species. We believe that it is not yet possible to disentangle whether the germline mutation rate or the somatic mutation rate is

most important for modifying the mutation process across vertebrates, including how these two rates feed into each other.

We have now mainly used the estimated yearly rates to study the pace of evolution across species and how this relates to the longer term evolutionary rate, which is important for phylogenetic dating. For evolutionary forces acting on the rate, we are now using per generation rates as suggested by the reviewer. Therefore, we have now changed the life history trait analyses to investigate the impact of each trait on the per generation mutation rate. We have now used the modeled per generation rate to remove any possible effects of independent ages of reproduction, and instead compare the per generation rate at the generation time for each species.

Line 110 – what is meant by a “weak base”?

Response: “Weak bases” referred to the weaker hydrogen bond between A and T pairing than between C and G we have now replaced this term by “strong base pairing to weak base pairing” (page 4 line 121).

Plots 3f and 3g – Presumably the scale in the plot is based on natural logs, but things will be more easily interpretable if log10 is used.

Response: As suggested by the reviewer, we now used a log10 scale for both the x and y axis as suggested by the reviewer for the plot of the effective population size (now Figure 3d).

Issues regarding the drift-barrier hypothesis:

Here the authors have an enormous opportunity, as one of the leading hypotheses for the evolutionary diversification of the mutation rate across the Tree of Life is the idea that the efficiency of selection for mutation-rate reduction is compromised in species with low N_e .

Response: We agree with the reviewer, and a main objective of the project from the onset was to investigate this hypothesis across vertebrates to complement previous reports over much longer phylogenetic distances. We agree that the drift-barrier hypothesis well explains the broad-scale differences in per-generation rates across the tree-of-life, but it is an open question how fast the rate will equilibrate over more shallow time-spans, which depends on the rate at which mutational modifiers occur. We can only estimate the effective population size over the past 1 million years (20,000 to 200,000 generations depending on the species) and this estimate is for the neutral effective population size which may differ from the unknown effective population size of relevance for natural selection if populations undergo complex demographics such as population structure and population bottlenecks.

The authors use the PSMC method to estimate harmonic N_e in their study species, but this needs some further discussion, as the PSMC often results in very similar demographic patterns in diverse data sets, suggesting that the method can lead to biased results owing to the many internal assumptions being made. There are alternatives that could be used, and perhaps should explored if for no other reason than to demonstrate the robustness of the results.

The simplest approach would be to simply start with silent-site heterozygosity, set this equal to $4 * N_e * u$, and then factor out the mutation rate (u). This does assume that average heterozygosity within a single individual is representative of the species-wide diversity, but the PSMC makes the same assumption about the utility of single individuals.

Response: We thank the reviewer for this suggestion, as we agree that there are some limitations to the PSMC analysis. Taking into consideration the reviewer's suggestion, we have now estimated N_e using $N_e = \pi/4\mu$ using nucleotide diversity for π and the UCE substitution rate for μ . We were pleased to see that N_e calculated with PSMC and N_e calculated with $\pi/4\mu$ were strongly and highly significantly correlated ($r^2 = 0.91$, $p = 2.2 \times 10^{-16}$). Therefore, this additional analysis validates the negative correlation between the per generation rate and the effective population size. However, nucleotide diversity was inferred using the whole genomes and was not restricted to silent sites (see later response). We have now added this in a Supplementary Figure 8.

With either approach, the mutation rate itself must be used to get an estimate of N_e (a point that is not made by the authors). This introduces a couple of key issues. 1) The measure on the x axis (N_e) is subject to estimation error, which means that the true slope of the regression must be stronger than what the authors show (this effect might be factored out by subtracting the error variance of estimate of N_e from the total variance of the N_e used in the denominator of the regression coefficient. 2) Because estimates of N_e are a function of the estimate of u used, there can be an intrinsic negative correlation between the two estimates (i.e., if u is overestimated, the derived N_e estimate will be underestimated). This second problem is not too severe if the errors in estimation of u are sufficiently small (it wouldn't be a problem at all if u were measured without error....) In any event, these two sources of potential error operate in opposite directions. Moreover, the first source of error applies to all of the regressions that the authors apply, including those based on life-history traits (which are certainly not error-proof).

Response: We did not make this point clear enough in the previous version, but we did indeed consider this issue of circularity. In our analysis of N_e , which now uses both PSMC and $\pi/4\mu$, the per generation rate used was not the pedigree rate, but the UCE rate. We have now added this information to the manuscript (page 11, line 302). However, this does not completely rule out errors in the estimation of mutation rates, so we have now acknowledged the limitation of estimating a correct N_e in this section (page 11 lines 297-313).

Still another kind of issue here concerns the interpretation of the various regressions on life-history parameters. As the authors note, such parameters are often inter-correlated, making strict interpretations difficult. However, life-history traits also may often be correlated with N_e , leaving the open the question of causality vs. indirect correlation in the evolution of the mutation rate. For example, the observed increase in the per-generation rate with generation length could be a consequence of the drift barrier, with longer-lived species tending to be larger in size and with lower N_e .

Response: Following the reviewer's suggestion, we have updated the life-history trait analyses with the per generation rate. We agree with the reviewer that it is hard to distinguish causality and indirect correlation. Indeed, we found a weak direct correlation between N_e and the per generation rate, but N_e is also correlated with generation time, which has a stronger correlation with the per generation rate.

Correlations between mutation rates and heterozygosities:

Response: We have now substantially reduced this part. First, because the correlations weakened when we used the new mutation rate estimates, and second, because the other reviewers found this section confusing. We therefore decided to focus on our strongest and most easily interpretable results.

From what is describe in the text, these appear to be based on genome-wide variation, whereas, as noted above, a comparison against silent-site diversity would be more meaningful. Indeed, it is difficult to ascertain what the authors are trying to do here. Total variation need not strictly be correlated with mutation rates (u). On the other hand, under the assumption of neutrality, silent-site variation (at mutation-drift equilibrium) as an expected value of $4 * N_e * u$.

Response: We agree that the silent-site variation is what is meaningful for the analysis and interpretation. Yet, the gene regions represent a small percentage of the genomes and the silent sites are even fewer. Moreover, with such a long evolutionary time scale, the silent sites would reach saturation and cause bias in estimation of mutation rate. Nevertheless, we believe that the genome-wide nucleotide diversity should be a good proxy of the silent site π . Thus we prefer to use nucleotide diversity, π , calculated based on the whole genome data to estimate N_e .

However, this part of the manuscript is quite confusing, as at line 228, it is argued that π is proportional to N_e and u , leaving me to wonder whether the authors really are reporting silent-site variation. To make things even more confusing, it is stated that N_e “depends on the mutation and generation time.” I am unaware of the basis of this statement, but it is not the reference given, and to my knowledge is not something that is commonly believed. Arguably, the direction of causality is reversed here.

Response: We thank the reviewer for pointing that out and we have now substantially reduced this section of the manuscript.

Use of per-year estimates:

Above, I outlined some of the problems in making these estimates (and I wish to emphasize that the power of the paper is in the observed per-generation rates), but there are additional interpretative issues when arguments towards the end of the paper rely on the former.

Response: We have been convinced by the referees that the per generation rate is most important for evolution of the mutation rate, and have therefore changed Figure 1 and the first

part of the results to focus more on the conserved per generation rate across vertebrates. We also modified the section on the life-history traits to focus on the per generation rate.

Is the ~100-fold range of variation in the yearly mutation rate really surprising? As noted above, selection operates on the per-generation rate, which is the natural timescale of evolution, and so depending on the life-history strategy, the per-year rates will just be adjusted accordingly. More generally, all evolutionary processes are most reasonably viewed as per-generation processes, so I am not convinced these data represent variation in “the capacity of lineages to evolve novel phenotypes, consistent with GMR-induced phenotypic evolvability across lineages and therefore plays a major role in long-term macroevolutionary processes.” On a per-generation basis, there is only a few-fold range of variation of the mutation rate among diverse vertebrate species, and it should be kept in mind that mechanisms of evolution depend on the joint distribution of mutation rates and N_e , so this quote strikes me as a bit of an exaggeration that I don’t think is an essential selling point for this paper.

Response: We agree with the reviewer on this and we have now removed this part, instead we focus on the ~40-fold variation in per generation rates in species with large variation in life-history traits.

The following conclusion also does not ring correct to me: “The strong correlation between fecundity and GMR is consistent with an evolutionary trade-off between the number and quality of offspring, such that GMR is a general indicator of average offspring quality optimized by natural selection in each species’ gene pool.” If there is any tradeoff between offspring number and quality, it is only in a physiological sense (more offspring implying less provisioning), not in a genetic sense that has anything to do with mutation or genetic variation. So either more explanation here is desirable, or I would drop this as well.

Response: Our new analyses show that the generation time is the most important trait that influences the per generation rate evolution. Thus, we have removed our previous claim.

Similarly, “Because domesticated animals have been consistently artificially selected for high fecundity rates, their exceptionally high yearly mutation rates can be considered as a direct test of our conclusions based on wild animals.” It is unclear that the animals the authors are referring to are really under selection for higher growth rate (guinea pigs, cats, and dogs?), and in any event, domesticated species have very low historical effective population sizes, so the pattern observed could be a simple consequence of the drift barrier. The authors do mention this, but combined with the more general relationship with N_e in nondomesticated species, this would seem to be a more unifying explanation.

Response: Domesticated species might have a very low N_e at the beginning of the domestication process. However, in our dataset, we found a higher N_e for domesticated species compared to non-domesticated species. Thus, N_e does not seem to explain the high mutation rate observed in those species. Moreover, we found that when we use the modeled rate of the domesticated species, using the generation time they would have in the wild, the yearly rates are not significantly higher than in the other species. Instead, the realized yearly

rate after domestication is higher because domestication leads to a much shorter generation time under artificial conditions of domestication, which probably also means that the generation time of these species is now smaller (due to earlier maturation) as a result of artificial selection. Therefore, we now interpret the higher yearly rate observed in domesticated species as a consequence of the reduction of generation time more than selection for a higher growth rate.

Referee #2 (Remarks to the Author):

I like this paper and I think it will make a valuable contribution to the literature – studies of mutation rate variation have tended to either focus on single species with relatively large amounts of data (through parent-offspring analysis) or across many species with relatively small amounts of data (using homologous DNA sequences), but this study combines the strengths of both approaches. However, the study currently has some weaknesses that could certainly be overcome with a little additional analysis, and then the conclusions would be much more robust.

Response: We thank the reviewer for this encouraging comment on our work.

The study relies on the phylogenetic analysis, which is dependent on strong assumptions about calibrations and other aspects of the evolutionary process, yet the phylogenetic analysis is described in only a few sentences. Many of the assumptions underlying this analysis could be questioned (for example the calibration Primate/Glires split at 61). The point here is not to say whether the assumptions are wrong or right, but that we rarely have confidence that we know for certain that particular models, calibration dates etc are appropriate, and yet the results might be quite strongly dependent on particular dates. This could be circumvented by showing the results are robust to a wide range of assumptions (ie not just running on a sample of trees from one analysis with one model but on many independent runs with different assumptions), or by using an approach that does not require dating the tree or resolving the lower nodes of the tree – for example a sister pairs approach (This would also help to provide a counterpoint to PGLS, which relies on strong assumptions about the model of evolution underlying the traits - which are not discussed or explored in this paper). Simply saying that other people have used these calibrations is not sufficient defence of their veracity – actually many different calibrations have been used in the literature, and there are consequently a very wide range of date estimates for some of these nodes. The authors could rerun their analysis on different trees or on a wider range of phylogenetic hypotheses to prove that the results are robust.

Response: We agree with the reviewer that phylogenetic analyses rely on strong assumptions that can bias the final results. In our case, the estimated substitution rates are indeed affected by the calibration nodes. Attempting to quantify this and to add robustness to our results, we compared estimated substitution rates using the 14 initial calibration points with the inferred substitution rates when using only 13 calibration points (with 14 iterations to remove each

calibration node one by one). In general, there is a strong correlation between the rate estimated with 14 or 13 calibrations ($r^2 = 0.953$, $p = 2.2 \times 10^{-16}$). However, we found that some of the calibration points, especially on recent nodes, can have a stronger impact on the estimated substitution rates. For instance, removing the two bird nodes (7 or 10), the gekko node (9), the Canidae/Arctoidea node (13), or the Glires/Primate node (8) changes the substitution rate as anticipated by the reviewer. However, the overall correlation we report still remains significant even under these circumstances. Also, the tree with the 14 calibrations points was similar to previously published trees from the scientific literature. We have now added Supplementary Figure 15 showing those results.

I was confused by the macroevolutionary analysis that claimed to link the rates from the parent-offspring estimates to lineage level. This is done by estimating the average genome-wide heterozygosity for each species, but I am not clear on the logic of why this tests the continuity of microevolution and macroevolution. This needs much clearer explanation. Why not also estimate the mutation rate from the phylogeny by inferring the synonymous mutation rate? That way, the study could compare the rates from the parent-offspring estimates to the lineage level averages – this has been a topic much discussed and debated so this study could make a useful contribution.

Response: The other reviewers also found this section confusing. Moreover, when we used the modeled mutation rates, these results weakened. Therefore, we decided to reduce this section to focus on our stronger results. We now only present the correlation with UCE substitution rates.

The synonymous mutation rate method is likely to be most appropriate for more closely-related species. In the evolutionary timescale across all vertebrates, synonymous mutation rates tend to reach saturation, making distant relationship estimates unreliable. The UCE and their flanking regions are present in all examined species and even though the evolutionary forces conserving these sequences is not well known, they do appear to scale well with evolutionary time overall. We therefore believe they are a reasonable proxy for the underlying pace of neutral evolution. The rate estimated from these genomic regions should therefore in our opinion represent the relatively lineage-specific evolutionary rates and thus the underlying yearly mutation rate. The correlation we report with pedigree estimates indirectly supports this assumption.

The higher mutation rates in domesticated animals is interesting and reasonable, but actually this result is not particularly strong. The distribution of values between the domesticated and non-domesticated are not distinctly different, and there are many non-domesticates with higher values than the domesticates (fig 4), so it's simply not true to say “the exceptionally high mutation rates observed in domesticated animals”. The average is higher but this average is meaningless without phylogenetic correction, and actually there are many non-domesticates with higher rates than most of the domesticates. A better way to analyse this would be to compare each of the domesticated lineages to their non-domesticated relatives, not to lump them all together. I was surprised that there no reference at all to any published

studies on this topic, despite decades worth of research on this topic, and also no reference to alternative explanations such as relaxed selection or selection for high degree of generation of genetic novelty (as proposed, I think, in the classic paper by Burt and Bell on chiasma frequency in domesticated mammals). This needs to be more thoroughly investigated and argued more robustly if it is to form a key finding of this study – at the moment it seems to be essentially descriptive, anecdotal and not considered in the context of a large body of existing evidence on this topic.

Response: We agree with the reviewer that the results of this section could have been stronger. Using the modeled mutation rate, we now show that the domesticated species would not have a significantly higher yearly mutation rate if we had used the same generation time as that in the wild. Therefore, the selective pressure to reduce the generation time under domestication seems to be the cause of the higher yearly rates observed. This is due to the nonlinearity of mutation accumulation over time, and especially due to the strong intercept at maturity. We agree with the reviewer that selection might also function on other biological processes, such as the recombination rate. We also agree with the reviewer that the best way to analyze this data would be by comparing the rate of domesticated species with their wild relatives (e.g. dog/wolves, domestic cats/wild cats). However, collecting trio samples from wild species is a challenging task, limiting the application of this approach. Therefore, we addressed this criticism by comparing all of the species with PGLS in order to apply a phylogenetic correction.

In fact, the study does seem quite light on references to previous studies of variation in mutation rate between species, which given the amount of research done on life history patterns in mutation rates seems rather odd.

Response: We have now added more recent references to the manuscript.

Final point: please make the alignments freely and easily available when published. This is critical not just for people to be able to repeat this analysis for themselves, but also provides a useful resource for future studies.

Response: We have shared all data and analysis pipelines for full reproducibility and use for future studies and we regret that we did not include this information in the initial submission. We had already added all the raw sequences to NCBI, we also added all the scripts and the tree in GitHub

(https://github.com/Lucieabergeron/vertebrate_rate/tree/main/1.%20phylogeny), and the UCE probes are available online (<https://www.ultraconserved.org/>).

We have now further added the final alignments used to generate the tree in Figshare (private link: <https://figshare.com/s/4c2db0449ef5d3dffaa3>).

Summary of the key results: OK

Originality and significance: if not novel, please include reference: the results are in line with previous studies (e.g. Welch et al doi: 10.1186/1471-2148-8-53.) but here with a bigger and better dataset, so the originality is in the data used rather than the findings.

Response: We disagree with the reviewer on this point. Our study represents a major advance over Welch et al. finding the ecological and life historical factors that affect the substitution rate are a fundamental question. Thus it is not surprising to see that many previous studies have attempted to address this question. However, different from all other studies including Welch et al., which could only address this question by studying the substitution rate estimated from genome comparisons, we used triads to assess the links between germline mutation rate and life history traits in a comparative context. Previous studies using genomic substitution rates also have many limitations:

- 1) Substitution rates estimated from genomic comparisons will be biased because, with such a long evolutionary time scale spanning all vertebrates, analyses can only be performed on genomic regions that are highly conserved across all species. These regions are under strong purifying selection and the substitution rates, possibly resulting in smaller mutation rates than for neutrally evolving sequences. By contrast, our approach of sequencing triads from multiple species directly quantifies the genome-wide mutation rates across the generations.
- 2) Whether or not the substitution rate estimated from cross-species genomic comparison could reflect the mutation rates across generations was unknown until our study revealed this. Additionally, our study also reveals many novel factors that shape the GMR across vertebrates. Thus, the novelty of our study lies not only in the new data, but also in our capacity to tackle long-standing evolutionary questions.

Data & methodology: validity of approach, quality of data, quality of presentation: OK but not enough detail given

Response: We have now added a detailed explanation of our data and uploaded all of our scripts and files to the public domain.

Appropriate use of statistics and treatment of uncertainties: Little formal treatment of uncertainty.

Response: We have added more details of the statistic methods, especially regarding the 95% CIs of our point estimates and testing of the robustness of the calibrated tree.

Conclusions: robustness, validity, reliability: Not possible to evaluate given the information in the paper.

Response: We have now provided much more information, which should allow the reader to better judge the quality and reproducibility of our results.

Suggested improvements: experiments, data for possible revision; suggest applying to range of phylogenetic hypotheses or using an approach robust to phylogenetic uncertainty e.g. sister pairs.

Response: We have added more supplementary tables containing the raw data and have added a phylogenetic comparison of calibration to test the robustness.

References: appropriate credit to previous work? No.

Response: We have cited more of the previous literature in the revision, including a handful of studies that have appeared since our initial submission. If the reviewer has further specific suggestions, we will be more than happy to incorporate these.

Clarity and context: lucidity of abstract/summary, appropriateness of abstract, introduction and conclusions: yes, with the exception of the microevolution-to-macroevo- lution argument which is not clearly explained.

Response: We have now shortened this part substantially.

Reviewer #3

The germline mutation rate (GMR) is a key parameter to measure the pace of evolution. However, it is difficult to estimate GMR accurately or even to establish whether (or rather, how) it varies over evolutionary times and is influenced in response to molecular mechanism (i.e. mutagenesis and/or DNA repair), environmental factors or life-history traits (i.e. metabolic rate, generation time, reproduction mechanism, fecundity, population size...). While there are many different ways to estimate GMR, in this study, the authors have used direct whole genome sequencing (WGS) of 151 vertebrate parent-offspring trios from 68 species (mammals, fishes, birds and reptiles) to directly quantify species-specific transgenerational GMR and mutational patterns.

They report ~40-fold variation of GMR across species, which corresponds to a large (~ 125-fold) variation in yearly mutation rate ('u-yearly'), and a variable level of parent-of-origin effects among species/taxa. Their data point to fecundity (and selection via domestication) has a major life-history trait that may explain the variation in mutation rate among species. Their data are consistent with the concept that GMR can account for the observed genetic diversity (within individual genomes, their species or across species), confirming the idea that genetic changes (micro-evolution) and genomic (selective) macro-evolution occur as part of a continuum of effects.

Response: We thank the reviewer for this nice summary of our study.

I read this well-written manuscript with great interest and enthusiasm. In my opinion, this work represents a tour-de-force in sample collection, data acquisition and analysis. The authors have generated a very large, high-quality dataset of genomic sequences of family

trios following a systematic and consistent approach. This represents a unique and invaluable resource for the research community. It gives them the ability to compare directly de novo mutation (DNM) load across species and taxa, and explore patterns of mutations to test hypotheses (male/female ratio, mutation signatures, micro- vs macro evolution patterns...).

Response: We thank the reviewer for this positive comment on our work.

However, the drawback of presenting such an extensive dataset is the difficulty in choosing what to analysis or how to interpret the data given the many biological/environmental/reproductive strategies. While, overall, the authors have done a nice job in singling-out some features and general trends, the discussion (in comparison to the wealth of data) felt relatively superficial. I am not sure if there is a way to remedy this and overall the discussion is well balanced in highlighting the limitations of the work presented.

Response: We agree with the reviewer and have now focused our conclusion on the results where we have the strongest support from the analysis.

My comments below are fairly minor and mainly focus on the presentation of the discussion:

While I understand the importance of giving an overview of the data, I feel that the focus on the very wide variation of GMR across species (page 3 paragraph, entitled “More than two orders of magnitude variation in the GMR of vertebrates”) may be quite misleading. There are clearly limitations about the approach that will unavoidably result in uncertainty in the GMR estimates for the trio and/or the extent to which it reflects the GMR of its cognate species:

1. As mentioned by the authors, DNMs are rare events. Some of the GMRs (u-generation) are based on a very limited number of DNMs per trio (for example for the snowy owl only 2 DNMs were called, and 5 DNMs for the Griffon vulture – of note, these are the 2 species that are singled-out in the text for the lowest u-yearly rate). The ability to call DNMs depends on the quality of the reference sequence and the proportion of the ‘callable’ genome’ which may be problematic for the less-well characterized species (for example down to 17% for the Common carp) and/or for the species whose genomes contain large segmental duplications/genome duplication events.

While u-generation is described as varying by 40-fold across species, the general trend described on page 3 of GMRs being highest in birds and lowest in fish with mammals and reptiles in between (birds<mammals<reptiles<fish) only differs by 7.7-fold, with largely overlapping CIs. To my mind, this highlights the remarkable conservation of GMRs across species/taxa with very different life-traits and mechanisms of reproduction rather than their differences?

Response: We thank the reviewer for this useful comment which inspired us to adjust our interpretation of the data. Indeed, we agree that the per generation rates are quite conserved among vertebrates. Thus, we have now:

1. Changed Figure 1a to present per generation rates instead of yearly rates.
2. Depicted the uncertainty associated with our estimates as 95% CIs.

3. To investigate the impact of the reference genome, we added some supplementary figures showing the impact of a reference genome with a large percentage of Ns or gap numbers on the callable genome and the final result (Supplementary Fig. 12). Indeed, a reference genome of lower quality (with more gaps or Ns) will induce a smaller callable genome and a higher false-positive rate. Importantly, however, the resulting mutation rate estimates are not significantly correlated with reference genome metrics (as both the callable genome and the false positive rate changes).

- Moreover, to calculate the yearly GMR (u-yearly), the authors used the combined parental ages of the specific trio (when available). Using ‘u-yearly’ allows them to make the claim that GMR varies by 125-fold across species. While it may appear logical to use the average parental age of the trio to derive u-yearly, this introduces more uncertainty on the GMR estimates as parental age of the trio may not reflect the generation time or be indicative of the average reproduction age/sexual maturity of the species in the wild. (i.e. How relevant are measurements from trios maintained ‘artificially’ in a zoo in respect to wild populations – for example, the emperor penguins pair are 36 yrs old, but would only live a max of 15-20 yrs in the wild). Moreover, some of the longer-lived species will have relatively long pre-pubertal period (which I assume have not been subtracted – for example, the emperor penguins reach sexual maturity ~5-6 yrs of age), likely further biasing u-yearly estimates. Importantly if u-yearly allows a direct comparison for species with very different lifespan, the key impact of GMR is not the yearly rate but the rate across generations. Hence generation time is a key parameter in the interpretation of GMR variability.

Response: We agree with all of the points raised by the reviewer here. The per generation rate is less biased than the yearly mutation rate and therefore, we have now changed Figure 1a and the text of this section accordingly. We also now focus the life history trait analysis on the per generation rate instead of the yearly rate.

Moreover, based on this comment and the other’s reviewer’s comments, we have now introduced a model to derive the yearly mutation rate. Indeed, when multiple trios are available, it is possible to estimate an increase of rate with parental age and then estimate an average yearly rate at generation time. In our case, we lack this information for the majority of the species in our dataset, and small within-species sample sizes do not allow inference of the slope of increase for each species individually. However, we made a general model for all species to take into account the non linearity of mutation increase with parental age. Therefore, the estimated rates are now more representative of the species rate at generation time. With this, we could reinterpret the domestication results, as using the generation time of the wild would substantially reduce the yearly rate. Therefore, the high yearly rates observed in those species were mainly due to the reduction of generation time associated with domestication. We now focus our discussion on the importance of the generation time for the interpretation of GMR.

2. The authors also assessed the ratio of DNMs that could be phased to their parental chromosomes, to derive the male-to-female ratio (α). Again in this case, they analyse α by taxa and convincingly discuss some general trends in the paragraph entitled “Variable male-

driven evolution across vertebrates” on page 6 (and Sup Table S3) showing that overall their data are consistent with the idea that differences in gonadal stem cells biology between males and females drive GMR sex bias across taxa.

As mentioned by the authors at the end of page 7, “parents always carry a certain minimum number of mutations in their gametes regardless of their age”. An important source by which these ‘age- and sex-independent’ mutations can arise in parental gametes is gonadal mosaicism. With this in mind, I note that on page 3 the authors mentioned that: “we generated high-depth genome sequences (average coverage > 67X) for 323 individuals, representing 151 trios of 68 vertebrate species”. Although I can see that a parent pair can be part of several trios, I wonder how many ‘families’ actually had more than one offspring – i.e. some families will be quartets/quintets. Families with multiple offspring offer the possibility to quantify the impact of mosaicism on GMR, which maybe an important contributor to GMR. In a BioRxiv study from 2017, Harland et al (<https://doi.org/10.1101/079863>) showed that in cattle, a significant proportion (up to 30-50%) of apparently new mutations actually occurred in early cleavage embryos resulting in frequent mosaicism and a high rate of sharing of multiple mutations between siblings. Whether this phenomenon is common to other species (or other domesticated animals in particular) would be of great interest and could potentially alter the GMR.

Response: We agree with the reviewer that siblings are very useful to dissociate the germline mutations (which can present a sex bias) from the post zygotic mutations arising in the parents' primordial germ cells (which should not be sex bias). This information was lacking from our previous version of the manuscript. We have now added a supplementary table with this information (species with related offsprings, number of siblings, number of mutations, and number of shared mutations). In our dataset, 27 species were represented by pedigrees containing at least 2 siblings (sometimes more, sometimes half-siblings). Interestingly, it appears that the group with α close to 1 (the fish and reptiles) also had a higher number of mutations shared between siblings. Therefore, this could add an alternative explanation for a lower sex bias in those groups: despite the difference in gonadal stem cell biology between groups, differences in post-zygotic mutations might also contribute towards explaining variation in α . We have now added this in the main text (page 8 lines 204 - 211) and in Supplementary Table 4.

3. It is likely that the dataset is not large enough, but it may be of interest to quantify GMR within the coding genome of the different species - some species exhibit very ‘compact’ non-coding genomes and may not be able to accommodate higher GMR for this reason.

Response: Similarly to the previous suggestion, we agree that this is an interesting point to explore. We have looked at the location of mutations for the 31 species with an annotated reference genome and found that on average 2.3 % (se: 0.61 %) of the mutations were in coding regions. There was some variability among species, but this did not appear to be a function of the ratio of coding to non-coding regions. For instance, the Siamese fighting fish had 1 out of 10 mutations in a coding region, representing 10% of the mutations. This species has a large portion of the coding region which could explain the high percentage. Yet, the walrus also had a high percentage of mutations in coding regions with 13 % (6 out of 47

mutations in coding regions). However, only 2 to 3% of the walrus genome is made of coding sequences, which does not explain the high ratio. Here again, our small sample size makes it hard to interpret the result from species to species, yet the overall results are interesting and we have now added this information to the Supplementary Table 3.

4. ‘Domestication’ and selection for breeds can occur for different reasons – for example could the higher mutation rate of domesticated species be driven through changes in effective population size of a ‘pure breed’?

Response: In our dataset, we found that the domesticated species had higher N_e than the wild species; Thus, the drift barrier hypothesis does not appear to explain our findings. However, in our new analyses using the modeled yearly rate, we found no significant difference between domesticated and non domesticated species. This indicates that the younger age of reproduction in those domestic species may be the cause of the observed high mutation rates.

5. If the authors are not aware of the Biorxiv manuscript from Cagan et al, (doi: <https://doi.org/10.1101/2021.08.19.456982>) entitled “Somatic mutation rates scale with lifespan across mammals” they may want to consult it. In this work, Cagan et al show that despite widely different life histories, including ~30-fold variation in lifespan and ~40,000-fold variation in body mass, the somatic mutation burden at the end of lifespan varied only by a factor of ~3, suggesting that, surprisingly perhaps, somatic mutation rates are evolutionarily constrained across mammals. Again this illustrates my first point, the GMRs are remarkably conserved given all the factors that influence mutagenesis and result in DNA errors.

Response: We thank the reviewer for this suggestion and found the Cagan et. al paper which have now been published interesting. The per generation GMR in our study varies by a factor of 40 across all species. We agree with the reviewer that this variant is relatively modest if we compare with the variation in other life-historical traits. We have added a citation to the Cagan et al. paper in the first part of the results. Different with the yearly somatic mutation rate which showed strong negative correlation with life span, our study finds that GMR increased with generation time and other traits related to fecundity, but to a lesser extent with lifespan. The difference between germline and somatic rates are that we have an early boost of germline mutations whereas the yearly rate is much more constant for somatic mutations (and in the colon the yearly somatic mutation rate is much higher than in the germline). When we will have many trios per species we should be able to test whether the effect of adding more years to life span also decreases the yearly germline mutation rate, but our current data does not really allow this.

Referee #4 (Remarks to the Author):

The authors have generated high coverage whole genome sequences for 151 trios from 68 species of vertebrates including 36 mammals, 18 birds, 6 reptiles, and 8 fishes. All the data was uniformly analyzed, and dataset and pipelines are expected to be released upon publication. This is the most comprehensive study of pedigree mutation rates in vertebrates

and is an important contribution for understanding the determinants of mutation rate. Moreover, it will be an important resource for studies of molecular evolution and molecular dating. The analysis performed is interesting, but I have some technical concerns about the results that I detail below.

Response: We thank the reviewer for this comment on our manuscript.

1. Large variation in rates across individuals within a species, within a clade and across clades: Figure 1 and Table S2 show that there is large variation in per generation and yearly mutation rates across individuals within a species and also within clades. This variation seems larger than expected based on life history traits alone. For instance, across 8 trios of quails (same parental ages) the per generation mutation rate varies by 4x, across 6 trios of guinea pigs by 3x and even across two trios of humans by 40% ($0.99-1.40 \times 10^{-8}$), despite very similar parental ages. Moreover, there is substantial variation within clades as much as across clades. It is not clear how much of this variation is random variation (due to small sample sizes) and systematic variation (due to pipelines – see #2). I think the authors need to try to quantify these effects before any inferences can be gained from this analysis. They should provide uncertainty for all estimated parameters– mutation rate per generation, yearly rates, mutation spectrum etc. and also address #2 for characterizing the systematic biases that can be introduced due to the pipeline.

Response: We agree with the reviewer that this information was missing from the previous version of our manuscript. We have now changed Figure 1a to present the per generation rate per species along with the binomial 95% confidence interval of the rate, together with point estimates where multiple trios were available to show the variation within species.

2. DNM identification in trios

Given that DNMs are very rare and there only ~70-100 mutations (out of 6 billion bp) in humans per generation, it remains challenging to reliably infer DNMs given sequencing errors and somatic mutations. To perform a reliable comparison across species, the authors apply a uniform pipeline across all the trios in their study. However, given that the source of DNA and reference genome quality differs vastly this may not be justified. Specifically, I am concerned about the following:

a. Higher somatic mutations in DNA extracted from tissues: The authors have used a range of samples to obtain DNA for sequencing including whole blood, tissues, DNA (possibly extracted many years ago). Each of these would have a different % of somatic mutations. It would be useful if the authors can include specific details about the tissues used in the study, investigate how the source of DNA impacts the estimated number of DNMs in the samples and adjust the pipeline accordingly so the inferred DNMs are in fact germline DNMs and not somatic mutations in the offspring.

Response: We agree with the reviewer that there may be differences in somatic mutations from different tissues. Yet, we believe the filters we used in the bioinformatic analysis are strict enough to remove those somatic mutations. For instance, the allelic balance filter will remove any candidate mutations detected in the offspring in less than 30% of the reads, which could be due to somatic mutations. We have now estimated the rate of potential

candidate mutations with an allelic balance of between 0.1 and 0.3, which could be somatic mutations (Supplementary Fig. 13d). We found no link of these rates to tissue type.

The tissue types are presented in Supplementary Table 1, but we have now added more details (skin, fins, larvae, embryo, or muscles). We have also added Supplementary Figures to show that our mutation rate estimates did not differ significantly depending on the tissue type or age of the samples (Supplementary Fig. 13). However, we found an effect of tissue type on the percentage of false-positive calls, with more false-positives for trios with other tissues than blood. We explored the confounding effect of the taxonomic group, as more tissues samples (fins and larvae) were collected from fish and reptiles and more blood samples were collected for mammals and birds, and we found an independent effect of both variables on the false positive rate.

b. Estimation of false positive rate: To infer the false positive rate, the authors applied another variant caller (bcftools) in the region of the candidate DNMs and removed sites which did not show similar genotypes as GATK. While using multiple variant callers is a useful way to reduce sequencing errors, applying it only to candidate DNMs is likely to bias the mutation rate downward. Importantly, one is more likely to miss a het in the parents and hence its critical to apply the same approach to all regions in the genome and not just the numerator. An important limitation of this approach for inferring FPR is that it is not an independent validation – since it is unlikely to identify errors/ biases in the earlier steps of the pipeline. It would be useful if the authors can show the validity of this approach by validating a subset of mutations by another method – either sanger sequencing or using publicly available multi-generation pedigrees.

Response: We believe that we inadvertently misled the reviewer by saying that we used different callers to detect the false positive calls. Indeed, one method is to call variants in the whole genome using multiple software, and then to compare all of the resulting candidate mutations. This method has been shown to be efficient in selecting true positive candidates, but tends to be computationally demanding (DOI: <https://doi.org/10.1016/j.cell.2017.08.047>). Here, we did not compare the two callers but only called candidates with GATK, as is the case for many studies estimating GMRs. We then used BCFtools to detect potential false-positive candidates. Our approach shares similarities to a manual curation method.

Secondly, we agree with the reviewer that the best way to detect false positive calls is through independent validation with Sanger sequencing. It was not possible to do this because of the very large number of species and candidates. However, we have already validated our analytical methods and pipeline using Sanger sequencing in two previous studies. Briefly, in the rhesus macaque (*Macaca mulatta*) mutation rate paper (DOI: [10.1093/gigascience/giab029](https://doi.org/10.1093/gigascience/giab029)), we validated 24 candidate DNMs using Sanger sequencing and found that our method correctly detected a false positive candidate, while all of the others were true positives. Additionally, in a recent study comparing GMR estimation methods (DOI: [0.7554/eLife.73577](https://doi.org/10.1016/j.eLife.2017.07.3577)), we validated 39 candidate DNMs with Sanger sequencing, again in a pedigree of the rhesus macaque, and found that our method efficiently discarded the false-positive candidates.

c. Estimation of false negative rates: To estimate the false negative rates, the authors apply filters to polymorphic sites and estimate the proportion of sites where true polymorphic site will be removed. This approach by design assumes that polymorphic sites and DNMs are the same, which is not necessarily true as variant calling approaches like GATK use machine learning algorithms that give higher confidence to heterozygous sites previously seen in the sample and hence lower confidence to rare variants and DNMs. Thus, a more reliable approach to inferring the FNR would be by simulations using methods such as bamsurgeon (<https://github.com/adamewing/bamsurgeon>) where a DNM can be artificially added to the reads and then the entire pipeline is applied to recover the DNMs. This approach should be applied independently for each species – while a lot of work – given the differences in genome quality and resources, estimates in one species may not apply to others.

Response: Two methods have been used in the literature to estimate the false negative rate: the simulation method, especially on low coverage data, and the method we applied. We agree that our method assumes that the proportion of true heterozygote sites outside the allelic balance filter is similar to the proportion of true *de novo* mutations outside the allelic balance. Yet, the simulation method also has caveats. Indeed, simulating mutations on the bam files and remapping the reads (as done by Bamsurgeon) would result in different parameter estimated in the region of the simulated mutation (such as depth, genotype quality, allelic balance). For instance, the depth and genotype quality in the adjacent positions to the simulated mutation could change. Therefore, in such a case, a mutation could be missed by the pipeline and count as a false negative call, while the genomic region should not be counted in the callable genome any longer if a new callable genome was inferred. Thus, we believe that the simulation method is meaningful only if a new callable genome is estimated, which is not what has been done in the literature. Given the lack of a standardized simulation approach, we choose to use the method that we used in our previous work to estimate the false negative rate. As described above, previous papers from our group in which we used Sanger sequencing to validate candidate mutations have shown that this method performs very well.

d. Impact of filter cutoffs: The authors use specific cutoffs and thresholds for filters. Some of these filters have previously been shown to have a major impact on the estimated DNM rates, not just count (Besenbacher et al. 2019). Thus it would be useful if the authors can explore the impact of different threshold (for instance for depth, GQ etc.) and present results for other thresholds in the supplement.

Response: Although we agree with the reviewer that choosing the filter thresholds is a key step of the pedigree analyses, we recently published a paper (DOI: [10.1093/gigascience/giab029](https://doi.org/10.1093/gigascience/giab029)) showing that the impact of this on the estimated rates is not major. In this recent study, we show that the two filters with the largest effect on the inferred mutation rate were the genotype quality and allelic balance. Taking the reviewer suggestion into account, we did apply different GQ filters to some species of our dataset to evaluate the variation of the number of true positives, false positives, false negative rate, callable genome and mutation rates. We observed that for 6 mammals, 2 birds, and a reptile, the final rates do not change appreciably depending on these filters. For the fish species, we found some

variation in mutation rate estimates that was mainly caused by the large number of mutations observed with $GQ < 60$. However, as the number of false positive calls is high before this threshold, we restricted our analyses to candidates with $GQ > 60$. We have added this information to the Supplementary Fig. 10.

e. Association of genome quality and DNM rate: Given the genome quality across vertebrates varies vastly, it would be useful to see some results to test if the genome quality (N50 scaffold size, or other metrics) impacts the estimated rate.

Response: We follow the reviewer's suggestion and found no significant effect of N50, the percentage of Ns in the genome, the number of gaps, or the average scaffold size on the estimated rate. However, the percentage of Ns and the number of gaps in the genome impacted both the false positive rate and the percentage of the genome that we could call. Moreover, mapping quality did not impact the final rate, but it did impact the percentage of the callable genome and the false positive rate. Trios with a lower proportion of reads correctly mapped or a higher proportion of reads removed due to multiple hits, tended to have a lower CG and a higher false positive rate. We have now added these results in Supplementary Fig. 11 (for mapping quality) and Supplementary Fig. 12 (for reference genome quality).

f. Post-mapping QC: It would be useful if the authors can include a table with quality metrics for their dataset, including post-mapping coverage, ti/tv ratio, % CpG transitions, etc. The authors mention this in the main text but I couldn't find a table with the details in the supplement.

Response: We agree with the reviewer that these metrics were missing from the supplementary materials and we have now added two supplementary tables. Supplementary Table 6 contains different quality thresholds evaluated along the bioinformatic pipeline. Thus, for each sequence, we report information on quality control, trimming (e.g. number of N-bases, number of reads removed due to adaptor sequences or low-quality sequences), mapping success (e.g. the number of reads correctly mapped), post mapping (e.g. the number of duplicate reads, the number of final reads used, and coverage information in the final bam files). Supplementary Table 3 contains information on the spectrum of mutation for each species, the percentage of CpG mutations, and the ti/tv ratio.

3. UCE phylogeny: Comparing mutation rates with phylogenetic estimates is a good idea. However, I am surprised why the authors use UCE to compare the rates. The authors should be using putatively neutral sites for the comparison vs. UCE which is under strong constraint across species and an unreliable proxy for neutral substitution rates. The authors can use published phylogenies from UCSC browser for this analysis or perform the phylogenetic analysis for regions away from genes or ancestral repeats to get a more reliable inference. Moreover, it would be useful if the authors can show that the estimated rates are not sensitive to the assumed divergence times which are tentative and likely disagree with the mutation rates inferred in this study. Finally, some quantitative assessment of phylogenetic and pedigree rates would be useful (more on this #6).

Response: We agree that neutral sites are what we are looking for to compare the phylogenetic and pedigree rates. However, those sites are not easy to identify for such distant species. We choose UCEs to have as many homologous regions as possible. Following the reviewer's suggestion, we now used a second method based on MultiZ alignments of all of our species to the human genome. We estimated new substitution rates with this method and compared them to those obtained from the UCE alignment. We found that the two methods produced strongly correlated substitution rates on the terminal branches, especially for mammals (all species: $r^2 = 0.86$, $p < 2.2 \times 10^{-16}$; mammals: $r^2 = 0.99$, $p < 2.2 \times 10^{-16}$) and reported these results in Supplementary Fig. 14.

Regarding the second point, we compared the estimated substitution rates when using 14 initial calibration points or only 13 calibration points (with 14 iterations to remove each calibration node one by one). In general, there was a strong correlation between the rate estimated with 14 or 13 calibrations ($r^2 = 0.953$, $p < 2.2 \times 10^{-16}$). However, we found that some of the calibration nodes, especially on recent nodes, influence the estimated rates. We have now added these in Supplementary Fig. 15.

4. Effect of domestication: These conclusions seem speculative. The authors compare diverse species that vary in a range of criteria and suggest that the differences observed are related to domestication. Instead, the authors should perform comparisons of closely related species—for example, wolf and dog, and then see if after accounting for the within species differences, there are significant differences across species that could be associated to domestication. It is possible there are very few such comparisons available, in which case result should be removed or described as a tentative association.

Response: We agree that this section was weak compared to other results of our manuscript. We have now re-written this section as we found that using the wild generation time in our model to estimate germline mutation rate at generation time does not produce significantly higher rates for the domesticated species. Therefore, we conclude that the high rates observed for domesticated species are likely to be associated with lower ages of reproduction than occur in the wild.

5. Comparison with previous studies

A number of studies have previously reported pedigree-based mutation rates. It would be useful if the authors compare those findings with rates/ patterns inferred here.

a. Estimates of previously published mutation rates: For over a dozen species, pedigree mutation rates are available from previous analysis often with larger number of trios per species. It would be useful if the authors can show how their results compare with previous reports, especially after accounting for parental age.

Response: We did not include the species with available estimates in the literature as those were not sequenced to the same very high depth and were not estimated with the same method. However, we agree that a comparison of rates could be useful, and have now added Supplementary Fig. 6 showing our rates per generation compared to ones previously estimated in the literature.

b. Estimates of male bias: A recent study (Wu et al. 2020) showed that across mammals the male bias in mutation rate is ~3-4:1 across a range of mammals. This seems consistent with the authors results, though they find a non-negative slope across vertebrates. It would be useful if the authors can explicitly discuss the differences in the observations in the two studies and discuss how this might relate to the relative contributions of DNA damage and repair in contributing to new DNMs.

Response: In Wu et al. 2020, the authors obtained the DNMs data produced by previous studies which used different sequencing strategies and DNM calling pipelines and did not find an increased number of DNMs with paternal age. It is hard to assess this result because of potential artifacts caused by methodological differences. Our previous study (DOI: 10.1093/gigascience/giab029) suggests different methods can detect different DNMs, resulting in a two-fold difference in DNM rate estimation. Our study overcame this issue, and supports a strong contribution of parental age across all species.

7. Comparison of micro and macro evolution: It would be useful if for pairs of species, the authors can compare how the yearly mutation rate compares with heterozygosity and pi and phylogenetic rates. The authors compare broad scale patterns but would be useful to provide more detailed comparison.

For example, the authors suggest their results are consistent with hominoid rate slowdown hypothesis. However, given the large variation in the rates within species groups, the signal is not obvious. The rates in pedigrees of OWM in this study are puzzlingly lower than previous estimates, including from another study led by the same authors. The ratio of mutation rates across primates are -

Chimp/H = 1.59

Gibbon/ H = 1.97

OWM/H = 0.83-1.43

NWM/H = 1.55-3.53

More importantly, the quantitative estimates from phylogenetic analysis suggest much lower differentiation in substitution rates across these species pairs, see Moorjani et al. PNAS 2016, Kim et al. 2009. While selection is a possibility, the authors should include quantitative comparisons of the phylogenetic and pedigree rates across species and reassess if the patterns agree / disagree with previous model predictions.

Response: We have now removed this part on micro to macroevolution continuity as it appears confusing and decided to focus on the strongest results of our manuscript. Yet, we agree with the reviewer that our estimate for the Old World Monkey the Drill is a little on the low side compared to previous estimates for this group. However, our small sample size for this species does not allow to us to reach conclusions; this is why we have focused on more general patterns across species.

Minor comments:

1. Heterozygosity: The authors infer heterozygosity for all individuals in the trio, which is likely to be underestimated as related individuals are included. The authors should exclude the offspring and rerun this analysis only for the parents. Also removing closely related parents is advisable (which can be assessed by estimating the %IBD shared across the two parental genomes).

Response: Heterozygosity calculated from unrelated individuals and from all individuals was highly correlated ($r^2 = 0.99$, $p < 2 \times 10^{-16}$). We agree that the solution proposed by the reviewer is more meaningful, but we have now removed this part of our manuscript (see answer above).

2. Drift-barrier hypothesis: In order to investigate this, the authors compare current mutation rate with historical N_e . This choice is odd as I don't see how historical effective population size should have any impact on the current mutation rate. Moreover, N_e is correlated to a number of LH and so the authors should examine if the effects are driven by N_e or other factors. Perhaps using the PGLS or PPA approach.

Response: In the revision, we used both historical N_e inferred with PSMC and the recent N_e using $N_e = \pi/4\mu$, with π the nucleotide diversity and μ the mutation rate per site per generation estimated from the UCEs. Both results suggested a weak correlation between N_e and per generation rate. The reviewer is right that N_e is correlated with other life-historical traits as revealed by the PPA study. We only observed a weak correlation between N_e and per generation rate, but the N_e is also correlated with generation time which is a stronger correlation factor with generation rate.

3. Estimation of yearly mutation rate: The authors use a “weight of paternal contribution” – Is this the alpha or male bias estimated using phased DNMs? Please clarify?

Response: Indeed, this is the α , which we now clarify this in the manuscript (page 16 line 454). We now use a modeling method to estimate the yearly rate at generation time per species.

4. It would be useful if the authors can discuss a limitation of their study that it is based on samples from zoos and animal centers, which may have different life history traits than the wild. For many analysis, they interchangeably use the life history parameters of their sample or in the wild, current or historical estimates. These approximations maybe necessary but could lead to a bias the associations.

Response: We now added a sentence to discuss the limitation of our study in the conclusion (page 14 lines 376 - 380).

5. The following sentence should be revised to reflect that the rates in human trios differ by 40% or removed.

“Note that our estimate for humans is consistent with the previous estimates based on larger pedigrees at $\mu_{\text{yearly}} \sim 0.42 \times 10^{-9}$ mutations per site per year 25,72.”

Response: We removed this sentence.

6. Supplementary table with all DNM information should be provided.

For each species, please report

<species> <genomic location> <maternal>

These results will be helpful resource for others interested in investigating mutation rates and patterns in future studies.

Response: We agree that this is useful information and have now added Supplementary Table 7 with the location of all DNM candidates.

8. Rates for X and Y chromosome: It is unclear why most DNM studies leave out sex chromosomes. It would be useful if the authors can provide DNM rates for X and Y chromosomes as these can be very useful for understanding sex-bias evolutionary patterns.

Response: We agree this is an interesting topic to be explored in DNM rates. However, it is hard to infer rates on the X and Y chromosomes. First, they are small regions compared to the whole genome, and very rare DNMs are usually located on these two chromosomes. Second, sex-linked annotation information was missing in many genomes.

Reviewer Reports on the First Revision:

Referees' comments:

Referee #1 (Remarks to the Author):

The results reported in this paper remain a tour de force, which will certainly stimulate a lot of downstream thought and further research. In their revision, the authors had a quite substantial burden, given the four lengthy reviews provided. In general, I think they have done a fairly good job, but there are places in which it seems that the authors have pushed a bit too hard in trying to come with explanations for the patterns of data. Owing to the still limited data, some of the rate estimates have quite large standard errors, and an alternative view is that things like life-history traits explain only a small fraction of the variation (although we don't know how much this would improve were the estimates to be more refined).

Line 96 – 98. Need to define CI – presumably this is of the overall group means, but we'd like some idea of the range of variation among species within groups, which the CI provides no information on. For example, the well-established rate for humans is outside the CI range for mammals. Insight might be gained by a simple analysis of the among-species and error components of variance, to yield a clean estimate of the former?

In addition, there seems to be an error here – the rate for reptiles is the highest, not the lowest.

Lines 236-237 – The conclusion that the rate at birth is nearly constant across species is a bit of an overstatement, as it is inconsistent with the data, which show a nearly 10-fold range of “time-zero” variation (admittedly, much of this is due to sampling error). In any event, this seems to be another example of drawing too strong a conclusion from the data. I have no problem at all with the authors being a bit more realistic of the limited reach of the data, and indeed this would be a useful cautionary note given the massive nature of this study.

Line 270 – define UCE; most readers will have no idea. Given that these are ultraconserved elements, why would we expect them to reflect mutation rates at all? One might argue that these are the least useful sequences in this context. Again, wouldn't silent-site divergence be more useful here, at least for species pairs where things aren't saturated?

Figure 2 – Related to the preceding point, the correspondence between mutation rates and substitution rates are not very impressive, explaining just 20% of the variation. Most significantly, it seems to be driven by a single data point from fish, so one is left wondering about the utility of this overall analysis.

Likewise, although nice to have included, the life-history results are not very impressive in terms of explanatory power. For example, Line 295 – explaining 18% of the variance in rates is not a sign of terribly successful model. This is not the fault of the authors, but one again gets impression that there is a tendency for the authors to read the cup as being more full than empty.

Figure 3e. Along these same lines, the structure of the path analysis is surprising, given that the general intention of path analysis is to at least attempt to infer causality. I would think that N_e is an emergent property of life-history traits, not the other way around. Did the authors check this? In addition, it would be useful to know how much of the total variation in u is explained by the path analysis. My overall take on this is that very little of mutation-rate variation is explained by life-history variation.

Lines 300 – 313. There seems to be confusion here on the matter of N_e estimation. A number of us have multiple concerns with the PSMC approach, but I am also confused by why the authors used the substitution rate, when it is clear that this is only vaguely related to their direct mutation-rate estimates and also is subject to errors associated with fossil dating. Moreover, the comments on the timescale over which the PSMC and the π approach measure N_e does not seem correct; the latter approach itself cannot extend back to more than $4N_e$ or so generations. Moreover, it is unclear why the authors think they need a measure of ancient N_e , given that the mutation rate is a highly malleable trait evolutionarily. I would think that the more recent the N_e estimate, the more informative, and I'm not at all convinced that N_e over the past million years is what is needed here.

Domestication effects. This effect too seems quite weak and too uncertain to be included.

In summary, this remains an impressive piece of work, but some of the interpretations do seem questionable and/or overly speculative. At least part of the problem may be that the authors were attempting to mutually satisfy a diverse set of comments from so many reviewers. Ultimately, the handling editor will need to make a decision on what should be included or not.

Referee #3 (Remarks to the Author):

I have re-read this new version of the manuscript and feel that it has been greatly improved. The presentation of the data, the argument and figures are now much more focused and provide a compelling overview of the analysis of a large dataset.

I have no further comments and I am satisfied that the authors have adequately addressed my previous queries. This is an excellent piece of work which will be extremely valuable to the wider scientific community.

Referee #4 (Remarks to the Author):

The revised manuscript is much improved. However, there are still some minor issues remaining that need to be resolved.

1. New model to estimate generation time.

I carefully reviewed the supplementary Note 1 for the new model. There are several assumptions the authors make that need to be explicitly stated and justified. Some assumptions would actually be

useful to test with the author's data (as described below).

a. The proposed model combines replication-driven and non-replicative mutations; both would have different dependences on G. Both be very useful if the authors can incorporate these differences more explicitly.

b. The authors assume the intercept (a) and amplitude (b) of the linear model should be positive – though the intercept may actually be meaningful to understand how mutation rate depends on G (Gao et al. Plos Biology). It would be useful if the authors can relax this assumption to learn a instead of assuming $a > 0$. The reason this is critical is that their results that G is the strongest predictor could just be an artifact induced by the model.

c. Similar intercept across species. Can the authors justify this assumption?

2. Estimation of FNR and FPR

I am still not convinced that FPR and FNR are well characterized in the author's estimates, especially since the mutation estimates in humans and primates disagree with published literature.

- FPR – the dependence on genome quality is a bit concerning. The estimated rate should be adjusted to account for this effect.

- FNR – A bigger is that FNR most likely also depends on genome quality. I am not sure if I understand the author's argument that methods like bamurgeon will not work or that there is no standard pipeline will not work. The callable genome does not need to be re-estimated as one only introduces a small number of DNMs (say, 100-10,000) and so this will only have a tiny, unnoticeable impact on the callable genome. While I agree it will be a lot of work to repeat this on all species, it will be useful to include the results for two species with the most extreme genome quality (best and worst) so one has an idea of the range of FNR to expect and how it varies by genome quality.

More importantly, how much of the variation is due to these systematic effects vs. random errors assumed by authors in computing the SE/ CI.

3. Variation in alpha with G.

It would be useful if the authors can comment on the consistency of their results with published results/ models by other groups like Wu et al. 2020 PLoS biology and Wu et al. 2022 Biorxiv. Also can the authors comment on how the alpha in pedigrees compares with alpha estimated from substitution rates (Wu et al. 2022 Biorxiv or Sayres et al. 2011). I agree with the authors that this is the first direct estimate to assess the validity of the models proposed previously and so in that regard, it would be useful to include this discussion- even if the results differ.

4. Comparison of Ne with UCE mutation rate

I am still not convinced that the authors should use UCE mutation rate since it is not a proxy of neutral substitution rate. And even the use of genome-wide Multiz rates is not applicable, since there is no attempt to control for effects of selection. It would be useful to explicitly point out these assumptions so the reader can appreciate that the UCE rate should not be confused with neutral substitution rates.

5. Minor points

a. Figure 1 – why are the CI intervals asymmetric?

b. Mutation spectrum (Cpg%, ti/tv, etc) – Please add CI – based on bootstrap resampling or other

approaches.

c. For the following result, it would be useful to add some additional details here based on phased DNMs. What fraction of shared DNMs can be phased? Is there any bias in the parent of origin?

“An explanation for the repeated occurrence of those mutations is that they appear during the primordial germ cell specification (PGCS) in one of the parents. The occurrence of PGCS mutations is independent of parental sex. Consequently, a higher number of PGCS mutations in some vertebrate groups could be an alternative explanation for the lower male-biased contribution to DNMs.”

d. Abstract – Was the abstract updated to reflect the current scope of the study?

Author Rebuttals to First Revision:

Referees' comments:

Referee #1 (Remarks to the Author):

The results reported in this paper remain a tour de force, which will certainly stimulate a lot of downstream thought and further research. In their revision, the authors had a quite substantial burden, given the four lengthy reviews provided. In general, I think they have done a fairly good job, but there are places in which it seems that the authors have pushed a bit too hard in trying to come with explanations for the patterns of data. Owing to the still limited data, some of the rate estimates have quite large standard errors, and an alternative view is that things like life-history traits explain only a small fraction of the variation (although we don't know how much this would improve were the estimates to be more refined).

Response: We thank the reviewer for the positive comments and have strived (see below) to revise and tone down our interpretations to more carefully acknowledge the uncertainty of individual estimates and alternative explanations of our findings.

Line 96 – 98. Need to define CI – presumably this is of the overall group means, but we'd like some idea of the range of variation among species within groups, which the CI provides no information on. For example, the well-established rate for humans is outside the CI range for mammals. Insight might be gained by a simple analysis of the among-species and error components of variance, to yield a clean estimate of the former?

Response: We thank the reviewer for this useful comment. A definition of CI was missing. We have now defined the mean and associated CI. We also agree with the criticism that the mammalian mean rate is lacking information on variation within the group. We have now changed the text accordingly and have added a sentence that focuses on the variation within major groups as shown in Figure 1.

Lines 100 - 103: "...birds (mean of all trios of the group: 8.08×10^{-9} , 95% CI of the mean = 6.23×10^{-9} , 9.93×10^{-9}). Furthermore, the amount of variation in $\mu_{\text{generation}}$ among species is overall higher for birds and lower for mammals and fishes (Fig. 1a), although this variation is arguably modest given large differences in life-history traits among these species..."

In addition, there seems to be an error here – the rate for reptiles is the highest, not the lowest.

Response: We thank the reviewer for noticing this mistake and we have now corrected this accordingly.

Lines 95 - 101: "...with the highest mean estimates obtained for reptiles (mean of all trios of the group: 1.04×10^{-8} , 95% CI of the mean = 6.49×10^{-9} , 1.42×10^{-8}), intermediate values

obtained for mammals (mean of all trios of the group: 7.77×10^{-9} , 95% CI of the mean = 7.11×10^{-9} , 8.42×10^{-9}) and fish (mean of all trios if the group 6.31×10^{-9} , 95% CI of the mean = 4.78×10^{-9} , 7.83×10^{-9}) and the lowest estimates obtained for birds (mean of all trios of the group: 8.08×10^{-9} , 95% CI of the mean = 6.23×10^{-9} , 9.93×10^{-9}).

Lines 236-237 – The conclusion that the rate at birth is nearly constant across species is a bit of an overstatement, as it is inconsistent with the data, which show a nearly 10-fold range of “time-zero” variation (admittedly, much of this is due to sampling error). In any event, this seems to be another example of drawing too strong a conclusion from the data. I have no problem at all with the authors being a bit more realistic of the limited reach of the data, and indeed this would be a useful cautionary note given the massive nature of this study.

Response: We agree with the reviewer that our conclusion was too strong, which was due to our initial surprise that there were no strong differences. This purported constancy of the rate at birth is the null model, which we cannot reject based on our data and previous studies. However, our data have limited power in distinguishing smaller differences and we now investigate this in the supplemental note in more detail and have aimed to make this clearer in the main text.

Removing lines 244: “This good fit of a general model leads us to conclude that the number of mutations at birth is remarkably constant among taxonomic groups and makes up a large proportion of the mutations in species with short generation times.”

Line 270 – define UCE; most readers will have no idea. Given that these are ultraconserved elements, why would we expect them to reflect mutation rates at all? One might argue that these are the least useful sequences in this context. Again, wouldn’t silent-site divergence be more useful here, at least for species pairs where things aren’t saturated?

Response: We have now defined UCEs and added some justification of why and how we used them. Indeed, it seems that those justifications were not clear enough in the previous version and may have given a false impression that we only used the 2,800 UCE sequences. However, we also analysed the flanking regions (1,000 bp on each side of the UCE probes) where there is no evidence of strong conservation and hence they may provide a better proxy of the neutral rate. The flanking regions make up 94% of the total UCE data and are therefore expected to dominate the estimates (also because they are indeed more variable). Therefore, the substitution rate estimated with this method is closer to the neutral substitution rate than if had we only used the UCEs.

Lines 277 - 280: “To obtain an estimate of the long-term substitution rate, we used the alignment of Ultraconserved Elements (UCEs), which are more likely to align among taxonomically distant species, plus 1,000 bp of flanking regions on each side of the UCE sequences, which will more closely reflect the neutral substitution rate⁵⁵.”

We have also implemented a whole genome alignment across all vertebrate species using MultiZ, yet, as we expected, only a small portion of the genome aligned with 1,345 regions of at least 1,000 bp, resulting in a minimum alignment of 1,345,000 bp, which is less than the alignment of UCEs and their flanking regions. However, the MultiZ alignment produced very similar substitution rates to the UCEs (presented in supplementary Fig. 14). We also added these results to the main text and Fig. 2. We furthermore estimated a substitution rate using 4D sites of 163 orthologous genes (5,919 bp). The substitution rates estimated with 4D were not significantly correlated with those based on UCEs or MultiZ. As pointed out by the reviewer, this is likely due to the saturation of the silent sites over a long period of evolution.

Lines 285 - 287: “We also found a significant correlation between $\mu_{\text{yearly_modeled}}$ and the long-term substitution rate inferred using whole genome alignments (Fig. 2b).”

Figure 2 – Related to the preceding point, the correspondence between mutation rates and substitution rates are not very impressive, explaining just 20% of the variation. Most significantly, it seems to be driven by a single data point from fish, so one is left wondering about the utility of this overall analysis.

Response: This figure has now been changed, with the reptile species having a higher mutation rate. However, the overall pattern stays the same, revealing a positive association between the pedigree-based mutation rate and the long-term substitution rate. We have now added a comparison with the whole genome alignment method to estimate the substitution rates, which produce very similar results. We also emphasized that these results are particularly strong for the mammalian species, for which the variance explained is 44%. We have retained this analysis but reduced our emphasis on the results and toned down our conclusions accordingly.

Lines 283 – 287: “This pattern is especially pronounced for mammals (PGLS: adjusted $r^2 = 0.44$, $p = 0.0008$), even after removing the two outliers (PGLS: adjusted $r^2 = 0.32$, $p = 0.009$). We also found a significant correlation between $\mu_{\text{yearly_modeled}}$ and the long-term substitution rate inferred using whole genome alignments (Fig. 2b).”

Likewise, although nice to have included, the life-history results are not very impressive in terms of explanatory power. For example, Line 295 – explaining 18% of the variance in rates is not a sign of terribly successful model. This is not the fault of the authors, but one again gets impression that there is a tendency for the authors to read the cup as being more full than empty.

Response: While 20% of explained variation may not seem impressive to some, we would argue that it is not insignificant given that our study cannot account for all possible contributing sources of variation, and due to sampling error. Nevertheless, we have toned down our conclusions about the role of life history traits. In addition, we have also changed the title of the manuscript to increase the focus on our strongest results.

New title: “Evolution of the germline mutation rate across vertebrates”

Figure 3e. Along these same lines, the structure of the path analysis is surprising, given that the general intention of path analysis is to at least attempt to infer causality. I would think that N_e is an emergent property of life-history traits, not the other way around. Did the authors check this? In addition, it would be useful to know how much of the total variation in u is explained by the path analysis. My overall take on this is that very little of mutation-rate variation is explained by life-history variation.

Response: In the phylopath analysis, we tested different hypotheses, in which N_e was both an emergent property of life-history traits and the other way around. The most significant model was the one we presented. However, this analysis is dependent on the number of variables and hypotheses proposed. Therefore, we have now changed and simplified our analysis to focus on fewer variables (only the marginally significant life history traits), and simpler hypotheses (presented in supplementary Fig. 11). In this case, the causality of N_e was reversed. We have decided to present this analysis in the main manuscript (Fig. 3e).

Lines 300 – 313. There seems to be confusion here on the matter of N_e estimation. A number of us have multiple concerns with the PSMC approach, but I am also confused by why the authors used the substitution rate, when it is clear that this is only vaguely related to their direct mutation-rate estimates and also is subject to errors associated with fossil dating. Moreover, the comments on the timescale over which the PSMC and the π approach measure N_e does not seem correct; the latter approach itself cannot extend back to more than $4N_e$ or so generations. Moreover, it is unclear why the authors think they need a measure of ancient N_e , given that the mutation rate is a highly malleable trait evolutionarily. I would think that the more recent the N_e estimate, the more informative, and I'm not at all convinced that N_e over the past million years is what is needed here.

Response: We agree with the reviewer that estimating N_e is complex. Indeed, the time-scale of relevance depends both on the time scale at which we expect the mutation rate to evolve and where we have sufficient coalescence information to estimate the effective size with some precision.

First, regarding the rate used in the PSMC, we used the substitution rate instead of our pedigree-based rate to avoid circularity. Indeed, we would expect a stronger signal when looking at the correlation between N_e and the per generation rate if N_e was estimated with the pedigree-based rate. However, we agree that both approaches are informative and we have now added a supplementary figure with the correlation between N_e and the mutation rate when N_e is estimated with the pedigree-based mutation rate. As expected, the pedigree-based estimate produces a stronger correlation, but we caution that this could potentially be (at least partly) a consequence of the aforementioned circularity. We have now added this in the main text.

Second, the timescale we used was not reflecting a prior hypothesis on the timescale over which we expect the mutation rate to adapt to a change in N_e . We used the harmonic mean

between 30,000 years and 1,000,000 years because PSMC poorly estimates N_e during the past 30,000 years, where, in most species, there are few expected coalescence events. However, in response to the reviewer's comment, we also tried using the harmonic mean over a more recent period of time (between 30,000 and 130,000 years) and the negative correlation remained significant for mammals (PGLS: adjusted $r^2 = 0.104$, $p = 0.040$; as in the previous analysis PGLS for mammals: adjusted $r^2 = 0.305$, $p = 0.0006$) but not for all of the species together (PGLS: adjusted $r^2 = -0.02$, $p = 0.13$). Moreover, we obtained a stronger correlation between the N_e estimated based on nucleotide diversity and the harmonic mean of N_e over the past million years (correlation test: $r^2 = 0.91$ $p < 2.2e-16$), compared to the past thousand years (correlation test: $r^2 = 0.50$ $p = 2.7e-05$). Since we agree with the reviewer that there might not be an ideal way to estimate N_e , we have now added all of the new analyses as plots contained in a new supplementary figure (see below).

Modification in paragraph lines 314 - 340

Supplementary Fig. 9 – Comparison of different time and mutation rate parameters used to estimate N_e . **a.** The correlation between N_e and the mutation rate per generation is not significant when using the most recent value before 30,000 years estimated by PSMC. **b.** The correlation is neither significant with the harmonic mean over a more recent period of time (30,000 years to 130,000 years ago) over all species. Yet, this correlation is significant for mammals (adjusted $r^2 = 0.104$, $p = 0.04$). We used the harmonic mean over the past million years in the main text, as PSMC is not reliable over recent periods. **c.** When looking at the correlation between the mutation rate and N_e , estimated using the pedigree-based mutation rate, we found a stronger signal over the past 1,000,000 years, due to the circularity of this analysis. **d.** However, the correlation is still not significant when using the most recent time point or **e.** the average over the past 100,000 years.

Domestication effects. This effect too seems quite weak and too uncertain to be included.

Response: We have now changed the title of this section in the manuscript to make more clear what we believe this analysis shows, which is that yearly rates in domesticated species are much higher than in comparable non-domesticated species. We cannot demonstrate with the available data that there is an inherent change in the mutational process in domesticated species, as we might expect if domestication were associated with relaxed purifying selection. However, we can show that the shorter generation times of domesticated species due to domestication practices have a large influence on yearly mutation rates. We believe this is an important and interesting observation that could interest many readers.

Line 366: “Higher yearly mutation rates of domesticated animals due to reduced generation times”

In summary, this remains an impressive piece of work, but some of the interpretations do seem questionable and/or overly speculative. At least part of the problem may be that the authors were attempting to mutually satisfy a diverse set of comments from so many reviewers. Ultimately, the handling editor will need to make a decision on what should be included or not.

Response: We thank the reviewer for the helpful comments. It has indeed been challenging to incorporate so many diverse comments, but we believe that our manuscript is now much stronger.

Referee #3 (Remarks to the Author):

I have re-read this new version of the manuscript and feel that it has been greatly improved. The presentation of the data, the argument and figures are now much more focused and provide a compelling overview of the analysis of a large dataset.

I have no further comments and I am satisfied that the authors have adequately addressed my previous queries. This is an excellent piece of work which will be extremely valuable to the wider scientific community.

Response: We thank the reviewer for their positive appraisal of our work.

Referee #4 (Remarks to the Author):

The revised manuscript is much improved. However, there are still some minor issues remaining that need to be resolved.

Response: We thank the reviewer for this comment.

1. New model to estimate generation time.

I carefully reviewed the supplementary Note 1 for the new model. There are several assumptions the authors make that need to be explicitly stated and justified. Some assumptions would actually be useful to test with the author's data (as described below).

a. The proposed model combines replication-driven and non-replicative mutations; both would have different dependences on G . Both be very useful if the authors can incorporate these differences more explicitly.

Response: We agree with the reviewer that time dependencies could likely be different from replicative and non-replicative mutations and it would be interesting to distinguish them in an analysis. However, at the current time, very little is known about the differences in the type and context of these two classes of mutations, even in well-studied species such as humans and mice. Consequently, we are not yet in a position to be able to meaningfully implement such an analysis, although this would certainly be an interesting line of enquiry for future studies using even larger datasets.

b. The authors assume the intercept (a) and amplitude (b) of the linear model should be positive – though the intercept may actually be meaningful to understand how mutation rate depends on G (Gao et al. Plos Biology). It would be useful if the authors can relax this assumption to learn a instead of assuming $a > 0$. The reason this is critical is that their results that G is the strongest predictor could just be an artifact induced by the model.

Response: To address the assumptions and choices that we made in the new model, we have added a section called “Validating model choice” to the supplement describing the model. In the subsection called “Checking assumptions on parameters”, we test a model that does not assume that the intercept is positive. The results show that we still get a positive intercept even if we remove that assumption.

Supplementary note 1

c. Similar intercept across species. Can the authors justify this assumption?

Response: We have also added a subsection called “model selection” to the supplementary note describing the model. This subsection compares the used model to models that have multiple intercepts or multiple slopes. The results show that we obtain similar intercepts and slopes across phylogenetic groups, while the slight increase in model fit for the more complex models does not justify the added number of parameters included.

Supplementary note 1

2. Estimation of FNR and FPR

I am still not convinced that FPR and FNR are well characterized in the author's estimates, especially since the mutation estimates in humans and primates disagree with published literature.

Response: We agree that the proper FPR and FNR are important, not least if these are systematically biased by some external factor such as genome quality or heterozygosity/divergence. However, we do not agree that our estimates are in conflict with previously published estimates. For example, for humans, we found a per generation rate of 1.17×10^{-8} mutation per site per generation (95% CI 0.95×10^{-8} , 1.41×10^{-8}) which is very close to the estimate of 1.2×10^{-8} mutation per site per generation generally accepted for humans. Moreover, we added a supplementary figure 6 showing that most previous pedigree-based estimations of GMR fall within the confidence interval that we estimated for the same (or closely related) species, especially for the primates. To make these results more clear, we have now also added this in a supplementary Table 9.

- FPR – the dependence on genome quality is a bit concerning. The estimated rate should be adjusted to account for this effect.

Response: We believe that the rate does account for this effect. Indeed, the estimate from a poor genome quality will lead to a higher number of candidates, among which a higher number of false positive calls are removed, yet the final estimated rate will be also divided by a smaller genome size. This is why the estimated rates are not dependent on genome quality (see supplementary Fig. 14). To clarify this point, we have now added a sentence in the Methods section.

Lines 462 – 465: “Additionally, we showed that sample type, reference genome quality, and mapping quality can affect the results on the number of candidates, the FPR and FNR, yet, the estimated mutation rates are not affected (supplementary Figs. 13, 14, and 15).”

- FNR – A bigger is that FNR most likely also depends on genome quality. I am not sure if I understand the author's argument that methods like Bamsurgeon will not work or that there is no standard pipeline will not work. The callable genome does not need to be re-estimated as one only introduces a small number of DNMs (say, 100-10,000) and so this will only have a tiny, unnoticeable impact on the callable genome. While I agree it will be a lot of work to repeat this on all species, it will be useful to include the results for two species with the most extreme genome quality (best and worst) so one has an idea of the range of FNR to expect and how it varies by genome quality.

Response: We agree with the reviewer that it will not change the callable genome size to introduce 100 simulated mutations, yet it would change the coverage and genome quality in the regions of the simulated mutations. For instance, a mutation simulated with Bamsurgeon on a site that we know is callable (because it has sufficient coverage and quality), could be missed by the pipeline, and then declared as a false negative. Yet, if by estimating a new callable genome we found that this position is not callable any longer (because the introduced mutation has reduced the genotype quality below 60 for instance) then this mutation should not be counted as a false negative call, as this position will no longer be in the callable genome. This is why we believe that only simulating mutations is not an appropriate method to estimate false negative rates here. We have now added a sentence in the Methods section to justify our choice.

Lines 472 – 474: “Two methods are used in the literature to estimate FNR; one is the simulation of mutations and the other is a correction on the filters that are not accounted for in the callable genome. As in our previously study of GMR¹⁰, we used the latter method, which is more conservative.”

More importantly, how much of the variation is due to these systematic effects vs. random errors assumed by authors in computing the SE/ CI.

Response: We believe that obtaining an idea of the systematic effect of reference genome quality, and the uncertainty of each filter applied, along with the one from the FPR and FNR correction, would require an intensive effort on a larger dataset. We agree that this is an interesting question, but we believe this is beyond the scope of our study.

3. Variation in alpha with G.

It would be useful if the authors can comment on the consistency of their results with published results/ models by other groups like Wu et al. 2020 PLoS biology and Wu et al. 2022 Biorxiv. Also can the authors comment on how the alpha in pedigrees compares with alpha estimated from substitution rates (Wu et al. 2022 Biorxiv or Sayres et al. 2011). I agree with the authors that this is the first direct estimate to assess the validity of the models proposed previously and so in that regard, it would be useful to include this discussion- even if the results differ.

Response: We agree that this was missing from our paper. We have now corrected this and compared our results with the two main studies suggested by the reviewers. We have added a sentence in the main text, as well as both a supplementary figure (supplementary Fig. 7, also below) and a table (supplementary Table 5) showing that the alpha and CI of the common species between our dataset and the two studies mentioned by the reviewer are in broad agreement.

Lines 177 -178: “In general our α estimates are in line with previous estimates derived for similar species based on genome alignments^{28,29}”

Supplementary Fig. 7 – Comparison of published male bias estimates (α) using genome alignments and our male bias estimates (modified Fig. 1c of the main text).

The yellow dots are α estimates from Wilson Sayres et al., 2011, and the purple dots are α estimates from Wu et al., 2022. Most of the common species reveal similar estimates with overlapping 95% confidence intervals. However, the estimated α based on genome alignments are generally lower for dogs and cats than our estimates, yet the pedigree-based estimate of α for cats (Wang et al., 2022; green point) is similar to our estimate. See also supplementary Table 5.

4. Comparison of Ne with UCE mutation rate

I am still not convinced that the authors should use UCE mutation rate since it is not a proxy of neutral substitution rate. And even the use of genome-wide Multiz rates is not applicable, since there is no attempt to control for effects of selection. It would be useful to explicitly point out these assumptions so the reader can appreciate that the UCE rate should not be confused with neutral substitution rates.

Response: We have now extended this section on the justification of using UCEs. Indeed, it seems that those justifications were not clear enough in the previous version and misled the readers that we only used the 2,800 UCEs sequences, whereas in fact we also included some flanking “neutral” regions (of 1,000 bp on each side of the UCE probes, which represented 94% of the total data). Therefore, the substitution rate estimated with this method will be closer to the neutral substitution rate. We have also updated this part of the manuscript with a comparison of the multiz data and reduced our claim as this association appears to be mainly driven by the mammalian species.

Lines 277 – 280: “To obtain an estimate of the long-term substitution rate, we used the alignment of Ultraconserved Elements (UCEs), which are more likely to align among taxonomically distant species, plus 1,000 bp of flanking regions on each side of the UCE sequences, which will more closely reflect the neutral substitution rate⁵⁵.”

5. Minor points

a. Figure 1 – why are the CI intervals asymmetric?

Response: Those intervals are computed with `binconf()` function in R. This is the binomial confidence interval based on Wilson scores, which are asymmetric. We have now added this information to the methods.

Lines 482 - 483: “We estimated the 95% binomial confidence interval per species using the `binconf()` function in R, with the default Wilson score.”

b. Mutation spectrum (CpG%, ti/tv, etc) – Please add CI – based on bootstrap resampling or other approaches.

Response: We have now added a 95% confidence interval for CpG%, ti/tv and the direction of mutation based on the binomial distribution.

Lines 123 - 128: “... including a ratio of transitions over transversions (ti/tv) of 2.3 (95% CI on binomial distribution = 2.2, 2.5) and a high proportion (48.5%, 95% CI on binomial distribution = 46.7%, 50.3%) of transitions from strong base pairing to weak base pairing (C:G > T:A) across all DNMs (Supplementary Table 3). Among C:G > T:A mutations, 42.4% (95% CI on binomial distribution = 39.9%, 45.0%) occurred at CpG sites.”

c. For the following result, it would be useful to add some additional details here based on phased DNMs. What fraction of shared DNMs can be phased? Is there any bias in the parent of origin?

“An explanation for the repeated occurrence of those mutations is that they appear during the primordial germ cell specification (PGCS) in one of the parents. The occurrence of PGCS mutations is independent of parental sex. Consequently, a higher number of PGCS mutations in some vertebrate groups could be an alternative explanation for the lower male-biased contribution to DNMs.”

Response: We agree that these are interesting data that were missing from the previous version of our manuscript. Indeed, of the 26 positions shared by more than 2 individuals, 12 could be phased and we found 6 maternal and 6 paternal mutations. Eleven positions were not phased and 3 positions were not correctly phased as they were paternal for one individual and maternal for the other. Nevertheless, we believe that overall we have too few data to draw robust and generalizable conclusions. We have therefore only added this information to supplementary table 4 on the shared mutations.

d. Abstract – Was the abstract updated to reflect the current scope of the study?

Response: No, but we have now updated the abstract accordingly. Specifically, we have removed the sentence on yearly rates and long term substitution rates and refocused the abstract on the effect of generation time on per generation rate and the drift barrier hypothesis to reflect both reviewers' suggestions.

Reviewer Reports on the Second Revision:

Referees' comments:

Referee #1 (Remarks to the Author):

For the most part, the authors have done a good job in responding to the reviewer comments. This is a complex study though, and seemingly each reading uncovers new statistical issues that need clarification. My apologies for this, but it is likely that readers will have similar concerns, so I am hopeful that in the final revision, the authors can simply address the issues to the editor's satisfaction.

Specific comments:

Yellow in Figure 1 is very hard to see.

Lines 96-101 – the differences between groups here seem not to be impressive, as the CIs between the upper and lower groups overlap. Also, the means are said to be based on “all trios,” whereas some species results are based on more trios than others. A more reasonable estimate of the group average would seem to be based on the species averages, so as not to give artificially high weighting to single species (indeed, the average could also be weighted by the species-specific inverses of the sampling variance (the optimal weight typically used in these sorts of averaging)).

Lines 267-269 – would be good to give CIs on these, as the bird estimates are extraordinarily low.

Line 283 – define PGLS

Figure 2 – the two axes should be given the same scale, and the null line of equality should be added to show if the observed rates are in fact quantitatively consistent with the estimated substitution rates, or systematically under- or overestimate them. It looks like the majority of points will be below the line, which would mean that the genomic data are somehow overestimating the mutation rate. Given the dispersion of the data, the plots (and the significance tests) might also be more meaningful on a log scale?

Lines 302-306 – The authors need to state why the correlations here are not statistical artifacts. The issue is that generation time is seemingly used to estimate the mutation rate that is applied, introducing potential circularity (and also that maturation time must be correlated with generation time, making these two analyses nonindependent). This becomes relevant in the next section as well.

Figure 3 – axes labels need time units (presumably years).

Typos – lines 79, 401, and probably more that need to be tidied up.

Referee #4 (Remarks to the Author):

n/a

Author Rebuttals to Second Revision:

Referees' comments:

Referee #1 (Remarks to the Author):

For the most part, the authors have done a good job in responding to the reviewer comments. This is a complex study though, and seemingly each reading uncovers new statistical issues that need clarification. My apologies for this, but it is likely that readers will have similar concerns, so I am hopeful that in the final revision, the authors can simply address the issues to the editor's satisfaction.

Response: We thank the reviewer for again carefully reading our manuscript and providing further useful comments.

Specific comments:

Yellow in Figure 1 is very hard to see.

Response: We have now changed the yellow to a darker version of yellow in all of the figures for ease of reading.

Lines 96-101 – the differences between groups here seem not to be impressive, as the CIs between the upper and lower groups overlap. Also, the means are said to be based on “all trios,” whereas some species results are based on more trios than others. A more reasonable estimate of the group average would seem to be based on the species averages, so as not to give artificially high weighting to single species (indeed, the average could also be weighted by the species-specific inverses of the sampling variance (the optimal weight typically used in these sorts of averaging)).

Response: We agree with the reviewer that the difference between groups is not impressive. Therefore, we have now changed the corresponding paragraph to emphasize that the difference among groups is not statistically significant. We also changed the average as suggested by the reviewer, using the mean for each species instead of all trios. However, we did not weigh each species as we do not have the sampling variance for all the species with a single trio available.

Previous paragraph:

“Despite an appreciable stochastic element due to the small number of mutations per trio, $\mu_{\text{generation}}$ is significantly different among the four major classes of vertebrates, with the highest mean estimates obtained for reptiles (mean of all trios of the group: 1.04×10^{-8} , 95% CI of the mean = 6.49×10^{-9} , 1.42×10^{-8}), intermediate values obtained for mammals (mean of all trios of the group: 7.77×10^{-9} , 95% CI of the mean = 7.11×10^{-9} , 8.42×10^{-9}) and fish (mean of all trios if the group 6.31×10^{-9} , 95% CI of the mean = 4.78×10^{-9} , 7.83×10^{-9}) and the lowest estimates obtained for birds (mean of all trios of the group: 8.08×10^{-9} , 95% CI of the mean = 6.23×10^{-9} , 9.93×10^{-9}).”

New paragraph:

“ On average, the mutation rates per generation are higher in reptiles (average of all species 1.17×10^{-8} , 95% CI of the mean = 5.34×10^{-9} , 1.80×10^{-8}) and birds (average of all species 1.01×10^{-8} , 95% CI of the mean = 6.10×10^{-9} , 1.42×10^{-8}), than in mammals (average of all species 7.97×10^{-9} , 95% CI of the mean = 7.04×10^{-9} , 8.90×10^{-9}) and fishes (average of all species 5.97×10^{-9} , 95% CI of the mean = 4.39×10^{-9} , 7.55×10^{-9}). However, the difference among the four major classes of vertebrates is not overall statistically significant (ANOVA: $F = 1.86$, $p = 0.15$).”

Differences in average:

Group	Sample size trios	Average rate all trio	95% CI min	95% CI max	Sample size species	Average rate species	95% CI min	95% CI max
Reptiles	18	1.04×10^{-8}	6.49×10^{-9}	1.42×10^{-8}	6	1.17×10^{-8}	5.34×10^{-9}	1.80×10^{-8}
Mammals	72	7.77×10^{-9}	7.11×10^{-9}	8.42×10^{-9}	36	7.97×10^{-9}	7.04×10^{-9}	8.90×10^{-9}
Fish	19	6.31×10^{-9}	4.78×10^{-9}	7.83×10^{-9}	8	5.97×10^{-9}	4.39×10^{-9}	7.55×10^{-9}
Birds	42	8.08×10^{-9}	6.23×10^{-9}	9.93×10^{-9}	18	1.01×10^{-8}	6.10×10^{-9}	1.42×10^{-8}

Lines 267-269 – would be good to give CIs on these, as the bird estimates are extraordinarily low.

Response: We have now added the CIs for the estimates presented here.

“...Texas banded gecko at 1.96×10^{-8} mutations per site per year (95% CI = 1.23×10^{-8} , 2.83×10^{-8}), while the lowest $\mu_{\text{yearly_modeled}}$ estimates are obtained for two bird species, the griffon vulture and the snowy owl, both with $< 0.18 \times 10^{-9}$ mutations per site per year (snowy owl: $\mu_{\text{yearly_modeled}} = 0.16 \times 10^{-9}$, 95% CI, 0.05×10^{-9} , 0.34×10^{-9} ; griffon vulture: $\mu_{\text{yearly_modeled}} = 0.17 \times 10^{-9}$, 95% CI, 0.07×10^{-9} , 0.32×10^{-9}).”

Line 283 – define PGLS

Response: This was missing and we have now added the definition at the first occurrence.

Figure 2 – the two axes should be given the same scale, and the null line of equality should be added to show if the observed rates are in fact quantitatively consistent with the estimated substitution rates, or systematically under- or overestimate them. It looks like the majority of points will be below the line, which would mean that the genomic data are somehow overestimating the mutation rate. Given the dispersion of the data, the plots (and the significance tests) might also be more meaningful on a log scale?

Response: We have now changed figure 2 to add this useful observation of the reviewer. We have also added a supplementary figure with a log scale.

Lines 302-306 – The authors need to state why the correlations here are not statistical artifacts. The issue is that generation time is seemingly used to estimate the mutation rate that is applied, introducing potential circularity (and also that maturation time must be correlated with generation time, making these two analyses nonindependent). This becomes relevant in the next section as well.

Response: We agree with the reviewer that those correlations could be biased by different joint effects. Indeed, generation and maturation time are correlated (adjusted $r^2 = 0.77$ p-value $< 2.2 \times 10^{-16}$). This is why we also did the Phylopath analysis which disentangles the causality in the case of co-dependence of the variables.

Moreover, we agree that there is a certain circularity between the modeled rate and the generation time. However, this correlation was also present between the observed per-generation rate (before modeling) and the generation time. Nevertheless, we used the modeled rate as we believe this is closer to the species rate.

Figure 3 – axes labels need time units (presumably years).

Response: We have added the units on the two panels where it was missing.

Typos – lines 79, 401, and probably more that need to be tidied up.

Response: We have removed those two.